# Combining Neural Networks and Data Assimilation to enhance the spatial impact of Argo floats in the Copernicus Mediterranean biogeochemical model

Carolina Amadio[1], Anna Teruzzi[1], Gloria Pietropolli[1,2], Luca Manzoni[2], Gianluca Coidessa[1], and Gianpiero Cossarini[1]

[1]Istituto Nazionale di Oceanografia e di Geofisica Sperimentale – OGS, Trieste, 34100, Italy
[2]Dipartimento di Matematica e Geoscienze, Università degli Studi di Trieste H2bis Building, Via Alfonso Valerio 12/1, 34127 Trieste, Italy;

**Correspondence:** C. Amadio (camadio@ogs.it)

**Abstract.** Biogeochemical Argo (BGC Argo) float profiles provide substantial information for key vertical biogeochemical dynamics and are successfully integrated in biogeochemical models via data assimilation approaches. Although results on the BGC-Argo assimilation are encouraging, data scarcity remains a limitation for their effective use in operational oceanography.

To address availability gaps in the BGC-Argo profiles, an Observing System Experiment (OSE), that combines Neural Network (NN) and Data Assimilation (DA), has been performed here. NN was used to reconstruct nitrate profiles starting from oxygen profiles and associated Argo variables (pressure, temperature, salinity), while a variational data assimilation scheme (3DVarBio) has been upgraded to integrate BGC Argo and reconstructed observations in the Copernicus Mediterranean operational forecast system (MedBFM). To ensure high quality of oxygen data, a post-deployment quality control method has been developed with the aim of detecting and eventually correcting potential sensors drift.

The Mediterranean OSE features three different setups: a control run without assimilation; a multivariate run with assimilation of BGC-Argo chlorophyll, nitrate, and oxygen; and a multivariate run that also assimilates reconstructed observations.

The general improvement of skill performance metrics demonstrated the feasibility in integrating new variables (oxygen and reconstructed nitrate). Major benefits have been observed in reproducing specific BGC process-based dynamics such as the nitracline dynamics, primary production and oxygen vertical dynamics.

The assimilation of BGC-Argo nitrate corrects a generally positive bias of the model in most of the Mediterranean areas, and the addition of reconstructed profiles makes the corrections even stronger. The impact of enlarged nitrate assimilation propagates to ecosystem processes (e.g., primary production) at basin wide scale, demonstrating the importance of the assimilation of BGC-profiles in forecasting the biogeochemical ocean state.

## 1 Introduction

The Argo programme appears to be one of the better examples of countries and human resource capacities in working together to provide global data coverage (Miloslavich et al., 2019) that supports the investigation of the present (analysis), future (fore-

cast) and past years (reanalysis) ocean state conditions. In the last 10 years, the increase of in situ observations from autonomous platforms (Johnson et al. 2013 and Johnson and Claustre 2016) has opened up new perspectives for biogeochemical oceanographers. Indeed, BGC Argo (Argo 2023) has yielded new insights in describing the interior of the global ocean (Le Traon, 2013)

and key processes such as the deep chlorophyll maximum (Mignot et al. 2014, Barbieux et al. 2019, D'Ortenzio et al. 2020, Ricour et al. 2021, and Barbieux et al. 2022), nutrients vertical fluxes (Taillandier et al. 2020 and Wang et al. 2021b), carbon exports (Dall'Olmo and Mork 2014 and Wang and Fennel 2023) and oxygen dynamic (Capet et al., 2016).

With approximately 270,000 profiles worldwide (as of July 2023), oxygen ($O_2$) is currently the most commonly measured variable. The count of $O_2$ profiles is double that of suspended particles and chlorophyll, and more than four times that of

30 nitrate, downwelling irradiance, and pH (source: https://biogeochemical-argo.org). Since 2019, the availability of nitrate and chlorophyll profiles has progressively decreased due to the high cost of the sensor (Dall'Olmo personal communication). The number of oxygen profiles instead decreased initially (2019-2022), but since 2022 is stable or slightly increasing. In the future, Argo Italy envisages mounting oxygen sensors on all Argo floats in the Mediterranean Sea (Discussion in the workshop on "Copernicus Marine requirements for the in situ Observing System", 14-15 September 2023).

The BGC-Argo data are distributed by the Global Data Assembly Centres (GDACs, e.g., Coriolis, NOAA) in Real Time (RT) Adjusted (AM) and Delayed Mode (DM). The quality of AM data is controlled within 24 hours using internationally agreed-upon and automatic quality-control (QC) procedures, while DM data are generally distributed a few months later (nearly six months) in a more rigorous form (Li et al., 2020). The QC tests, conducted across all the data mode levels, aim to assign a quality flag to every observation. Data labeled as 1, 2, 5, and 8 are categorized as good, probably good, changed, and

interpolated value, respectively. The flag 9 indicates missing data, while flags 3 and 4 denote data as probably bad or bad.

In the case of oxygen, the QC mainly perform at the surface, along the entire vertical profiles and along the trajectory (Thierry and Bittig, 2021), excluding specific tests at depth. The implementation of $O_2$ QC tests is mainly devoted to improving the long-term reliability and accuracy of autonomous measurements (Sauzède et al., 2017), particularly concerning sensor drift (the optode drift).

When the sensor drift exists, it is higher in the storage, out of the water, than during the deployment. As described in Takeshita et al. (2013) and in Maurer et al. (2021), raw oxygen data from floats may exhibit errors of up to 20% in terms of oxygen saturation (at the surface) due to sensor drift occurring during the storage. This drift is typically corrected by multiplying the oxygen concentrations for a gain factor term that is derived from a reference dataset (Johnson et al., 2015). Despite efforts to correct drift during storage, which may enhance accuracy by 5-10%, it is likely that an in situ (or during deployment) drift is

still observed. For instance, Maurer et al. (2021) observed drift rates in about 25% of the 126 floats analyzed for the Southern Ocean Carbon and Climate Observations and Modeling (SOCCOM) project. These drift rates spanned a total range of -1.1 to 1.2% per year, with a standard deviation of 0.65% per year. Similarly, Bushinsky et al. (2016) found the presence of a drift rates in about 70% of the floats deployed in the Northern Pacific Ocean. Notably, both positive and negative drift rates were observed across various studies, including those by Johnson and Claustre (2016) , Bushinsky et al. (2016), Bittig et al. (2018a)

and Maurer et al. (2021).

The development and dissemination of a post-deployment oxygen QC aims to avoid spurious results (Wang et al., 2020) and to distinguish between ocean signals or trends (e.g., deoxygenation) from potential drifts. This allows to obtain more robust datasets suitable for specific numerical modelling applications.

Aiming at optimally combining observations and model information to obtain a closer description of reality, the data assimilation (DA) underpins decades of progress in ocean prediction (Geer, 2021). On one hand, progresses began with an increase in the number of available observations over the past decade encompassing both the number of measured variables and the total observations used for model tuning (Wang et al. 2020, Yumruktepe et al. 2023 and Wang and Fennel 2023) and validation (Terzić et al. 2019, Salon et al. 2019 and Wang et al. 2021a). On the other hand, DA scheme were progressively updated to enable multivariate and multiplatform assimilation (Cossarini et al. 2019, Teruzzi et al. 2021 and D'Ortenzio et al. 2021), retrieve associated uncertainty in prediction models, and solve problems connected to uneven distribution and/or scarcity of the observations (Buizza et al., 2022).

In recent years, data assimilation (DA) techniques have increasingly incorporated neural network (NN)-based tools. The main strength of NN algorithms lies in their ability to approximate continuous functions (Hornik et al., 1989) in remarkably low computational times. These NN-based tools have been integrated into DA frameworks to tackle various DA challenges, such as bias correction (Kumar et al. 2015 and Zhou et al. 2021), reformulation of observation operators (Storto et al., 2021) and cross-calibration (Lary et al., 2018). Furthermore, NN algorithms are frequently used as independent tools, distinct from data assimilation, for generating new products and/or reconstructing datasets (Lary et al., 2018). The use of reconstructed datasets may compensate for potential gaps in observation availability, potentially enhancing the predictive skill of numerical models. As an example, ocean colour (OC) datasets were employed to test Multi-Layer Perceptrons (MLP), the most common NN, for retrieving past and long-term BGC timeseries of phytoplankton and chlorophyll (Martinez et al. 2020a, Martinez et al. 2020b, Roussillon et al. 2023). Moreover, in Sauzède et al. (2016), MLP serves to infer chlorophyll vertical BGC distribution from OC. High performance in predicting biogeochemical states (e.g., oxygen) from physical profiling floats measurements was achieved in Stanev et al. (2022) for the Black Sea.

In Sauzède et al. (2017), a MLP-NN is used to approximate nutrient concentration and carbonate system from physical Argo and BGC-Argo oxygen profiles. The updated version of the method presented in Bittig et al. (2018b) allows for further refinement of this approach with the so-called CANYON-b NN method. A configuration to adapt the global CANYON-b NN in the Mediterranean Sea region is developed by Fourrier et al. (2020). A further update of the application of the MLP method in the Mediterranean Sea is provided in Pietropolli et al. (2023), by achieving a lower error in the nutrients predictions through a larger training dataset, a hyperparameter refinement and a two-step quality control of the input data. Given its potential in predicting nutrient profiles, the MLP-NN model outputs are valuable datasets that can be used to fill the gap in the availability of in situ observations in data assimilation.

In the context of operational oceanography, the biogeochemical modelling component of the Copernicus Marine Service for the Mediterranean Sea (MedBFM) provides analysis, short term forecast (Salon et al., 2019) and long term reanalysis (Cossarini et al., 2021), including the assimilation of satellites OC and BGC-Argo observations (Salon et al., 2019). In the MedBFM, the 3DVarBio variational assimilation scheme, has evolved over time by including a greater number of observation

types and variables. Starting from the first release that included OC data assimilation in the open ocean (Teruzzi et al., 2014), the assimilation has progressively developed to handle coastal OC observations (Teruzzi et al., 2018), chlorophyll and nitrate profiles from BGC Argo (Cossarini et al. 2019 and Teruzzi et al. 2021 respectively). Considering the growing availability of O2 from BGC Argo, this paper presents an additional upgrade of the MedBFM to include BGC-Argo oxygen assimilation, with a novel post-deployment quality control, and the integration of NN reconstructed profiles in the assimilation scheme.

The constant evolution of the observation networks and assimilation capacities requires an updated understanding of the impact of observation on the numerical model result (Gasparin et al., 2019). This can be achieved by using the numerical assimilative models in Observing System Experiments (OSEs) where the impact of existing observations on the model performance is assessed (Le Traon et al., 2019). In this paper, the OSE experiment, which combines data assimilation and neural network in a modular approach, aims to quantify how the Argo and BGC-Argo network can be exploited. The sequential use of the NN and DA schemes provides flexibility in using one module independently of the other, depending on the needs of the overall system (Buizza et al., 2022). The DA module used in this work is the 3DVarBio data assimilation scheme described in Teruzzi et al. 2021 and updated to assimilate BGC-Argo oxygen profiles. The NN module is the NN-MLP described in Pietropolli et al., 2023 for the Mediterranean Sea (hereafter NN-MLP-MED).

Spatial and temporal impacts of the OSE have been evaluated using classic and new skill performance metrics in three two-year (2017-2018) numerical experiments performed using the MedBFM coupled with the 3DVarBio: a control run (HIND) without assimilation; a multivariate run (DAfl) with assimilation of BGC-Argo chlorophyll, nitrate, and oxygen profiles; and a multivariate run that also assimilates the in situ observations and NN reconstructed profiles (DAnn). Given its characterization as a miniature ocean suitable for climate studies (Bethoux et al., 1999) and considering the density of BGC-Argo profiles, the Mediterranean Sea represents an ideal site for conducting Observing System Experiment (OSE) studies to assess the feasibility of assimilating BGC-Argo profiles and analyzing their impacts.

Indeed, the Mediterranean Sea is an anti-estuarine semi-enclosed sea (Pinardi et al., 2015) with a complex overturning circulation. This circulation consists of horizontal mesoscale and sub-basins scale gyre structures, transitional cyclonic and anticyclonic gyres and eddies. These dynamics are influenced by bathymetric features interconnected by currents and jets (Oddo et al., 2009), along with vigorous vertical velocities. Furthermore, the shallow Sicily Strait, with a depth of approximately 500 meters, separates the Western Mediterranean from the Eastern Mediterranean. This geographical feature allows different processes to dominate in each of the two regions and limits exchanges only between surface and intermediate waters (Pinardi et al., 2015). Even from a biogeochemical (BGC) perspective, the Mediterranean Sea can be roughly subdivided into the Western and Eastern Mediterranean sectors, characterized by an oligotrophic West-East gradient. This gradient results in low nutrient availability at the surface, which is generally insufficient to sustain high phytoplankton biomass (Siokou-Frangou et al. 2010 and Marañón et al. 2021). Additionally, there is a deeper nitracline in the east (>120m) compared to the west (<100m). Chlorophyll has a particular seasonal cycle with pronounced winter/early spring surface blooms only in the western part and a few locations in the eastern part. During summer, a deep chlorophyll maximum follows the stratified and oligotrophic conditions at increasing depth moving eastward (>100m at East and <100m in the West) (Teruzzi et al., 2021). Dissolved oxygen has a subsurface maximum at about 50m, with higher values in the west (partly due to the dependence of oxygen solubility on

temperature). Noticeable differences are observed in the intermediate layers where the oxygen minimum ranges between 300 (west) and 1000 m (east) (Di Biagio et al., 2022).

While the general dynamics of biogeochemical processes can be summarized in a two-basin gradient, it's important to note that mesoscale and sub-mesoscale events can impact the Mediterranean Sea at the sub-basin scale. These events can create intense local dynamics, such as blooms and water column stratification, which are often associated with eddy activities and peculiar vertical circulation. Reproducing these phenomena in numerical model simulations can be more challenging, as they are prone to encountering high model bias or representativeness error.

The paper is organized as follows. After a brief presentation of the OSE approach, each component and the experimental setup are described in detail (Section 2). In the following section (Section 3), we describe the results of the novel NN-MLP-MED and the assimilation simulations by using different skill metrics to assess model capability in reproducing the main biogeochemical seasonal dynamics. A discussion of some key issues involved in the NN and DA is provided in Section 4, then the paper closes with some final remarks (Section 5).

## 2 Methods

A novel combined Neural Network (NN-MLP-MED) and Data Assimilation (3DVarBio) approach is included in the Mediterranean MedBFM model system to integrate BGC-Argo and NN reconstructed profiles into biogeochemical simulations of the Mediterranean Sea.

Our OSE experiment is based on a sequential modular approach (Buizza et al., 2022) consisting of a post-deployment quality control method of O2, hereafter QC O2 procedure, a trained multi-layer perceptron NN (Pietropolli et al., 2023) and a data assimilation scheme (the 3DVarBio variational scheme of MedBFM, Figure 1).

The first two modules, QC O2 and NN-MLP-MED, uses BGC-Argo and Argo datasets as input. The 3DVarBio module takes the enhanced dataset, quality checked O2 (QCed O2) and reconstructed nitrate (recNO3, Figure 1), as input.

In the following sections, we introduce the components of the MedBFM system, including the transport model (OGSTM, Foujols et al. 2000, Lazzari et al. 2012 and Lazzari et al. 2016 ) and the biogeochemical flux model (BFM, Vichi et al. 2007a and Vichi et al. 2007b). Additionally, we describe the novel modules, namely the QC O2 procedure and the NN-MLP-MED scheme. Furthermore, we outline the dataset, which comprises BGC-Argo and NN reconstructed datasets, and discuss the revised 3DVarBio approach.

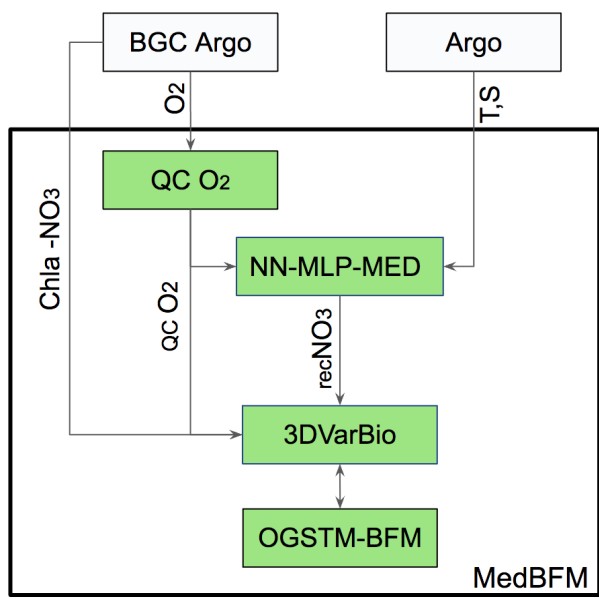

**Figure 1.** Flowchart of the NN-MLP-MED and DA approach. In green boxes: the modules. In plain boxes: the datasets. Arrows refers to Argo (temperature and salinity) and BGC Argo profiles of chlorophyll (Chla), oxygen (qc$O_2$) nitrate ($NO_3$) and reconstructed nitrate (rec$NO_3$).

## 2.1 The regional model for the Mediterranean Sea, MedBFM

The MedBFM consists of the OGS transport model (OGSTM) based on the OPA 8.1 system (Foujols et al., 2000) and updated according to the Lazzari et al. (2012) and Lazzari et al. (2016) versions, the BFM, Biogeochemical Flux Model described in Vichi et al. (2007a) and Vichi et al. (2007b), and the 3DVarBio variational assimilation scheme as in Teruzzi et al. (2014) and Teruzzi et al. (2018).

OGSTM solves for advection, diffusion, sinking terms, and considers the effects of the free surface and variable volume-layer effects on tracer transport (Salon et al., 2019). It is forced by output variables such as current, temperature (T), salinity (S), and sea surface height from the NEMO3.6 model (Clementi et al., 2017). OGSTM and NEMO3.6 share the same bathymetry and z* grid configuration, as well as open boundary and river conditions (Coppini et al., 2023). Atmospheric forcing, including solar shortwave irradiance and wind stress, is acquired as 2-D daily fields from the European Centre for Medium-Range Weather Forecasts (ECMWF), as detailed by Salon et al. (2019).

The Biogeochemical Flux Model, BFM, is a biomass and functional group based marine ecosystem model. BFM solves governing equations for nine living-organic state variables (diatoms, autotrophic nanoflagellates, picophytoplankton, dinoflagellates, carnivorous and omnivorous mesozooplankton, bacteria, heterotrophic nanoflagellates, and microzooplankton, macronutrients (nitrate, phosphate, silicate and ammonium ) and labile, semi-labile, and refractory organic matter and oxygen. In addition, the BFM includes a carbonate system model (Cossarini et al. 2015a and Canu et al. 2015).

## 2.2 3DVarBio data assimilation scheme

Based on 3DVarBio (Teruzzi et al. 2014, Teruzzi et al. 2018, Cossarini et al. 2019 and Teruzzi et al. 2021), the assimilation
module adopted in the present work integrates oxygen, chlorophyll and nitrate to update all the assimilated variables as well as
all the phytoplankton biomasses and phosphate.

The 3DVarBio is a variational data assimilation scheme (Teruzzi et al., 2014) based on the minimization of a cost function
*(J)*. This function comprises two terms: (i) the misfit between the model background ($x_b$) and the model control state variable,
or analysis (i.e., the assimilation result $x_a$) and (ii) the mismatch between the observations (y) and the analysis ($x_a$). Both terms
are weighted by their respective error covariance matrices (*B* and *R*) as follows:

$$J(x_a) = (x_a - x_b)^T B^{-1} (x_a - x_b) + (y - H(x_a))^T R^{-1} (y - H(x_a)) \tag{1}$$

Here, the observation operator (*H*) maps the values of model background state in the observation space. Following Dobricic
et al. (2006), the background error covariance matrix, *B*, is factorised as $B=VV^T$ with $V=V_V V_H V_B$. The *V* operators describe
different aspects of the error covariances: the vertical error covariance ($V_V$), the horizontal error covariance ($V_H$), and the state
variable error covariance ($V_B$). $V_V$ is defined by a set of reconstructed profiles evaluated by means of an Empirical Orthogonal
Function (EOF) decomposition applied to a validated multi-annual 1998-2015 run (Teruzzi et al., 2018). EOFs are computed
for 12 months and 30 coastal and open sea sub-regions in order to account for the variability of biogeochemical anomaly
fields. $V_H$ is built using a Gaussian filter whose correlation radius modulates the smoothing intensity. As in Cossarini et al.
(2019), in this work the correlation radius is non-uniform, direction-dependent, and ranges between 12 and 20 km (16 km on
average). $V_B$ operator consists of prescribed monthly and sub-region varying covariances among the biogeochemical variables
(e.g., nitrate to phosphate). Specifically, for the assimilation of chlorophyll, the $V_B$ operator includes a balance scheme that
maintains the ratio among the phytoplankton groups and preserves the physiological status of the phytoplankton cells (i.e.,
preserves the internal ratios between the chlorophyll, carbon and nutrients as described in Teruzzi et al. 2014).

The operators $V_V$ and $V_B$ of the 3DVarBio have been updated for the assimilation of oxygen. $V_V$ involved the calculation
of specific EOF profiles for oxygen including a localization function to avoid unrealistic corrections due to possible spurious
error covariances in the deepest part of the water column.

$V_B$ included only a new direct relation for oxygen (i.e., oxygen assimilation update only the oxygen itself), given that it has
been shown that it barely affects other variables (Skakala et al., 2021). In the BFM model equations, few formulations depend
on oxygen concentration (e.g., nitrification). Indeed, when the euphotic zone of the open ocean is well oxygenated, oxygen
dynamics have a limited impact on the biogeochemical cycles.

The assimilated observations consist of the QCed BGC-Argo dataset listed in Table 1. Oxygen and nitrate profiles in the
0-600 m layer are used in the assimilation, while chlorophyll is assimilated in the 0-200 m layer.

The observation error covariance matrix R is diagonal with a monthly varying error in chlorophyll (Cossarini et al., 2019).
In both nitrate and reconstructed nitrate profiles, the observation error remains constant over time and increases along the
vertical direction. Within the 0-450 m layer the error is set at 0.24 $\mathrm{mmol\ m^{-3}}$ as in Mignot et al., then linearly increases up to
0.35 $\mathrm{mmol\ m^{-3}}$ between 450 and 600 m (the maximum assimilation depth). This adjustment aims to prevent inconsistencies

between the lower part of the assimilated layer (450-600m) and the deeper layer of the water column (below 600m). Although the accuracy of the reconstruction of profiles is $0.87 \, \mathrm{mmol \, m^{-3}}$ (Pietropolli et al., 2023), we decided to not use different values of error for the two nitrate subsets in order to show the highest potential impact of the OSE.

Observation error for oxygen is set to $5 \, \mathrm{mmol \, m^{-3}}$ in the upper 200 meters of depth and gradually increases to $20 \, \mathrm{mmol \, m^{-3}}$ in correspondence of the maximum assimilation depth. These values correspond to the uncertainty associated with the oxygen dataset described in Feudale et al. (2022).

## 2.3   The Architecture of the Neural Network module and the Reconstructed Nitrate Dataset

The NN-MLP-MED (Pietropolli et al., 2023) is the evolution of previous MLP architectures developed to predict low-sampled
variables (e.g., nutrients) starting from high-sampled ones (e.g., temperature) (Sauzède et al. 2017 , Bittig et al. 2018b and Fourrier et al. 2020).

NN-MLP-MED is a deterministic Feed-Forward Neural Network based on a MLP structure. It consists of the merging of 10 different MLP architectures, each one with the same input and output features, composed by two hidden layers with varying numbers of neurons per layer. The final prediction resulting from the NN-MLP-MED is the mean of all the predictions of
these components. The data flow of the MLP-based approach follows the forward direction from the input to the output layers through the neurons which compose the layers. In our OSE experiment, the trained NN-MLP-MED reconstructs nitrate profiles from sets of temperature, salinity, oxygen, date, latitude and longitude BGC-Argo profiles.

The NN-MLP-MED introduces several innovative features compared to the mentioned methods (e.g., CANYON-Med; Fourrier et al. 2020) leading to improved results.

Firstly, the input dataset encompasses a larger sample size and broader coverage of the Mediterranean Sea region. The EMODnet (European Marine Observation and Data Network) data collection, as described by (Buga et al., 2018), consists of multi-platform data gathered from different research cruises and monitoring activities in Europe's marine waters and global oceans. This dataset is characterized by its multivariate nature, including various biogeochemical observations such as chlorophyll, nitrate, phosphate, dissolved oxygen, DIC, and alkalinity, collected between 1999 and 2018. Additionally, this dataset
is further enriched with in situ observations spanning the period from 1999 to 2016, as detailed in Lazzari et al. (2016) and Cossarini et al. (2015b).

Secondly, the input dataset benefits from a two-step quality check process, removing noisy and unreliable samples. The neural network architecture was also modified to enhance prediction performance by accurately selecting a performing non-linear function, adjusting and optimizing the amount of neurons for each layer of the MLP model, and choosing a different
optimization strategy to train the algorithm. NN-MLP-MED also includes a vertical smoothing step (running mean of 5-10 m window) and a climatological adjustment at depth (600 m) derived from the EMODnet dataset (Salon et al., 2019).

The uncertainty of the reconstructed nitrate associated with the EMODnet validation dataset is $0.5 \, \mathrm{mmol \, m^{-3}}$, while it reaches $0.87 \, \mathrm{mmol \, m^{-3}}$ when predicting the BGC-Argo dataset (Pietropolli et al., 2023).

After incorporating the NN reconstructed profiles (recNO3), the nitrate dataset used for assimilation expands to 2146 profiles
from the initial 938 nitrate (NO3) profiles (Table 1). Generated by the NN-MLP-MED module, the reconstructed dataset offers

broad spatial coverage across the 16 regions of the Mediterranean Sea (Figure 2), as well as a quite balanced distribution of nitrate data throughout the seasons (Figure 3), with the addition of 218 NN reconstructed profiles of nitrate in winter and 361 in summer, respectively.

## 2.4 BGC-Argo data and the post-deployment QC O2 module

BGC-Argo profiles from 2017-2018 were downloaded from the Coriolis GDAC (Argo 2022, last visited in July 2022). We collected both AM and DM data for oxygen and chlorophyll. For nitrate we selected DM data, while AM data were incorporated after undergoing correction via CANYON-b NN method or using the World Ocean Atlas (WOA18) collection (Garcia et al., 2019) as explained in Johnson et al. (2021). For the three variables we use data flagged as good, probably good, changed and interpolated values (flags 1,2, 5 and 8).

Table 1 reports the total number of BGC-Argo profiles, characterized by a high number of oxygen and chlorophyll data against the relative paucity of nitrate. Figure 2 shows the spatial distribution of BGC profiles of chlorophyll and nitrate across the Mediterranean Sea. The oxygen coverage can be approximated by merging nitrate and reconstructed nitrate profiles locations.

To provide more clarity in analyzing the data availability, the Mediterranean Sea has been divided into 16 sub-basins:

– in the Western Mediterranean Sea: Alboran Sea (alb), South Western Mediterranean west (swm1), South Western Mediterranean east (swm2), North Western Mediterranean (nwm), Northern Tyrrhenian (tyr1) and Southern Tyrrhenian (tyr2).

– in the Eastern Mediterranean Sea: Northern Adriatic (adr1), Southern Adriatic (adr2), Western Ionian (ion1), Eastern Ionian (ion2), Northern Ionian (ion3), Western Levantine (lev1), Northern Levantine (lev2), Southern Levantine (lev3)
Eastern Levantine (lev4) and Aegean Sea (aeg).

All the three BGC variables have a fairly homogeneous spatial coverage between the Western and Eastern Mediterranean Sea, except for few sub-basins not covered (alb, ion1 and adr1 see Figure 2) , and a generally 5-day temporal sampling frequency. Higher sampling frequencies (< 5 days) are registered for the 20% of profiles.

Since oxygen sensors may drift and lose accuracy over time, the accurate determination of dissolved oxygen is typically 260 more challenging and requires some form of correction (Johnson et al., 2015). The loss of accuracy, expressed as a percentage per year, is observed over time, particularly 12 months after deployment (source: https://www.euro-argo.eu).

Deep ocean drift is considered as a proxy for oxygen sensor drift because of the lack of seasonal and annual signals for oxygen at depth (Takeshita et al., 2013). Here, the optode drift is evaluated through non parametric methods (RANSAC and Theil-Sen) at two different depths (600 and 800 m) to avoid possible fake drift detection because of changes in the water 265 masses. Tests are applied when the life of a float is longer than one year. Conversely, if the available float time series is less than one year, the profiles are not corrected because the float lifetime is considered too short to account for in situ sensor drift.

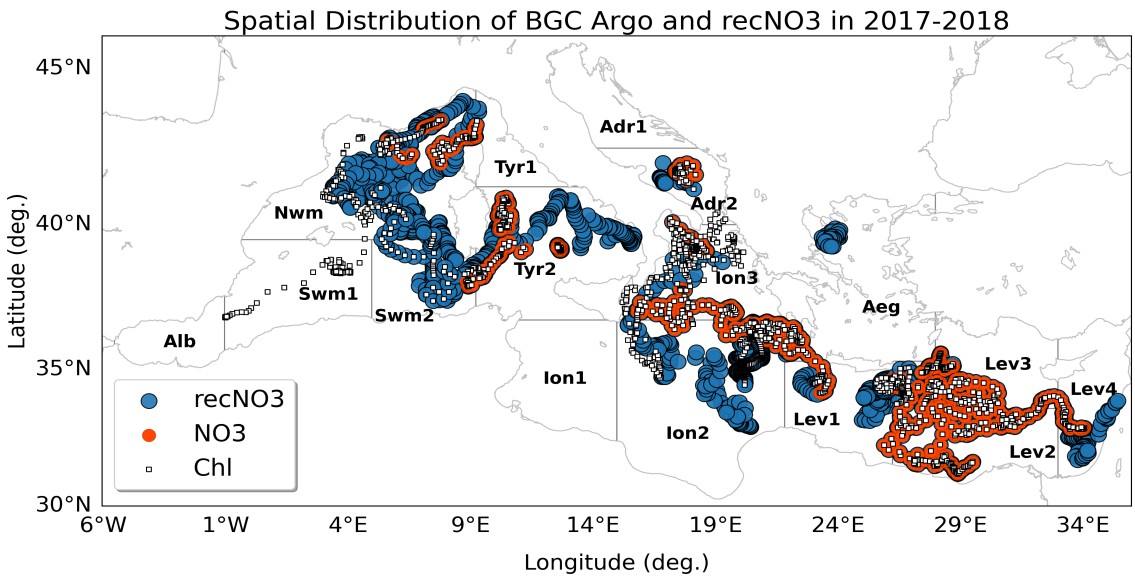

**Figure 2.** BGC-profiles of chlorophyll (Chl, in white), nitrate in situ (NO3, in red) and reconstructed nitrate (recNO3, in blue) assimilated in Mediterranean Sea (2017-2018). Subdivision of the Mediterranean domain in sub-basins used for the validation. According to data availability and to ensure consistency and robustness of the metrics, different subsets of the sub-basins or some combinations among them can be used for the different metrics: lev=lev1+lev2+lev3+lev4; ion=ion1+ion2+ion3; tyr=tyr1+tyr2; adr=adr1+adr2; swm=swm1+swm2.

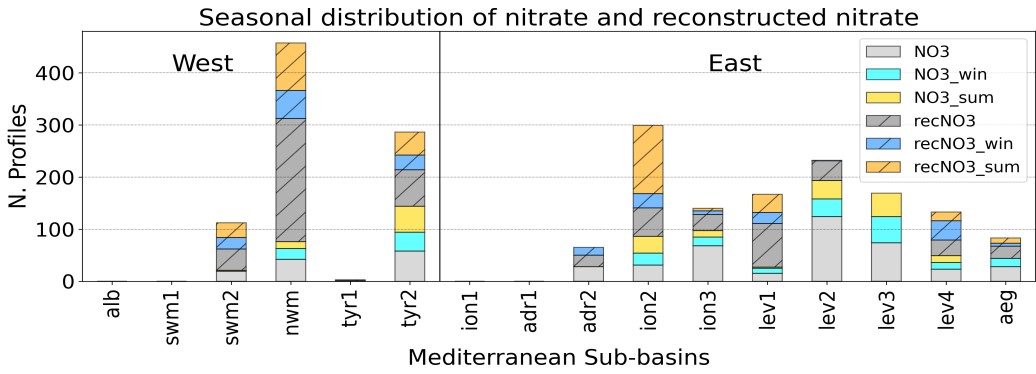

**Figure 3.** Nitrate and reconstructed nitrate profiles seasonal availability. Light gray (autumn and spring), cyan (winter), and yellow (summer) bars represent the availability of nitrate in situ data (used in the run called DAfl). Gray (autumn and spring), light blue (winter), and orange (summer) striped bars indicate the availability of reconstructed nitrate (used in the run called DAnn).

Used for linear and non-linear regression problems, the RANSAC and Theil-Sen methods automatically partition the oxygen dataset into inliers and outliers. In order to avoid possible biases (Dang et al. 2008 and Fischler and Bolles 1981), these methods calculate the drift based on the data subset identified as inliers.

| Test Case | Chl | O2 | NO3 | Updated variables |
|-----------|-----|-----|------|-------------------|
| HIND | – | – | – | – |
| DAfl | 1773 | 1924 | 938 | phyto biomass, NO3 , O2 and PO4 |
| DAnn | 1773 | 1924 | 2146 | phyto biomass, NO3 , O2 and PO4 |

**Table 1.** Summary of the numerical experiments and assimilated BGC-profiles.

In our approach, the presence of a drift is established when all four drift estimates (RANSAC at 600 and 800 m, Theil-Sen at 600 and 800 m) agree in sign and their average value (D_avg) exceeds 1 $\mathrm{mmol\ m^{-3}\ y^{-1}}$. This threshold is chosen on the basis of results in Bittig et al. (2018a). Subsequently, the identified drift is removed from the oxygen profiles. This is achieved by setting the D_avg at 600 meters and linearly interpolating toward the surface, where drift is set equal to zero. As highlighted by Thierry and Bittig (2021), there is a lack of specific tests at depth, although several tests are performed near the surface by the GDACs. The presence of near-surface tests motivates our decision to mitigate the correction's impact at the surface.

## 2.5 Design of numerical experiments

Three numerical experiments are performed to analyze the impact of different assimilation setups. The simulated period is 1.1.2017-31.12.2018, and the MedBFM module setup mostly corresponds to the standard adopted in the Mediterranean Analysis and Forecast biogeochemical system of the Copernicus Marine Service. This set up includes: open boundary conditions in the Atlantic; climatological input of nutrients, carbon and alkalinity for 39 rivers and the Dardanelles Straits; initial conditions from EMODnet dataset (details are provided in Salon et al. 2019); and a 3-years spin up using the 2017 forcings in perpetual mode.

Our experimental setup differs from the standard setup for physical forcing, which is sourced from the Mediterranean Copernicus reanalysis (Escudier et al., 2021), as well as for the initial oxygen conditions. These conditions are derived from the BGC-Argo dataset by generating 16 climatological profiles of oxygen after performing the QC O2 procedure, and then uniformly assigning them to each grid point of the 16 sub-basins shown in Figure 2.

The three simulations, which share the same setup except for the assimilated datasets, are: (1) control run without assimilation (HIND); (2) assimilation of BGC-Argo chlorophyll, nitrate and oxygen (DAfl) and (3) assimilation of additional reconstructed nitrate profiles used to enhance the DAfl assimilative set up (DAnn).

Before integrating data in the 3D-VarBio, the same pre-assimilation assessment described in Teruzzi et al. (2021) is applied to the chlorophyll profiles. Nitrate profiles are rejected if concentration at the surface is higher than 3 $\mathrm{mmol\ m^{-3}}$. At surface, the oxygen profile exclusion is evaluated by calculating the difference between the uppermost oxygen measurement and the oxygen saturation (derived from temperature and salinity data from the Argo dataset as in Garcia et al. 2019). Profiles are excluded when this difference reaches the threshold of 10 $\mathrm{mmol\ m^{-3}}$. At 600 meters, the difference between oxygen and a climatological reference oxygen at depth is calculated. Profiles are excluded when the difference reaches the threshold of

2 times the standard deviation of the same reference dataset. As reference dataset, we chose the EMODnet2018_int data collection that integrates the in situ aggregated EMODnet data (Buga et al., 2018)) and the datasets listed in Lazzari et al. (2016) and Cossarini et al. (2015b). The EMODnet2018_int dataset is available for 16 sub-basins in the Mediterranean Sea (Figure 2).

During the data assimilation, profiles are excluded when innovation exceeds specific threshold rules. For chlorophyll, the threshold is set at $2 \mathrm{mg}\,\mathrm{m}^{-3}$. For nitrate, the thresholds are 1 and $2\,\mathrm{mmol}\,\mathrm{m}^{-3}$ for the 0-50 m and 250-600 m layers, respectively (as in Teruzzi et al. 2021). Oxygen thresholds are 30 and $50\,\mathrm{mmol}\,\mathrm{m}^{-3}$ for the 0-150 m and 150-600 m layers respectively (thresholds are roughly 3 times the standard deviation of the climatology computed on EMODnet data for the different sub-basins). Exceeding values have to be found in at least 5 vertical levels within the specified layers. These exclusions aim to

prevent corrections that could trigger unstable dynamics after the assimilation (Teruzzi et al. 2021, Storto et al. 2011, Sakov and Sandery 2017 and Waller et al. 2018). The excluded profiles range from 0.1% for chlorophyll to less than 1% for nitrate.

## 3   Results

### 3.1   The post-deployment QC O2 module

The product of our QC O2 module is a QCed dataset available at https://zenodo.org/records/10391759.

The QC O2 module enabled the automatic correction of in situ sensor drifts. Of the 40 floats available between 2017 and 2018, we performed the drift analysis on 16 floats, while 24 floats remained unanalyzed due to the limited length of the timeseries. Of these 16 floats, we found a drift in 13: 4 with a positive drift and 9 with a negative drift. For the remaining 3 floats, the drift values were below the prescribed threshold (Section 2.4). At a depth of 600 meters, the absolute average correction for the 13 floats is approximately $4.3\,\mathrm{mmol}\,\mathrm{m}^{-3}$ y. This value aligns with the ranges expressed in terms of sensor

drift percentage in Bittig et al. (2018a) (1-1.5%).

    Figure 4 shows the evolution of oxygen profiles for a quasi-stationary float (6902687) after applying the drift correction. Consistent with findings in various studies (e.g., Bittig et al. (2018a) and Maurer et al. (2021)), the detection of drift by our QC O2 suggests a possible tendency of the optode to slowly degrade over time. After 2 years, the bias due to the drift reaches approximately $5\,\mathrm{mmol}\,\mathrm{m}^{-3}$ (profiles from December 1, 2017 in Figure 4).

The removal of drift brings the oxygen concentration at 600 m closer to the EMODnet climatological data (as shown in Figure 4, green star). This leads us to infer that our drift correction enables the inclusion of more profiles in the assimilated oxygen datasets.

### 3.2   Validation using Satellite and BGC-Argo datasets

Skill performances of the simulations listed in Table 1 are evaluated by comparing model results with (i) the satellite Ma-

rine Copernicus OC product (i.e., non-gap-filled L3 product OCEANCOLOUR_MED_BGC_L3_MY_009_143 from marine.copernicus.eu, last visited in July 2023) of chlorophyll and (ii) BGC-Argo profiles of chlorophyll, nitrate, and oxygen

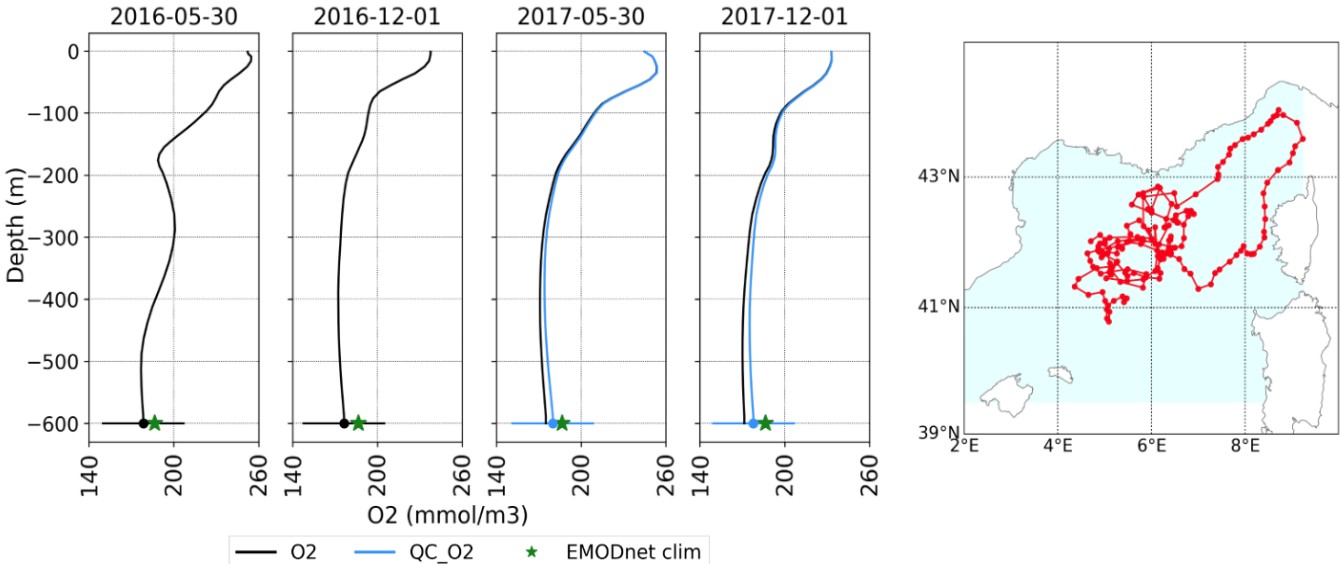

**Figure 4.** Depiction of original (black) and corrected (blue) oxygen profiles for float 6902687 across four selected dates. The green star refers to the EMODnet O2 climatological value in the nwm sub-basin and the horizontal line to the EMODnet O2 standard deviation at 600 m.

(Argo, 2022). The OC L3 satellite products downloaded from the Copernicus Marine Service catalogue are interpolated from 1 km to 1/24° model resolution.

Specifically, we compared the daily model output with the satellite dataset and the model's first guess (i.e., the model state at 1pm before assimilation) with the BGC-Argo profiles. While the use of the first guess is a common practice in data assimilation (Hollingsworth et al., 1986), it is worth to remind that this comparison should be considered as a semi-independent validation, given that two consecutive profiles of the same BGC-Argo float can share a certain degree of correlation in their errors.

The Root Mean Square Error (RMSE) metric is chosen to quantify the model capability to reproduce seasonal variability of the main biogeochemical (BGC) processes at the surface (satellite dataset) or along the vertical column (BGC Argo dataset), such as phytoplankton surface bloom and dynamics during water column stratification.

Indeed, the RMSE is evaluated during winter (from February to April, FMA) and summer (from June to August, JJA) 2017 and 2018 within 16 sub-basins of the Mediterranean Sea (as described in Section 2.4 and in Figure 2) or in an aggregated combination of them.This latter includes six macro-basins: the South Western Mediterranean Sea (Swm) consisting of swm1 and swm2; the North Western Mediterranean (Nwm) represented solely by the nwm; the Tyrrhenian Sea (Tyr), consisting of tyr1 and tyr2; the Ionian Sea (Ion), consisting of ion1, ion2, and ion3; the Adriatic Sea (Adr), consisting of adr1 and adr2; and the Levantine Sea (Lev), consisting of lev1, lev2, lev3 and lev4.

The Winter RMSE concerning the OC chlorophyll in HIND spans between approximately 0.09 to 0.21 mg m$^{-3}$ with a maximum in the alb region (Figure 5). The inclusion of multivariate DA (in DAfl) positively impact the model performances,

reducing surface errors by 6.5% mainly observed in the eastern sub-basins. A further reduction of RMSE (up to 10%) with

345 respect to HIND is then obtained with DAnn highlighting that enlarging the nitrate float network leads to improvements in reproducing surface phytoplankton dynamics. Except for alb and swm1, where no nitrate data (in situ and reconstructed) were available, all the Mediterranean sub-basins exhibit a reduction in RMSE during winter. In the nwm, the RMSE in the DAfl assimilative setup is higher than in the HIND run. However, in DAnn (light-blue striped bar of nwm in Figure 3) the enlarged nitrate dataset positively affects the chlorophyll dynamics at surface.

A generalized slight worsening in the assimilated runs can generally be observed during the summer stratification period and especially the Eastern sub-basins. From DAfl to DAnn, the value of RMSE slightly increases in all sub-basins. These values correspond to an average worsening of about 6% in DAfl and 7.5% in DAnn compared to the HIND run. Despite the introduction of a high number of reconstructed nitrate profiles in some sub-basins (e.g., orange striped lines of nwm and ion2 in Figure 3), this inclusion does not positively impact the summer chlorophyll RMSE at the surface. The RMSE values in summer

are an order of magnitude lower than in winter, reflecting the seasonal chlorophyll variability in the Mediterranean Sea (i.e., the very low values of chlorophyll at the surface).

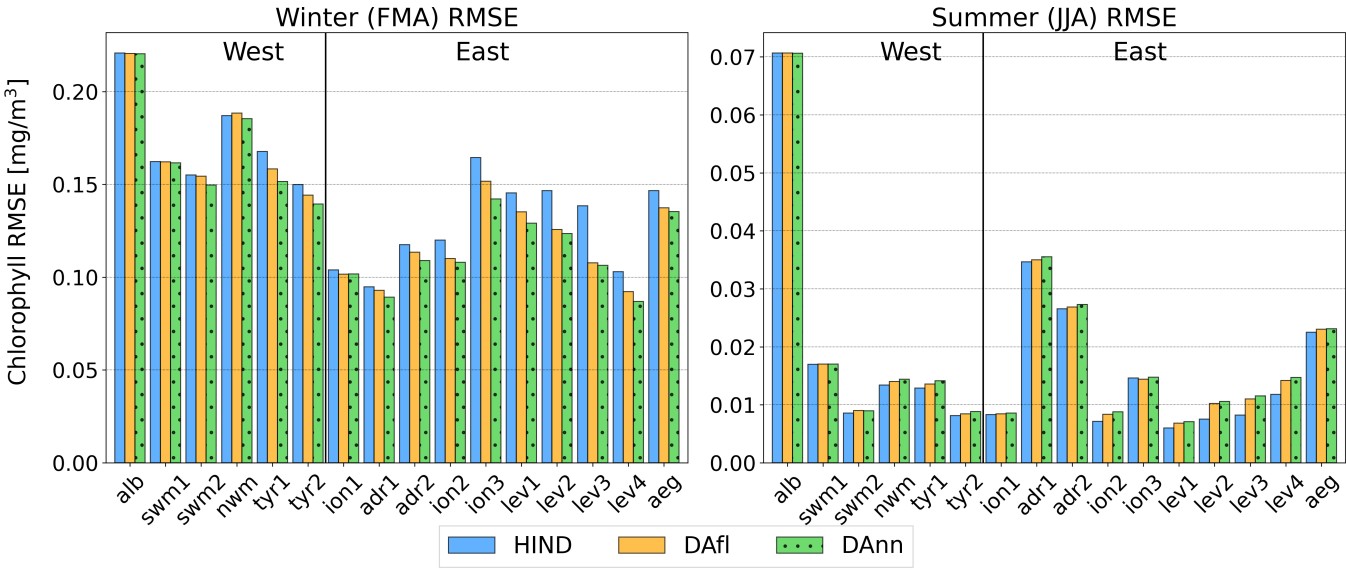

**Figure 5.** Seasonal chlorophyll RMSE of the model runs with respect to Satellite OC observations: Winter bloom and Summer stratification seasons in the Mediterranean Sea sub-basins for the HIND run (light blue), the DAfl run (in orange) and the DAnn run (dotted-green). The black vertical line represents the subdivision of the Mediterranean Sea in West and East sectors.

The RMSE metrics based on BGC-Argo are computed for the six selected aggregated macro-basins and in selected layers (0-10 m, 10-30 m, 30-60 m, 60-100 m, 100-150 m, 150-300 m and 300-600 m) and are shown in Figure 6 for nitrate (top panel), chlorophyll (middle panel) and oxygen (bottom panel). The statistics computed over the aggregate basin provide a more robust

results (e.g., they are computed over a larger number of profiles) even if possible spatial patterns of the errors can be damped. Thus, this choice might limit the analysis on whether/how different nitrate assimilation setups affect chlorophyll and oxygen dynamics (see Section 3.3).

As expected, the assimilation of in situ BGC-Argo considerably improves the quality of modelled nitrate with respect to the HIND run. During winter, the average RMSE reduction is 40% in DAfl, and increases to 46% in DAnn, while in summer the average reduction reaches 59% in DAfl and 63% in DAnn (first row in Figure 6). The most significant RMSE reduction of DAnn compared to DAfl is observed in Nwm and Tyr (0-450 m) during winter, and in Ion (0-100 m) in summer. This impact can be directly ascribed to the profiles availability (Figure 3 ) and additional profiles generate more persistent corrections.

Since the DAfl and DAnn simulation share the same chlorophyll assimilative setup, the RMSE improvements in terms of chlorophyll assimilation can be evaluated comparing the HIND with the DAfl or DAnn simulations (Figure 6 middle panel). We observe slight enhancements in simulating chlorophyll in Nwm (0-100 m) and Lev (0-200 m) during winter and in Tyr, Ion and Lev (50-200 m) during summer (Figure 6 middle panel). Even if phytoplankton dynamics depend on nutrients dynamics, the positive impact of DAnn on nitrate RMSE does not transfer to the the vertical chlorophyll statistics in the DAnn.

Assimilating oxygen profiles enables reducing the model-BGC floats RMSE by about 30% during winter and summer. In winter, the correction involves the whole water columns in the East (Lev and Ion, third row in Figure 6) and deeper layers (150-600 m) in the West (Swm, Nwm) and Adr. In summer, the impact is mainly observed in Tyr, Ion and Lev. The integration of NN reconstructed profiles in the DAnn simulation does not significantly affect oxygen dynamics compared to the DAfl simulation, given that oxygen has already been markedly modified by the O2 assimilation occurring at the same location as nitrate NN-reconstructed profiles.

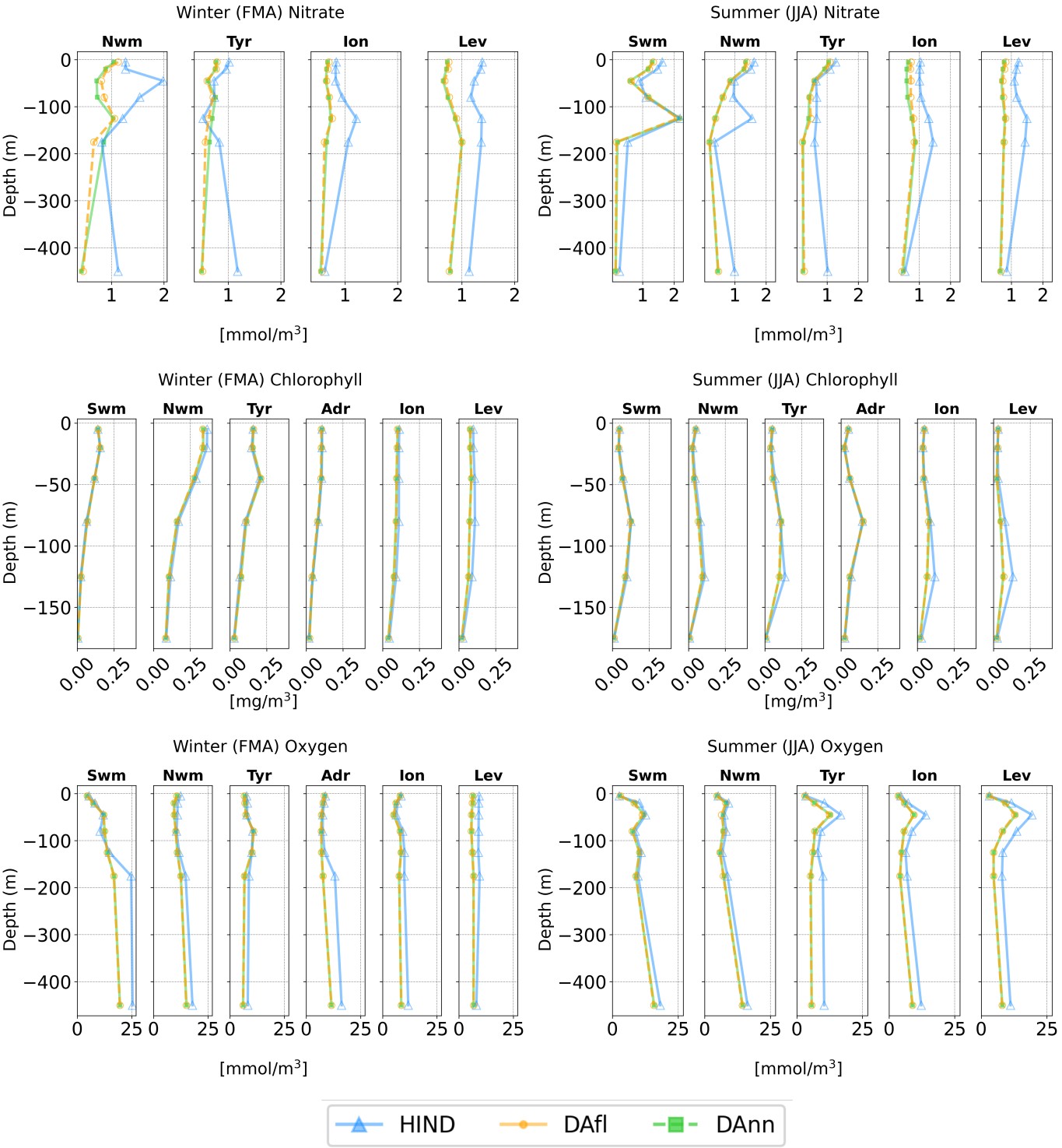

**Figure 6.** Seasonal Nitrate, Chlorophyll and Oxygen profile of RMSE (top, middle, bottom) of the model runs with respect to BGC-Argo observations: Bloom (left) and Stratification (right) seasons in the Mediterranean Sea aggregated sub-basins for the HIND run (pale blue), DAfl run (orange) and DAnn (green).

### 3.3 Integration of NN-MLP-MED and DA modules: the impact

### 3.3.1   Impacts on biogeochemical vertical dynamics

To assess the impact of profile assimilation in changing the vertical gradients of biogeochemical variables, Figures 7, 8, 9 and 10 show the Hovmöller diagrams of the spatial averages of nitrate, phosphate, chlorophyll and oxygen for two selected sub-basins (first and second columns for nwm and ion2 with boundaries indicated in the map of Figure 2) and for the entire Mediterranean Sea (third column). This representation offers additional details on the vertical impact of the reconstructed

nitrate profile assimilation with respect to the validation of Figure 6 that considers only model points corresponding to the location of BGC-Argo profiles. nwm and ion2 represent distinct trophic conditions in the Mediterranean Sea and are also characterized by high number of assimilated reconstructed nitrate profiles (Figure 3). The North Western Mediterranean has higher level of nutrient concentrations and more intense surface blooms in winter (Siokou-Frangou et al. 2010 and Di Biagio et al. 2022). During summer, nwm exhibits a shallow nitracline, higher chlorophyll concentration at the deep chlorophyll

maximum (DCM) and shallow subsurface oxygen maximum (SOM) (first column in Figures 7, 8, 9 and 10). Conversely, the eastern sub-basin is characterized by a deeper nitracline and DCM and more oligotrophic conditions (ion2, second column of Figures 7, 8, 9 and 10).

Considering nitrate, the multivariate assimilation (DAfl) reduces a general positive bias of the model in all the Mediterranean areas (blue pattern in Figure 7). The addition of NN reconstructed profiles makes the corrections stronger. On average, the

nitrate concentration below the nitracline (the depth at which nitrate concentration is $2 \ \mathrm{mmol \ m^{-3}}$) decreases by 8% and 11% in DAfl and DAnn runs, respectively. Both the assimilation runs also exhibit changes of the nitracline depth with more intense deepening in the DAnn simulation. Differences between the assimilation and the HIND run accumulate over time. The rate of this accumulation is highest during the first year and decreases during the second year. These differences remain almost constant in sub-basins with a high number of BGC-Argo and NN reconstructed profiles (e.g., nwm in Figure 7). On the other

hand, considering the ion2 and the whole Mediterranean Sea, which comprises some under-sampled areas (e.g., ion1 and ion3), the effect of DA corrections is still propagating after the two years (third column of Figure 7).

Very similar patterns are also observed in the Hovmöller diagrams of phosphate (Figure 8), which is an updated variable of the multivariate variational assimilation scheme through nitrate-phosphate covariance. In fact, the general negative corrections on phosphate fields are linked to the high positive values of the covariance matrix between nitrate and phosphate (Teruzzi et al.,

2021).

Considering chlorophyll (Figure 9), the main difference between DAfl and HIND is a slight reduction of the DCM chlorophyll concentration (e.g., variation smaller than 5% with respect to HIND simulation) and a correction of the timing of the surface winter blooms (second row in Figure 9). Even if the chlorophyll validation (Figure 6) does not show strong differences between DAfl and DAnn, the basin wide averages of DAnn display more intense corrections with respect to DAfl in terms of

DCM depth and chlorophyll intensity and overall chlorophyll concentration (Figure 9). Over the 0-200 m layer of the whole Mediterranean Sea, the chlorophyll decreases with respect to HIND are 4% and 5% for DAfl and DAnn, respectively.

Corrections on oxygen dynamics after the multivariate assimilation (DAfl, second row in Figure 10) are either positive or negative depending on the area and the period of the year. In particular, corrections are mostly positive in ion2, while the nwm sub-basin shows negative corrections in the subsurface layer and positive ones in the upper layer of the second year. On the Mediterranean basin-wide scale, the average correction is 0.2% for the 0-200 m layer. The addition of the nitrate reconstructed profiles does not alter the correction pattern with an average correction of 0.3%. However, the largest differences between the two assimilation runs can be spotted in areas with a high density of NN reconstructed profiles during summer (e.g., nwm, first column in Figure 10). As observed in the nitrate and chlorophyll Hovmöller diagrams, the assimilation of NN reconstructed profiles causes a decrease of the summer productivity in the DCM layer. Consequently, less oxygen is produced generating the negative changes in the DCM layer in the bottom left panel of Figure 10. Because of the smaller amount of subsequent sinking organic matter, less oxygen is consumed in remineralization processes in layers below the DCM in late summer and autumn, and positive oxygen changes are generated, particularly during 2018.

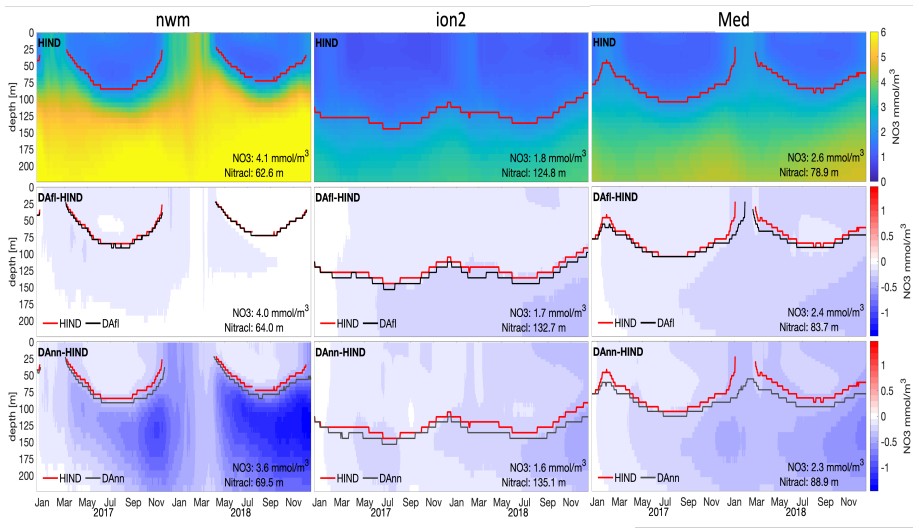

**Figure 7.** Hovmöller diagram of nitrate of HIND simulation (first row) and differences between assimilation runs and HIND (second and third rows) for 2 sub-basins (nwm and ion2) and the Mediterranean Sea (med). Evolution of the depth of nitracline (the depth at which nitrate concentration is 2 mmol m$^{-3}$) for the three runs: red (HIND) and black (DAfl and DAnn) lines. The averages of 0-200m concentration and of nitracline for the simulated period are reported.

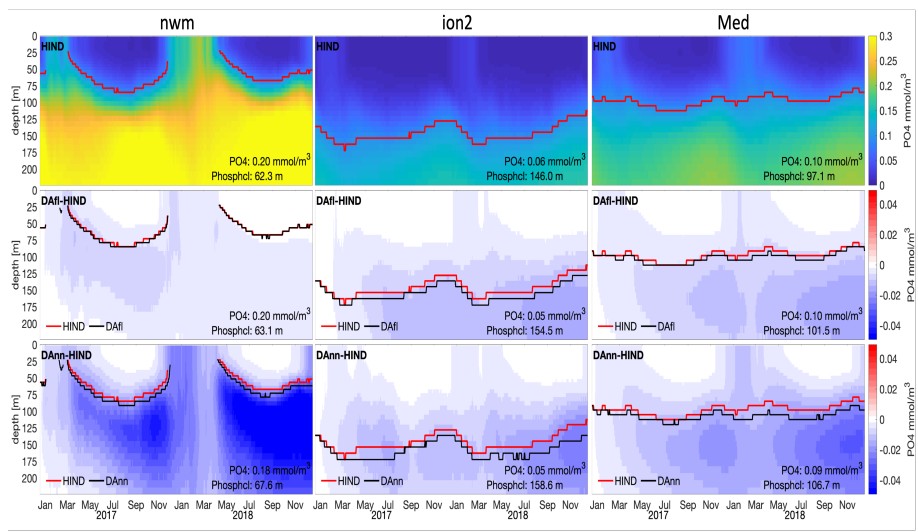

**Figure 8.** Hovmöller diagram of phosphate of HIND simulation (first row) and differences between assimilation runs and HIND (second and third rows) for 2 sub-basins (nwm and ion2) and the Mediterranean Sea (med). Evolution of the depth of phosphocline (the depth at which phosphate concentration is $0.1 \; \mathrm{mmol} \; \mathrm{m}^{-3}$) for the three runs: red (HIND) and black (DAfl and DAnn) lines. The averages of 0-200m concentration and of phosphocline for the simulated period are reported.

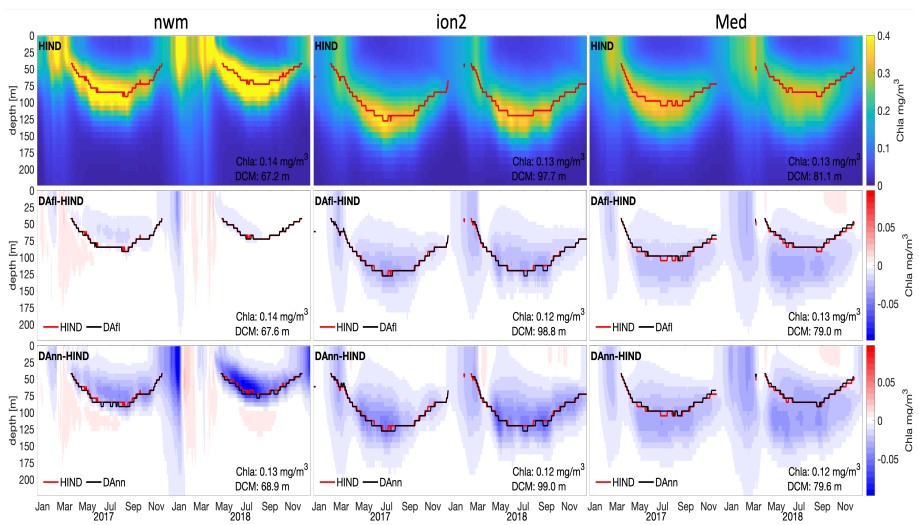

**Figure 9.** Hovmöller diagram of chlorophyll of HIND simulation (first row) and differences between assimilation runs and HIND (second and third rows) for 2 sub-basins (nwm and ion2) and the Mediterranean Sea (med). Evolution of the depth of Deep Chlorophyll Maximum (DCM) for the three runs: red (HIND) and black (DAfl and DAnn) lines. The averages of 0-200m concentration and of nitracline for the simulated period are reported.

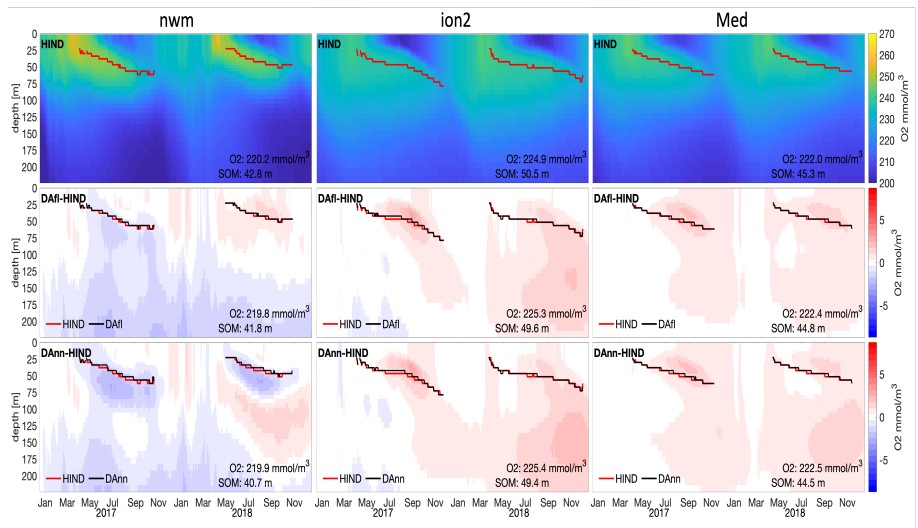

**Figure 10.** Hovmöller diagram of oxygen of HIND simulation (first row) and differences between assimilation runs and HIND (second and third rows) for 2 sub-basins (nwm and ion2) and the Mediterranean Sea (med). Evolution of the depth of subsurface oxygen maximum (SOM) for the three runs: red (HIND) and black (DAfl and DAnn) lines. The averages of 0-200m concentration and of SOM for the simulated period are reported.

### 3.3.2 Impact on ecosystem indicator (net primary production)

Net primary production (NPP) integrates phytoplankton growth and respiration processes which are at the basis of the marine trophic food web. The assimilation of chlorophyll and nitrate together with the updates of phosphate, directly and indirectly affect primary production, as they influence both phytoplankton biomass and nutrient availability. Thus, the comparison of primary production among the three simulations reveals how the assimilation impacts on a key indicator that integrates several marine ecosystem processes. Seasonal maps of net primary production integrated over the 0-200 m layer in the HIND, DAfl and DAnn simulations (Figure 11) confirm that the assimilation's impact varies spatially and temporally.

In the DAfl simulation, the most evident differences in primary production compared to the HIND simulation are located in the Eastern Mediterranean Sea with a decrease of NPP of nearly 10% in the Levantine macro-basin and in the Ionian Sea close to the Greek coast (first and second row of Figure 11). This reduction is particularly pronounced during winter. In the Western Mediterranean the impacts on primary production are less evident in both seasons with a slight reduction (5%) in winter in the Tyrrhenian Sea.

The DAnn simulation shows more pronounced impacts on primary production compared to the DAfl simulation (second and third rows of Figure 11). The main differences between the DAnn and DAfl are highlighted by the black contour line in Figure 11 (differences larger than 15 $\mathrm{mgC\ m^{-2}\ d^{-1}}$). Specifically, during winter, a decrease in NPP is mainly observed in the Nwm, Ion, and Tyr, while in summer the reductions in NPP is observable in the Nwm and Ion.

As shown in Figure 3, basins lev1 and lev4 have a high number of reconstructed nitrate profiles during both winter and summer seasons. This abundance of NN reconstructed profiles contributes to an increase in impact in reproducing the NNP dynamics, which is spatially localized. Conversely, lev2 and lev3 the sub-basins dividing basins lev1 from lev4, contain in situ nitrate and lack of reconstructed nitrate profiles. This lack may spatially limit the impacts that assimilating reconstructed nitrate profiles could have on NPP throughout the entire Levantine region (Lev).

In general, the impact on primary production is greater where nitrate observations or nitrate reconstructed observations are assimilated (Figure 3), suggesting a dynamical bottom-up control of primary production. In fact, the weaker fertilization of the surface layer in DAnn, which occurs for both macronutrients after assimilation (Figure 7 and 8), causes a reduction of the net primary production.

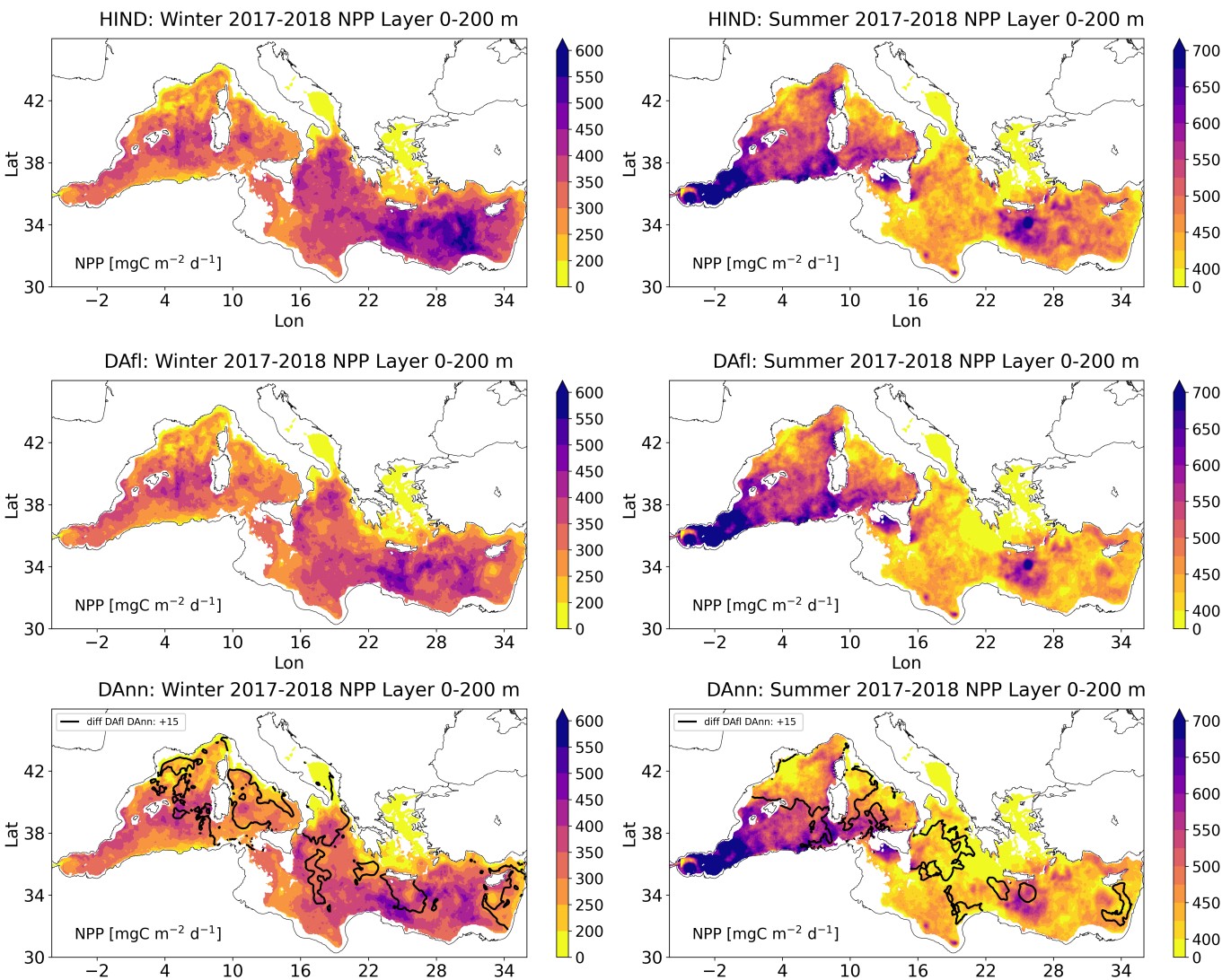

**Figure 11.** Maps of winter (FMA) and summer (JJA) net primary production [NPP, mgC m-2 d-1] in the three simulations: HIND (first row); DAfl (second row). DAnn (third row). Seasonal averages were calculated for the period 2017-2018. The black contour lines in the third row encompass areas where the NPP difference between DAnn and DAfl exceeds $15\ \mathrm{mgC\ m^{-2}\ d^{-1}}$.

### 3.3.3 Impact on Argo Observing system design

Analyzing the departure of an assimilated simulation from a reference solution provides insights into the impact of the observing system design and several data impact indicators can be used (Ford 2021, Teruzzi et al. 2021 and Raicich and Rampazzo 2003). In this work, we adopted the impact indicator $I_{ij}(t)$ as described in Teruzzi et al. (2021). This indicator supports the quantification of the vertically integrated response resulting from assimilating BGC Argo profiles compared to the non-assimilative run:

$$I_{ij}(t) = \frac{|Sim_{ij}(t) - HIND_{ij}(t)|_{0-maxdepth}}{(HIND_{(0-maxdepth)})_{mean}} \qquad (2)$$

HIND is the reference, while Sim refers to one of the different DA set-ups (DAfl or DAnn). $|Sim_{ij}(t) - HIND_{ij}(t)|$ is the absolute difference between two simulations (for each day and grid point), while the subscript *maxdepth* indicates the vertical integrated layer of 0-300 m and 0-600 m for chlorophyll and nitrate respectively.

The indicator $I_{ij}(t)$ quantifies the departure of an assimilated run (DAfl or DAnn) from the reference simulation (HIND) for every grid point within the Mediterranean Sea domain over time, while the $95^{th}$ percentile of the $I_{ij}(t)$ highlights the areas where assimilation markedly increases this model solutions mismatch.

To compare the spatial extent of the Iij(t)95th percentile between the two pairs of runs (HIND-DAfl and HIND-DAnn), we choose a threshold value corresponding to the mean value of the HIND-DAfl and HIND-DAnn maps in Figures 12 and 13 (i.e., 0.1 and 0.4 for nitrate and chlorophyll, respectively) and calculate the areas with values above the threshold.

Figures 12 and 13 show the nitrate and chlorophyll $I_{ij}(t)$ $95^{th}$ percentile of the seasonal indicator in winter (left column) and in summer (right column) in the DAfl (first row) and DAnn (second row) simulations.

In DAfl, the extent of nitrate $I_{ij}(t)$ $95^{th}$ above the threshold of 0.1 is 16.5% and 18.7% in winter and in summer respectively, with a clear spatial distribution mapping the density of BGC-Argo floats. The introduction of NN reconstructed profiles in DAnn make it possible to increase the nitrate impacted areas up to about 35% and 39% in winter and summer respectively. The DAnn impact increase is mainly localized in the western Mediterranean Seas and in the Ion, while the less evident impact in the Lev, especially in summer, is mainly due to the low number of NN reconstructed nitrate in the area.

Chlorophyll impact maps (Figure 13) show that besides the direct impact of chlorophyll profiles assimilation, phytoplankton is also affected by the reconstructed nitrate assimilation. Compared to the threshold of 0.4, the impacted areas increase from 18.2% to 29.8% in winter and from 10.8% to 14.5% in summer in the DAfl and DAnn runs. These results suggest that the inclusion of reconstructed nitrate assimilation has the potential to extend its impact across the majority of the 16 sub-basins of the Mediterranean Sea. However, the scarcity or absence of available data for assimilation prevents us from observing an impact in the marginal seas (Adr and Aeg), the southern part of the Ionian (ion1), and Western sub-basins (alb and swm1).

Oxygen impact maps (not shown) are very similar to the nitrate DAnn maps and do not show differences between the two DA simulations, since the same QC oxygen dataset was assimilated in DAfl and DAnn and the oxygen assimilation largely overcome any other potential model adjustment after nitrate assimilation.

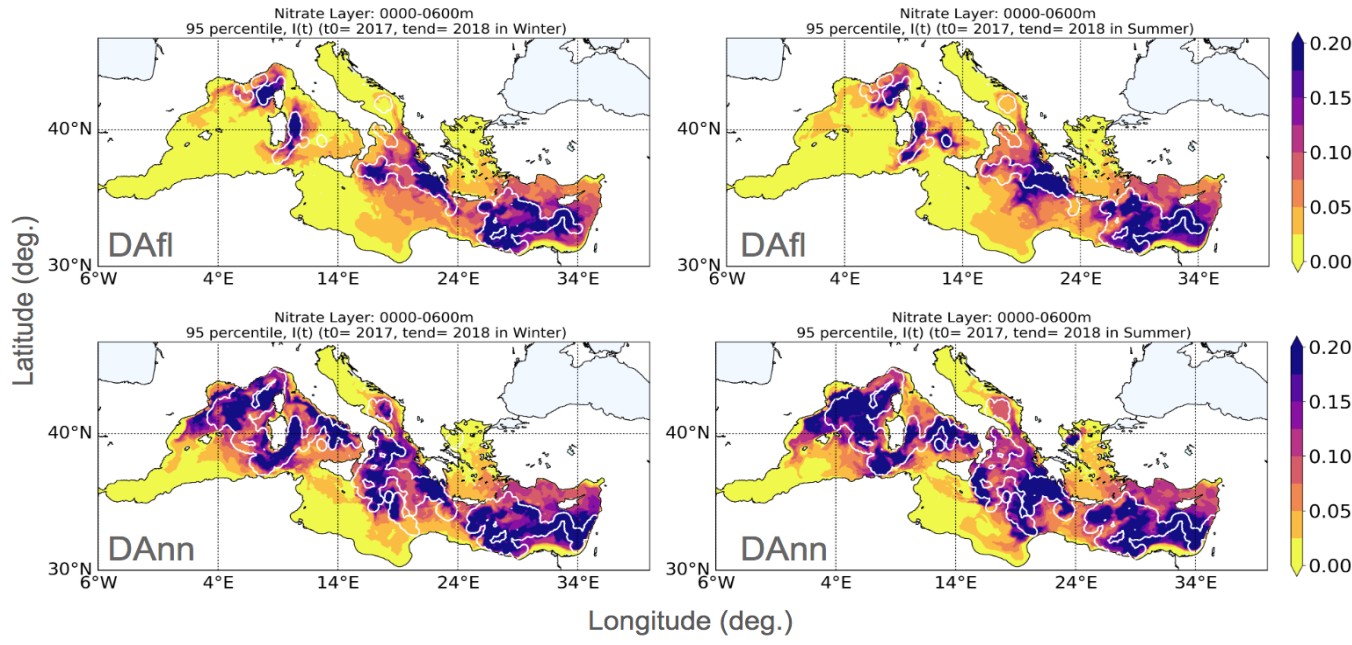

**Figure 12.** Maps of $I_{ij}(t)$ $95^{th}$ percentiles for Nitrate in winter (left column) and summer (right column) in the DAfl (first row) and DAnn (second row); white contour lines identify the areas within three correlation radii from the float profiles.

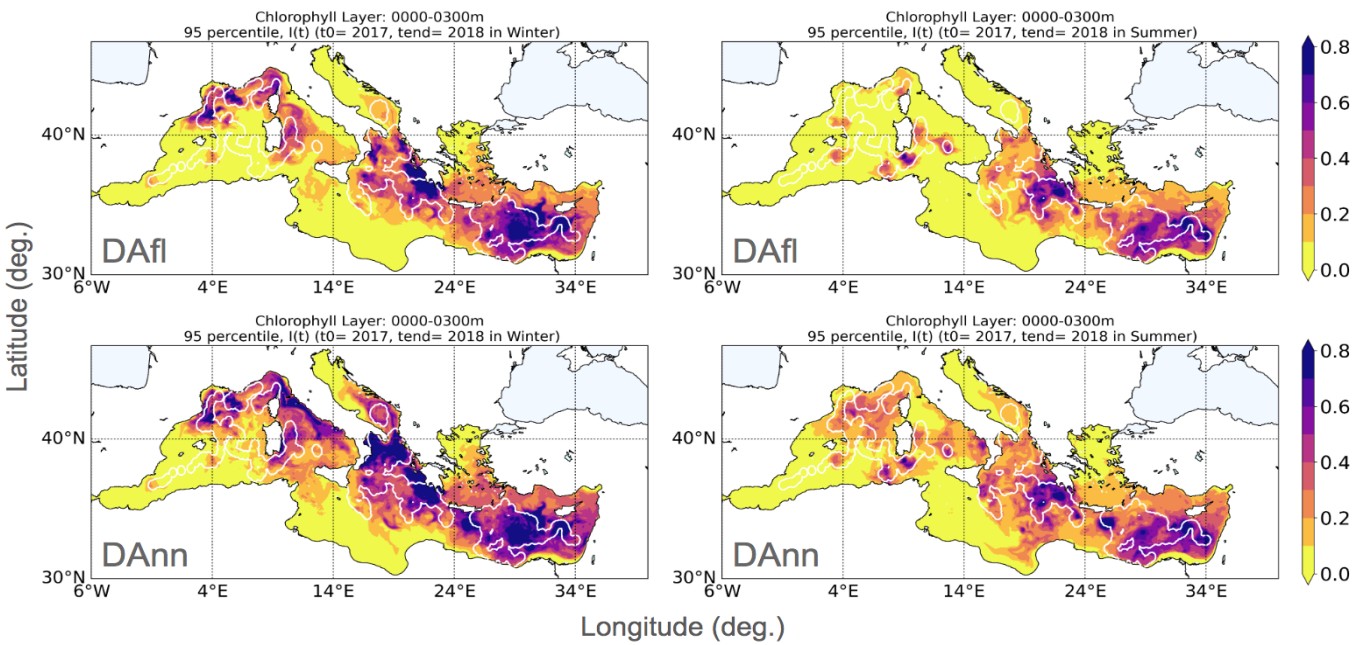

**Figure 13.** Maps of $I_{ij}(t)$ $95^{th}$ percentiles for Chlorophyll in winter (left column) and summer (right column) in the DAfl (first row) and DAnn (second row); white contour lines identify the areas within three correlation radii from the float profiles.

## 4 Discussion

Our quality check procedure (QC O2) for oxygen drift detection and comparison with a reference dataset successfully integrates the official BGC Argo (Argo, 2022), making the oxygen BGC Argo a robust and valuable dataset (Amadio et al., 2023) for initial conditions, data assimilation, validation and reconstruction of new datasets. Even if the distinction between real oxygen depletion signals and optode drift can remain problematic without high quality in situ data, we believe that literature and prior knowledge can be used as a baseline for distinguishing drift.

In particular, the oxygen concentration in the mesopelagic layer of the Mediterranean Sea can exhibit basin-scale variability (Mavropoulou et al., 2020) as well as local intense multiyear variability (Sisma-Ventura et al., 2021). For example, one of the most evident signals was the early 1990's East Mediterranean Transient (EMT) associated with the variations in thermohaline circulation. The EMT caused both negative and positive variations (e.g., about $10 \, \mathrm{mmol \, m^{-3}}$ on a decadal time scale) in oxygen levels in the Western and Eastern Mediterranean Sea (Mavropoulou et al., 2020). However, in the last decades, a much smaller inter-annual variability of oxygen in the mesopelagic layer has been observed in both western and eastern basins (Coppola et al. 2018 and Mavropoulou et al. 2020). Therefore, the threshold of $1 \, \mathrm{mmol \, m^{-3}}$ y at 600 and 800 meters appears to be a prudent limit for sensor discriminating sensor drift from real long term signals for our specific application.

Up to now, oceanographer visual checks have been necessary to distinguish ocean signals from sensor drift (Wang et al., 2020) and the ongoing debate about replacing visual checks with automatic statistical procedures is still open. Consequently, our work seeks to contribute by proposing a new tool designed to automatically handle deep ocean signal or optode drift issues. This method can be further developed by applying oxygen drift analysis at fixed isopycnals, in conjunction with analysis at constant isobaths. This approach might allow us to filter out potential oxygen concentration changes caused by floats moving across different water masses.

The assimilation of vertical profiles provides complementary information to satellite ocean colour assimilation ( Verdy and Mazloff 2017 and Cossarini et al. 2019), which remains the most commonly used method in operational systems (Fennel et al., 2019). In fact, the effectiveness of the profiles assimilation, which has the capability to constrain vertical biogeochemical dynamics in subsurface layers (Kaufman et al. 2018, Teruzzi et al. 2021, Ford 2021, Skakala et al. 2021 and Wang et al. 2022), depends on the availability of BGC-Argo data, which are generally insufficient to constrain a basin wide-simulation. Previous findings (Teruzzi et al., 2021) have primarily demonstrated the efficiency of ocean colour assimilation in constraining chlorophyll dynamics, especially during winter and the advantages of assimilating BGC-Argo profiles in summer. Our work highlights the larger and more extensive benefits of profile assimilation during summer due to the incorporation of reconstructed nitrate profiles.

Through the integration of NN and DA, the count of nitrate profiles ingested can potentially be as high as the BGC Argo equipped with an oxygen sensor (i.e., more than double of the nitrate profiles), which corresponds to a density of 1 profile in each 2.5deg x 2.5deg box every 10 days for the 2017-2018 period. This means that seasonal sub-basins scale dynamics (e.g., bloom or stratification) can effectively be constrained, while the mesoscale dynamics can be only locally constrained (D'Ortenzio et al., 2021).

Apart from an increase in the numbers of floats, a further increase of the area impacted from a float assimilation can be optimized by redefining horizontal covariance errors in the data assimilation scheme. Indeed, benefits of non-uniform correlation radius in the horizontal scale have been previously investigated (Cossarini et al., 2019) and additional improvements could be provided by a 3D varying correlation radius (Storto et al., 2014).

Looking at the recent evolution in the availability of BGC-Argo sensors (Figure 14), our combined NN and DA approach would allow keeping the benefits of the BGC-Argo Observing System in the Mediterranean operational system. Even if nitrate and chlorophyll profiles have dramatically decreased after 2020, the assimilation of NN reconstructed profiles can potentially overcome this lack. Nevertheless, as shown in our Observing System Experiment (Figure 12 and 13), there are still undersampled areas by the Argo and oxygen sensors, such as Alboran, Southern Ionian seas and the marginal seas (Northern Adriatic and Northern Aegean Sea) which would require specific deployments.

With respect to previous BGC Observing System Simulation Experiments, (Yu et al. 2018, Ford 2021), we show how to exploit the current Argo and BGC-Argo networks for reconstructing biogeochemical variables.

MLP feed-forward methods to reconstruct biogeochemical variables are good enough (Bittig et al. 2018b, Sauzède et al. 2020 Fourrier et al. 2021 and Pietropolli et al. 2023) to reach our purposes, even if their application to generate smooth and consistent profiles still has some limitations (Pietropolli et al., 2023). The MLP-NN-MED method exhibits a validation error of $0.50 \, \mathrm{mmol} \, \mathrm{m}^{-3}$ for nitrate when used to predict nitrate from the EMODnet data set, and $0.87 \, \mathrm{mmol} \, \mathrm{m}^{-3}$ when used to predict nitrate from BGC-Argo data (Pietropolli et al., 2023). These uncertainties related to the reconstructed nitrate dataset are higher then the one used in our study ($0.24 \, \mathrm{mmol} \, \mathrm{m}^{-3}$) for both BGC Argo and reconstructed profiles.

Thus, while it is reasonable to assign a higher observation error to NN reconstructed nitrate, applying the same error to both in situ and NN reconstructed datasets has resulted in a potential overestimation of the assimilation impact that can be achieved. On the other hand, using a possibly underestimated error could unbalance the assimilation results toward observation overfitting, and we recognize the potential benefits of using different error values for BGC-Argo and reconstructed profiles. Overfitting effects towards observations may similarly derive from our choice of not explicitly including the nitrate representation error. However, our nitrate error definition is an evolution of the approach used in Teruzzi et al. (2021), which demonstrated a well-established balance between assimilation impacts and over-fitting towards the observations.

The larger error in MLP-NN-MED prediction of BGC-Argo profiles derives the fact that the MLP methods, being pointwise based, are unaware of the vertical gradient (e.g., typical shape) of the profiles of biogeochemical variables that they seek to infer. This fact can lead to irregularities and lack of smoothness in the predicted profiles (Pietropolli et al., 2023), which we partly solved by adding a smoothing operator. However, one way to increase the reliability of profile reconstruction would be to include information with a physical meaning from observed data (Buizza et al., 2022). 1D Convolutional Neural Networks represent a viable alternative approach considering their ability to treat the coherence of the 1D signals (e.g., typical shapes of profiles) as shown in Li et al. (2021).

Integration of NN and DA have been tested in several geoscience applications (Buizza et al. 2022, Brajard et al. 2021, Stanev et al. 2022) to infer unresolved spatial scales or reproduce missing data. In our application, the integration of NN, which retrieves a large number of profiles (Pietropolli et al., 2023), and DA, which can apply the correction to all nutrients

through error covariances (Teruzzi et al., 2021), allows spatial and multivariate changes to be captured both at the local scale and across the basin to constrain Mediterranean productivity (Figure 11). Although the corrections take time to extend to the entire basin (Figure 7), our simulations have shown that constraining bottom-up ecosystem processes (e.g., productivity, organic matter sink) has proven effective and might be used in conjunction with the classical ocean colour correction to phytoplankton biomass.

Any plan to learn directly from observations will have to face with some challenges, such as the use of observations whose time and space coverage is uneven or related to specific processes (Geer, 2021). The modular approach followed in this work represents a successful example of exploiting the strengths of neural networks and data assimilation to enhance the observing system impact in the operational biogeochemical system of the Mediterranean Sea.

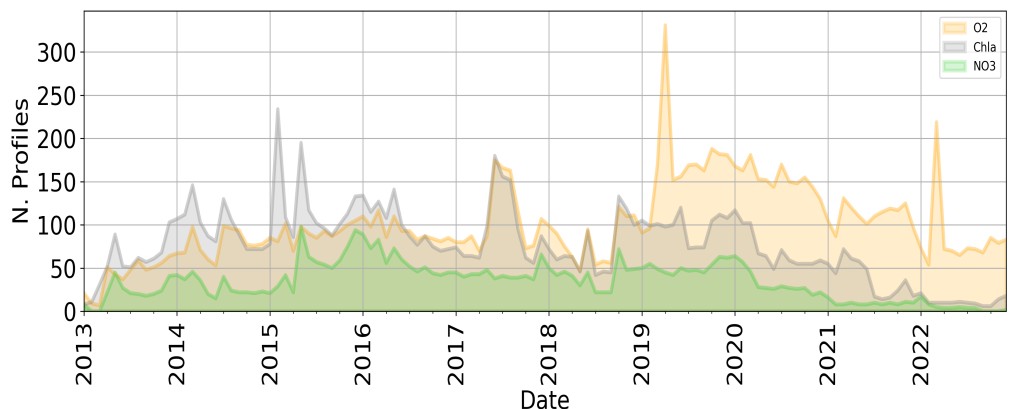

**Figure 14.** Monthly availability of BGC-Argo profiles (number of profiles/month) from 2013 to 2022 for nitrate (green), chlorophyll (grey) and oxygen (yellow).

## 5    Conclusions

Combining deterministic Feed-Forward Neural Network and Data Assimilation to design an Observing System Experiment has enabled demonstrating the enhanced positive impact of profiles assimilation in the Copernicus Operational System for Short-Term Forecasting of the Biogeochemistry of the Mediterranean Sea (MedBFM).

The development of the oxygen QC procedure allowed to statistically deal with optode in situ drift and to derive accurate reconstructed profiles of nitrate, keeping the number of assimilated observations at a much higher level despite the current negative trend in BGC-Argo availability.

The achieved density of BGC profiles provides valuable and additional information to complement ocean colour in the description of seasonal phytoplankton blooms and stratification dynamics at the sub-basin scale.

The assimilation of BGC-Argo nitrate corrects a general positive bias of the model in several Mediterranean areas, and the addition of reconstructed profiles makes the correction stronger.

Together with nitrate assimilation, the phosphate update through error covariances, sustains spatial and multivariate changes that are capable of correcting key biogeochemical processes (e.g., nitracline and deep chlorophyll maximum) and to constrain ecosystem processes (e.g., productivity) at basin-wide scale.

*Author contributions.* CA, AT and GCoss conceived the study. CA and AT updated the 3DVarBio code and GP and LM developed the MLP-NN-MED model. CA and GCoi performed the simulations. CA, AT and GCoss conducted the analysis of the simulation results. CA, AT, GP and GCoss wrote the draft. All authors have approved the manuscript and agree with its submission.

*Competing interests.* The contact author has declared that neither they nor their co-authors have any competing interests.

*Acknowledgements.* This research has been partly supported by the MED-MFC "Mediterranean Monitoring and Forecasting Centre" of Copernicus Marine Service, which is implemented by Mercator Ocean International within the framework of a delegation agreement with the European Union. (Ref. n. 21002L5-COP-MFC MED-5500).

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
