# Peer review of "Combining Neural Networks and Data Assimilation to enhance the spatial impact of Argo floats in the Copernicus Mediterranean biogeochemical model"

_EGUsphere, 2023_

## Author Comment (AC1)

**#REV 1**

Please note that this is a co-review by an early-career and mid-career scientist.

This paper details new processing of BGC-Argo data in the Mediterranean Sea, including oxygen sensor drift correction and the use of a neural network to reconstruct nitrate from other measured variables. An existing data assimilation scheme, previously used to assimilate chlorophyll and (non-reconstructed) nitrate from BGC-Argo, is extended to also assimilate reconstructed nitrate and measured oxygen profiles. Test runs demonstrate a positive impact on model analyses of assimilating these new variables.

The study is novel, of interest to the community, and within scope for Ocean Science. The study is generally well-conceived and well-presented, but there are aspects which should be more clearly explained.

In many places, the manuscript is hard to follow and would benefit from being made clearer. Some specific examples are given in the comments below, but are not exhaustive. As a general example, the use of passive voice, in particular in the methods section, makes it challenging in some parts to distinguish your work from previous studies. As another example, in the introduction the topics that should be introduced are introduced, but the text lacks flow and links between the topics, and so does not lead to the question you are addressing.
The manuscript would also benefit from English language copy editing, but we believe the journal offers this service as standard, so will not list such technical corrections as part of this review.

This paper details new processing of BGC-Argo data in the Mediterranean Sea, including oxygen sensor drift correction and the use of a neural network to reconstruct nitrate from other measured variables. An existing data assimilation scheme, previously used to assimilate chlorophyll and (non-reconstructed) nitrate from BGC-Argo, is extended to also assimilate reconstructed nitrate and measured oxygen profiles. Test runs demonstrate a positive impact on model analyses of assimilating these new variables.

The study is novel, of interest to the community, and within scope for Ocean Science. The study is generally well-conceived and well-presented, but there are aspects which should be more clearly explained.

We appreciate the constructive comments and suggestions from the Reviewer. We present our point-by-point responses to the Reviewer's comments below. The Reviewer's comments are in blue, our responses follow each comment in black. In each response, we detail the changes we propose to make to the manuscript and include the proposed modified text and/or figure (in red).
For clarity, we have numbered some of the reviewers' comments so that similar ones are aggregated to provide a single response. Comments are labeled and highlighted with specific colors to distinguish reviewers (e.g. Rev. 1: comment1a, Rev. 2: comment1b, Rev. 3: → comment1c).

In many places, the manuscript is hard to follow and would benefit from being made clearer. Some specific examples are given in the comments below, but are not exhaustive. As a general example, the use of passive voice, in particular in the methods section, makes it challenging in some parts to distinguish your work from previous studies. As another example, in the introduction the topics that should be introduced are introduced, but the text lacks flow and links between the topics, and so does not lead to the question you are addressing.

Thank you for the comment. We will revise the manuscript having particular attention to the use of the passive voice.

**comment1a**

As another example, in the introduction the topics that should be introduced are introduced, but the text lacks flow and links between the topics, and so does not lead to the question you are addressing.

We will revise the introduction to fill the lack of links between paragraphs.

Some examples are in **comment2a, comment4a, comment5a, comment8a, comment10a**.

The manuscript would also benefit from English language copy editing, but we believe the journal offers this service as standard, so will not list such technical corrections as part of this review.

Thank you for the comment. We will review the manuscript paying more attention to the English language.

**comment2a**

The paper aims to "address availability gaps" (Line 4) but this objective is not clear throughout the paper. The introduction does not clarify how, where, or why the data gaps affect the analysis. Results of the two model runs with and without reconstructed observations clearly show differences, but these are not always linked to the change in coverage.

The 'data availability gap' will be explained more explicitly in the introduction, as proposed hereafter (in bold the new text).

In addition, we will include information on the seasonal availability of the observations that will help to comment on the results throughout the manuscript considering the BGC-Argo data availability.

L29: Among the BGC sensors, oxygen ($O_2$) is currently the most common measured variable with approximately 270,000 profiles worldwide (**as of July 2023-07**), which is double that of suspended particles and chlorophyll and more than four times those of nitrate, downwelling irradiance, and pH (https://biogeochemical-argo.org).

**Since 2019, the availability of nitrate and chlorophyll profiles has been gradually decreasing due to the high cost of the sensor (Dall'Olmo personal communication). The number of oxygen profiles instead decreased initially (2019-2022), but since 2022 is stable or slightly increasing. In the future, Argo Italy envisages mounting oxygen sensors on all Argo floats in the Mediterranean Sea (Discussion in the workshop on "Copernicus Marine requirements for the In-situ Observing System", 14-15 September 2023).**

**comment3a**

Also, the abstract and discussion conclude with a note about Argo data being complementary to satellite ocean colour assimilation, but this study does not show that.

The complementarity with satellite assimilation will be removed from the abstract but we will keep the topic in the discussion modifying the sentence as follows (**comment25a**):

Previous works demonstrated the complementary of vertical profiles with respect to satellite ocean colour assimilation (Cossarini et al. 2019, and Ford et al., 2021, Skákala et al., 2021 and Teruzzi et al., 2021).

Line 20: "Array for Real-time Geostrophic Oceanography" - this acronym form does not appear to be widely or currently used, suggest just using the name "Argo".

Thanks for the comment. We will remove the acronym form.

Line 29: "approximately 270,000 profiles worldwide until now" - better to put "as of [date]" rather than "until now".

OK, we will insert the date as follows (in bold):

Among the BGC sensors, oxygen (O2) is currently the most common measured variable, with approximately 270,000 profiles worldwide **(as of July 2023-07)**, which is double that of suspended particles and chlorophyll and more than four times those of nitrate, downwelling irradiance, and pH (https://biogeochemical-argo.org).

**comment4a**

Line 39: "By improving the accuracy…" is the result of the QC and would therefore fit better at the end of the sentence to increase clarity.
Line 45: "encouraged" replace with "is necessary"

Considering all the comments on the Introduction section (e.g., **comment1a**) we propose the following modified version of the paragraph that does no longer include "By improving the accuracy". Moreover, the term "encouraged" is replaced in order to highlight the final aim of the oxygen QC method:

When the sensor drift exists, it is higher in the storage, out of the water, than during the deployment. As described in Takeshita et al. (2013) and in Maurer et al. (2021), raw oxygen data from floats can have errors of up to 20% in terms of oxygen saturation (at the surface) due to sensor drift during the storage. This drift is generally corrected by multiplying the oxygen concentrations for a gain factor term that is derived from a reference dataset (Johnson et al., 2015). Despite this correction can improve the accuracy up to 5-10%, Maurer et al. (2021) and Bushinsky et al. (2016) found a drift in about 25% (with a mean of -0.07% per year, a standard deviation of 0.65%, and a total range of 1.1 to 1.2% per year) and 70% of the analyzed floats, respectively. Given the logistical challenges in recovering deployed floats, an in situ (or during deployment) drift >1% per year can be likely observed (Bushinsky et al., 2016). Here the drift can be both positive or negative as found in Johnson and Claustre (2016) and Bittig et al. (2018b).

The development and dissemination of a post-deployment oxygen QC aims to avoid spurious results  (Wang et al., 2020) and to distinguish between ocean signals or trends (e.g., deoxygenation) and potential drifts. This allows to obtain more robust datasets suitable for specific numerical modelling applications.

Line 46: "optode" was not mentioned before
We will solve the comment by introducing the meaning of optode (before line 46) as follows (in bold):
in line 36:
The implementation of O2 QC is mainly devoted to improving the long-term reliability and accuracy of autonomous measurements (Sauzède et al., 2017) **in particular with respect to the sensor drift (the optode drift).**

**comment5a**
Line 47: Suggestion to link the topics for better flow of the text: say that purely observation-based Argo studies are regional, and using data assimilation has the potential to create a synthesis
Thank you for the suggestion. We propose the following version to better link the paragraphs (in bold the new insertion). The concept of "using data assimilation has the potential to create a synthesis" will be introduced later (L50, see **comment4a and comment6a**).
The development and dissemination of a post-deployment oxygen QC aims to avoid spurious results (Wang et al.,2020) and to distinguish between ocean signals or trends (e.g., deoxygenation) and potential drifts. **This allows to obtain more robust datasets suitable for specific numerical modelling applications.**
**Aiming at optimally combining observations and model information to obtain a closer description of reality,** DA underpins decades of progress in ocean prediction (Geer, 2021).

**comment6a**
Line 50: "DA underpins …" - suggest rephrasing this sentence slightly to articulate more clearly the aims and principles of DA.
Thank you, we propose the following correction:
**Aiming at optimally combining observations and model information to obtain a closer description of reality,** DA underpins decades of progress in ocean prediction (Geer, 2021). **In on one hand, progresses began with an increase in the number of available observations over the past decade (number of measured variables and number of observations) on the other hand DA scheme were progressively updated to be able to perform multivariate and multiplatform assimilation, retrieve associated uncertainty into a prediction model, and solving problems connected to uneven distribution and scarcity of the observations (Buizza et al., 2022).**

**comment7a**

Line 54: "NN algorithms" - "NN" hasn't been defined yet in the main text. Need to add at least a couple of sentences introducing neural networks at the start of this paragraph.

Line 54: "match specific DA tasks" - a better phrasing might be something like "have the potential to perform specific tasks related to observation processing and DA". The discussion of the following studies could be made clearer too, as well as stating that it is the method of Pietropolli et al. (2023) that is used in this study.

We will add the extended name and introduce the acronym NN. We will propose the following correction (in bold) to solve both comments referring to L54 (merged above). Your suggestion "*The discussion of the following studies could be made clearer too*", is further developed in the next comment for lines 57-66 (comment8a), however, we prefer to do not add information on the works cited at L. 55-56, since here the aim is to list some examples of NN applications in DA.

Finally, the suggestion "stating that it is the method of Pietropolli et al. (2023) that is used in this study" is fulfilled in a later comment (comment9a)

**In recent years, neural network (NN) algorithms have been increasingly used to solve and analyze specific tasks related to observation processing and DA. The main strength of NN algorithms lies in their ability to approximate continuous functions (Hornik et al., 1989) in remarkably low computational times**. For these reasons, DA techniques have been recently augmented with NN-based tools, e.g., for: bias correction (Kumar et al. 2015 and Zhou et al. 2021), cross calibration (Lary et al., 2018), reformulation of observation operators (Storto et al., 2021) and new product creation or dataset reconstruction (Lary et al., 2018).

comment8a

Line 58: May mention here or later that these examples are time series of chlorophyll, while the use of reconstructed nitrate is novel

Thank you, we propose the following corrections (also on the basis of the previous comment7a). The new insertions are in bold:

As an example, ocean color (OC) datasets were employed to test Multi-Layer Perceptrons (MLP, namely the most common NN) by retrieving past and long-term BGC time series **of phytoplankton and chlorophyll** (Martinez et al. 2020a, Martinez et al. 2020b, Roussillon et al. 2023). Moreover, in Sauzède et al. (2016), MLP serves to infer **chlorophyll** vertical BGC distribution from OC. High performance in predicting biogeochemical states (e.g., oxygen) from physical profiling floats measurements were achieved in Stanev et al. (2022) for the Black Sea.

In Sauzède et al. (2017), an [........] input data.

A further update of the application of the MLP method in the Mediterranean Sea is provided in Pietropolli et al. (2023), by achieving a lower error in the prediction of **nutrients** through a larger training dataset, a hyperparameter refinement and a two-step quality control of the input data. **Given its potential in predicting nutrient profiles, the MLP-NN model provides a valuable dataset to be used to fill the gap in the availability of in situ observations in data assimilation.**

Line 72: May be worth stating what the first release included for a full account of the developments

Thank you, we will correct as follows:

Starting from the first release that included OC data assimilation in the open ocean (Teruzzi et al., 2014), the assimilation has been progressively developed to handle coastal OC observations (Teruzzi et al., 2018), and chlorophyll and nitrate profiles from BGC-Argo (Cossarini et al. 2019 and Teruzzi et al. 2021, respectively).

**comment9a**

Line 81: Not clear at this point what "sequential modular approach" means

As described in Buizza et al. (2022) the combination of DA and NN can be addressed by 'fusing' the DA and NN modules together or keeping them as independent entities. In this work, we have chosen "*allowing the flexibility of choice between different modules depending on the needs of the overall system.*" (i.e., modular approach). To make this choice clearer in the manuscript, we propose to correct the paragraph as follows (new insertions in bold):

In this paper, the OSE experiment, which combines data assimilation and neural network in a modular approach, aims to quantify how the Argo and BGC-Argo network can be exploited. **The sequential use of the NN and DA schemes provides flexibility in using one module independently of the other, depending on the needs of the overall system (Buizza et al., 2022).** The DA module used in this work is the 3D-VarBio data assimilation scheme described in Teruzzi et al., 2021 and updated to assimilate oxygen BGC-Argo profiles. The NN module is the NN-MLP described in Pietropolli et al., 2023 for the Mediterranean Sea (**hereafter NN-MLP-MED**).

**comment10a**

Line 87-100: The paragraph about the MedSea oceanography feels out of place and may be covered at the beginning of the first results section.

We gently disagree. We believe that a paragraph about Mediterranean Sea oceanography in the Introduction can help the reader to understand the context (semi-enclosed regional sea but with marked internal variabilities) where our combined approach is tested. However, also following a previous comment (**comment1a**) we will change the paragraph to increase its flow as follows.

Spatial and temporal impacts of the OSE are evaluated using classic and new skill performance metrics in three two-year (2017-2018) numerical experiments performed using the MedBFM coupled with the 3DVarBio: a control run (HIND) without assimilation; a multivariate run (DAfl) with assimilation of BGC-Argo chlorophyll, nitrate, and oxygen; and a multivariate run that also assimilates in situ observations and reconstructed ones (DAnn). **Given its particular characteristics and the density of BGC-Argo profiles, the Mediterranean Sea represents an ideal site for OSE experiments to evaluate the potentiality of the BGC-profiles assimilation.**

The Mediterranean Sea is an anti-estuarine…

Line 114 & Figure 1: the flow of information between 3DVarBio and OGSTM-BFM is implied to be one-way, but presumably it's two-way, with OGSTM-BFM fields also an input to 3DVarBio? Also, "3DVarBio" is used in the text, but "3D-VarBio" in the figure (and on line 116) - these should be consistent.

Thank you for noticing the typo. We have modified the arrow in Figure 1 and corrected the "*3D-VarBio*" notation.

[Figure]

Line 150: "preserve optimal values" - a better wording would be "preserve existing values" or "preserve background values", there's no guarantee they're optimal.

Thank you, we propose the following clarification (in bold):

Specifically, for the assimilation of chlorophyll, the VB operator includes a balance scheme that maintains the ratio among the phytoplankton groups and preserves the physiological status of the phytoplankton cells (**i.e., preserve the internal ratios between the chlorophyll, carbon and nutrients as described in Teruzzi et al., 2014**).

Line 152: "spurious assimilation" - please be more specific. "spurious correlations"?

Thank you, we will change the text to be clearer. The localization is used to limit the impact of observations on physically distant state variables to reduce spurious error correlations.

[..] including a localization function to avoid **unrealistic corrections due to possible spurious error covariances in the deepest part of the water column.**

Line 154: "it barely affects other variables" - is it known how model-dependent this finding is? Since the models used here and in Skákala et al. (2022) are very similar, this is a reasonable approach to take here, but it could be worth clarifying that this lack of effect on other variables is in the model, not necessarily the real world.

Thank you for the comment. In the BFM model (which is similar to the ERSEM model used in Skákala et al., 2021), few formulations depend on oxygen concentration, for example nitrification and an oxygen regulating factor for the switch between aerobic and anaerobic conditions for bacterioplankton (Vichi et al., 2004 and 2007, Di Biagio et al., 2022). In general, for simulated values in the water column, the effect of oxygen on these dynamics is very small. On the other hand, oxygen is changed by several and contrasting processes (such as Primary production and respiration) which makes multivariate covariance including oxygen not reliable.

We propose the following clarification (in bold the new text):

VB included only a new direct relation for oxygen (i.e., oxygen assimilation updates only the oxygen itself), given that it has been shown that oxygen barely affects other variables (Skakala et al., 2021). **In the BFM model equations, few formulations depend on oxygen concentration (e.g. nitrification). Indeed, when the euphotic zone of the open ocean is well oxygenated, oxygen dynamics has a limited impact on the biogeochemical cycles.**

Line 161: "we decided to not use different values of error for the two nitrate subsets in order to show the highest potential impact of the OSE." A caveat needs adding either here or in the discussion that as a result of this decision, the assimilation may be non-optimal in terms of fitting the true state (as opposed to just fitting the observations). The same could be said about the lack of accounting for representation error.

Thank you. The nitrate error used in this work is as in Mignot et al., 2019 for the floats (observation error). In the Discussion (L. 418-422), we discussed the choice of using a uniform observation error among the two nitrate subsets. Based on the Reviewer's comment, we will discuss the representative error and possible over-fitting towards the observations. In particular: i) the nitrate error used in this work is an evolution of the one used in Teruzzi et al. (2021) with the addition of a larger error at depth to avoid inconsistency between the deeper part of the assimilated layer (0-600 m) and the lower one; and ii) the representation error was not added in this work, since the results of the previous works demonstrated a good balance between assimilation impacts and over-fitting towards the observations (instead that true state).

Line 163-4: Is there a reference for the oxygen observation error values used? If not, please state how these values were chosen.

We will add the reference on the oxygen observation error:

Observation error for oxygen is set to 5 mmol m$^{-3}$ in the upper 200 meters of depth and gradually goes to 20 mmol m$^{-3}$ in correspondence of the maximum assimilation depth. These values correspond to the uncertainty associated with the oxygen dataset described in Feudale et al., (2022).

**comment12a)**
Section 2.3: While it is fine to refer the reader to Pietropolli et al. (2023) for details, it would be helpful to have a slightly longer and clearer description of the NN-MLP-MED methodology in this section.
Line 174: "The error of reconstructed nitrate, obtained by using the EMODnet as validation dataset, was 0.5 mmol m−3". As this figure contrasts with the uncertainty value of 0.87 mmol m−3 given in the previous section, a little more context would be useful. For instance, introduce the EMODnet dataset (that hasn't been done yet), state that the NN-MLP-MED method was trained on 80% of the EMODnet data, then had an RMSE of 0.5 mmol

m−3 when tested against the remaining 20% of EMODnet data, and an RMSE of 0.87 mmol m−3 when the methodology is applied to BGC-Argo data that is not in EMODnet (if I have interpreted Pietropolli et al. (2023) correctly).

According to the suggestions of the Reviewers (e.g., comment1c), we propose the following more detailed version of the paragraph (in bold the new text). On the other hand, we will not specify the percentage of data used for training, since we adopted a standard approach (80% for training and 20% for testing)

The NN-MLP-MED (Pietropolli et al., 2023) is the evolution of previous MLP architectures developed to predict low-sampled variables (e.g., nutrients) starting from high-sampled ones (e.g., temperature) (Sauzède et al. 2017, Bittig et al. 2018c, and Fourrier et al. 2020). **NN-MLP-MED is a deterministic Feed-Forward Neural Network based on a MLP structure. The NN-MLP-MED consists of the merging of 10 different MLP architectures, each one with the same input and output features, composed by the same number of hidden layers (i.e., 2), but composed by a different number of neurons per layer. The final prediction resulting from the NN-MLP-MED is the mean of all the predictions of these components.**

**The data flow of this MLP-based approach follows the forward direction from the input to the output layers through the neurons which composed the layers.**

**In our OSE experiment, the trained NN-MLP-MED reconstructs nitrate profiles (output) from temperature and salinity (Argo), oxygen (BGC-Argo) and float date, latitude and longitude (inputs).**

The NN-MLP-MED model presents some novel elements with respect to the mentioned methods (and in particular with respect to Canyon-Med in Fourrier et al. 2020). Firstly, the input dataset includes a larger sample size and wider coverage of the Mediterranean Sea region, **i.e., the quality controlled EMODnet2018_int data collection which integrates the in situ aggregated EMODnet data (Buga et al., 2018) and direct observations (i.e., campaigns) as in Lazzari et al. (2016) and Cossarini et al. (2015b).**

Secondly, the quality of the input dataset benefits from a two-step quality check process, removing noisy and unreliable samples. The neural network architecture was also modified to enhance prediction performance by accurately selecting a performing nonlinear function, adjusting and optimizing the amount of neurons for each layer of the MLP model, and choosing a different optimization strategy to train the algorithm. NN-MLP-MED also includes a vertical smoothing (running mean of 5-10 m window) step and a climatological adjustment at depth 600m that is derived from EMODnet (Salon et al., 2019).

The input nitrate dataset for assimilation contains 938 BGC-Argo profiles and 2146 reconstructed nitrate profiles (Table 1). The reconstructed nitrate profiles are located 61% in the western and 39% in the eastern Mediterranean Sea, thus providing a larger and more homogeneous spatial coverage asin Figures 2.

Uncertainty of reconstructed nitrate associated to the EMODnet validation dataset is 0.5 mmol m−3, while it reaches 0.87 mmol m−3 when it predicts the BGC-Argo dataset (Pietropolli et al., 2023).

comment13a
Line 180: It would be useful to put the information about added reconstructed profiles into context. As a suggestion, that could be in the form of stating

for each aggregated region or the sub-regions how much reconstructed data is added. Having this information about added data per region may be useful in later sections e.g. when looking at RMSE changes between the DA runs, to enable linking the change in coverage to a change in RMSE (or highlighting where this does not link for any reason).

We propose the following figure in order to fill the gaps on the distribution of reconstructed nitrate and nitrate data (by subbasin, by season and by assimilated dataset).

[Figure]

**comment14a)**

Line 184: "Adjusted and delayed mode data were selected for oxygen and chlorophyll, while exclusively DM data were considered for nitrate." - A sentence or two explaining the reasons for these choices would be useful. In particular, what level of drift correction for oxygen has been done in these data sets?

Thank you for the comment. The information we provided is not clear enough. We propose the following changes taking into account all the reviewers' comments (e.g., **comment2b**), moreover the updated Introduction will reply to the Reviewer's comment on oxygen drift correction.

We select AM and DM data of oxygen and chlorophyll, and all DM data for nitrate. **AM data of nitrate are included after being corrected using CanyonB or WOA as explained in Johnson et al. 2021. For the three variables we use measures that are flagged as good, probably good, and interpolated.**

Fig. 2: "of chlorophyll-a (red), Nitrate in situ (orange) and reconstructed Nitrate (grey)" - this may just be a matter of perception, but the colours used don't look like red/orange/grey.

Following the comments of the other reviewers, we propose this revised version of the Figure 2:

[Figure]

Line 196: I may not understand the approach, but what happens if a float lives less than a year, which is when the largest drift occurs (Line 193)? Will the drift correction be applied to t0, or not because this is for operational purposes?

Thank you for the comment. If a float lives less than 1 year the trend analysis is not performed (the profile is never corrected).

Generally, the in situ drift is found after 1 year of the deployment (Bittig at al., 2018, Maurer et al., 2021 and Thierry et al., 2021) and more rarely after months. Consequently, at T0 the drift analysis is never calculated, because we believe that the effect of calibration is still strong.

Following the comments of the other reviewers, we will add more details on sensor drift in the rewritten version of the introduction. In this way, the reader should have enough information about the method. Moreover, we propose to add the following explanation (in bold):

Tests are applied when the life of a float is longer than 1 year. **Conversely, if the available float time series is shorter than 1 year, their profiles are not corrected because the float lifetime is considered too short to account for in situ sensor drift.**

Line 197: Please give more details about the splitting into "inliers and outliers".

Thank you for the comment. Both the RANSAC and Theil Sen methods automatically split the data into inliers and outliers. We can define some

parameters of the methods (e.g., the "number of iterations", "min_sample") but not choose the inliers and outliers. Detailed information can be found at  https://scikit-learn.org/stable/modules/linear_model.html#ransac-regression. We propose to correct the sentence as follows:

Used for linear and non-linear regression problems, the RANSAC and Theil Sen methods **automatically** split the oxygen dataset into a set of inliers and outliers. In order to avoid possible biases (Dang et al. 2008 and Fischler and Bolles 1981), the methods calculate the drift from the inliers set of data.

**comment15a**

Line 201: If drift is expected to linearly increase with depth, why use the average drift between 600 m and 800 m, rather than just the drift at 600 m? This may be reasonable (we're not experts on oxygen sensor drift), but it's not clear from the explanation.

Thank you for the comment. We chose two methods and two depths to obtain a solid basis in evaluating if a float has an oxygen sensor drift. The calculation at two depths avoids the possible fake detection of drifts because of changes in the water masses. Thus, given that the calculations have been done at the two depths we took the arbitrary, but we think more precautionary, choice to use their average. In fact, when the drift is detected (i.e., the four values have the same sign and are higher than 1 mmol/m$^3$ in absolute term), the two values are generally quite close. The sentences are rewritten as follows also taking into account of other comments (e.g., **comment3b** and **comment 4b**):

L. 195 Here, the optode sensor in situ drift is evaluated through non parametric methods (RANSAC and Theil Sen) at two different depths (600 and 800 meters) **to avoid possible fake drift detection because of changes in the water masses.**

[… ]

L. 201 The presence of a drift is established when all four drift estimates **(RANSAC at 600 and 800 meters, Theil-Sen at 600 and 800 meters)** agree in sign and their average value is greater than 1 mmol 200 m$^{-3}$ y. This threshold was chosen on the basis of results in Bittig et al. (2018b). **Then, the drift is removed from the oxygen profiles by setting the computed drift average at 600 meters and linearly interpolating toward the surface, where drift is set equal to zero. According to Thierry and Bittig, 2021, there is a lack of specific tests at depth, while several (i.e., 14) tests applied near-surface are already performed by the GDACs. The presence of near-surface tests motivates our choice to reduce the effect of our correction at the surface.**

Line 207: "Marine Copernicus Service" - "Copernicus Marine Service"
Sorry for the error, we will correct it.

Line 208: "initial conditions from EMODnet dataset (Details are provided in Salon et al. 2019 )." - does this include the same spin-up procedure as in Salon et al.? That should be detailed.
Thank you for the comment. The information are added as follows:
[..] initial conditions from EMODnet dataset (details are provided in Salon et al. 2019 ) and a 3-year spin up using the 2017 forcings in perpetual mode.

Line 216-222: This paragraph needs to be clearer, especially around the oxygen saturation procedure.

Thank you for the comment. Information has been added as follows (new text is in bold) taking into account other reviewers' comments (see **comment16b**):

Before integrating data in the 3D-VarBio, the same pre-assimilation assessment described in Teruzzi et al. (2021) is applied to the chlorophyll profiles. **Nitrate profiles are rejected if concentration at the surface is higher than 3 mmol m−3.**

**At surface, the oxygen profile exclusion is evaluated by calculating the difference between the uppermost oxygen measurement and the oxygen saturation (derived from temperature and salinity data from the Argo dataset as in Garcia and Gordon 1992). Profiles are excluded when this difference reaches the** threshold of 10 mmol m−3.

**At 600 meters, the difference between oxygen and a climatological reference oxygen at depth is calculated. Profiles are excluded when the difference reaches the threshold of 2 times the standard deviation of the same reference dataset. As reference dataset, we chose the EMODnet2018_int data collection** that integrates the in situ aggregated EMODnet data (Buga et al., 2018) and the datasets listed in Lazzari et al. (2016) and Cossarini et al. (2015b). The EMODnet2018_int dataset is available for 16 sub-basins (see Figure 2) in the Mediterranean Sea.

**comment17a**

Line 223-227: How were these thresholds arrived at?

According to comment from the other reviewer (**comment5b**) we will clarify the choice of the threshold and rewrite the paragraph as follows (in bold new text):

L223 During the data assimilation, profiles are excluded when innovation exceeds specific threshold rules. For the chlorophyll the threshold is set to 2 mg m−3, for nitrate, the two thresholds are 2 and 3 mmol m−3 for 0-50 m and 250-600 m layers, respectively (as in Teruzzi et al., 2021). Oxygen thresholds are 30 and 50 mmol m−3 in the 0-150m and 150-600m layers respectively (thresholds are roughly 3 times the standard deviation of the climatology computed on Emodnet data for the different sub-basins). Exceeding values have to be found in at least 5 vertical levels in the specific layers. Exclusions are set to avoid corrections that can trigger unstable dynamics after the assimilation (Teruzzi et al., 2021, Storto et al., 2011, Sakov et al., 2017 and Waller et al , 2018). The excluded profiles ranged from 0.1% for chlorophyll to less than 1% for nitrate.

Additionally, we will change "misfit" with "innovation", introducing the term at old L137:

which relies on the innovation (i.e., the difference between the observations y and the model background xb)

Fig. 3: What does the horizontal line at 600 m represent?

We apologize for the inaccuracy. The horizontal line refers to the standard deviation calculated on the EMODnet dataset at 600 m. We will include the information in the text.

Line 243: "After removing of drift, the deep oxygen concentrations results to be closer to the EMODnet climatological data, allowing to include a higher number of profiles" - does this mean that in the absence of the drift correction the profiles would be expected to fail QC checks and be excluded, rather than the uncorrected profiles being assimilated?

Thank you for the comment. We have not statistically analyzed it, but in most cases the correction makes our data at 600m closer to the EMODnet climatology. As a consequence, we logically have inferred that it is more likely that a profile is excluded when not corrected (drift correction). We propose the following modified text (in bold):

After drift removal, the oxygen concentration at depth is closer to the EMODnet climatological data. **This allowed us to infer that it is likely that the drift correction allows more profiles to be included in the assimilated oxygen datasets (Figure 1).**

==comment18a==

Line 246-247: "While for the satellite comparison the model daily averages are considered, the model first guess (i.e. the model state before the assimilation) is used for metrics based on BGC-Argo." - This is reasonable given that BGC-Argo is assimilated and ocean colour not, but a clearer reasoning for the decision should be given. Furthermore, is the first guess instantaneous (at midnight? at the observation time?) or an average? Also, it states here that for the satellite comparison the model is a daily average, but two paragraphs later that the observations are a weekly average?

Thank you for the observation. The weekly was a typo, we actually used the daily L3 map of satellite chlorophyll from Copernicus. They are given as daily maps thus the comparison uses the model as daily output. The first guess for BGC-Argo comparison is instantaneous at 1pm (i.e. right before the assimilation). Please consider that we will change old L246-247 lines also considering the comments of the other two Reviewers (==comment6b== ==comment7b== ==comment2c==) as follows:

Skills performance of the simulations listed in Table 1 are evaluated by comparing model results with satellite Copernicus OC product **(**i.e., **OCEANCOLOUR_MED_BGC_L3_MY_009_143 from marine.copernicus.eu, last visited in July 2023)** of chlorophyll and BGC-Argo profiles. **The satellite comparison used daily model output. The model first guess (i.e. the model state at 1pm before the assimilation) is instead used for the metrics based on BGC-Argo profiles. While the use of the first guess is a common practice in DA applications (Hollingsworth, et al., 1986), it is worth to remind that this comparison should be considered as a semi-independent validation, given that two consecutive profiles of the same BGC-Argo floats can share a certain degree of correlation.**

Line 248: RMSE has its place, including here, but could usefully be supplemented by other validation statistics. Furthermore, RMSE is only optimal for Gaussian variables, is this the case for the variables considered? If not, then more robust statistics may be preferable.

Following the comments of all reviewers, we aim to clarify RMSE results from Figure 5. We have decided to use RMSE to be consistent with product uncertainties (NN-MLP-MED or previous work such as Teruzzi et al., 2021).

Line 250: "the aggregated combination" was not mentioned before. Could be done with the description of Figure 2.

Thank you for highlighting the issue. We will list all the aggregates in the text. We decided to aggregate the basins to make the results clearer (since 16 sub-basin * 2 seasons * 3 variables makes a high number of profiles).

Line 253 and following: The changes in RMSE should be linked to the change in coverage. From visual inspection, most regions of reduced RMSE are regions of higher pseudo-nitrate (Figure 2), but not all of them e.g. Nwm. Other regions have no (additional) float data yet show changes in the RMSE.
Line 272: "directly ascribed to the increased number…" – this is not clear to me as the Figures do not show how the reconstructed obs are distributed over seasons.
Thank you for the comment. We will add information about the coverage in a table/plot as proposed for ==comment13a== and to make clearer the link between coverage and impacts of DA.

Fig. 5 and associated discussion: it is not at all clear what is displayed in the figure. Absolute values? RMS errors? Percentage RMS errors? Are the x-axis values identical for all variables or have they just been cut off for all except the bottom panel?
We will redo Figure 5 to make results clearer. Please, consider that the x-axis refers to the value of the RMS errors (model vs BGC Argo) for each variable. Moreover, in the present version of the manuscript, the x-axis cannot be seen due to a Latex-formatting error in the figure (we will increase the space between the rows to correct this error).

==comment19a==
Line 281: "Assimilating oxygen profiles enable reducing the model-BGC floats RMSE" - is it possible to know how much this is due to the oxygen assimilation, and how much to the chlorophyll and nitrate assimilation? The lack of impact of reconstructed nitrate is an indicator here, but some further comment would be useful.
Thank you for highlighting this point. The reduction in oxygen RMSE is to be ascribed to the oxygen assimilation. The lack of impact of reconstructed nitrate is mainly due to the fact that reconstructed nitrates come from the oxygen dataset. It means that everytime we assimilate reconstructed nitrate we also assimilate oxygen (that directly updates and affects oxygen dynamics). As commented in ==comment11a== oxygen dynamics is not directly affected by chlorophyll and nitrate increments by the assimilation.

==comment20a==
Section 3.3.1 may benefit from rewriting for clarity. It is difficult to pick out the key message. As a suggestion (definitely not a requirement) you may test describing the BGC differences one region at a time instead of structuring the paragraph by variable. Possibly that improves the understanding.
Thank you for the comment. We prefer to keep the description by variable, unless the reviewer strongly suggests otherwise. However, also taking into account the comments from other reviewers, we will change the order of the figures (i.e., nitrate, phosphate, chlorophyll and oxygen), and we will markedly revise the section to improve its readability. New version of the section reads as follows (as in ==comment9b==):

[revised manuscript text omitted]

Line 302: How do you distinguish if a region is still drifting in Figure 7? To me, ion2 (second column) looks as if it is drifting still, but the differences have smaller magnitude than in Med (third column)
Thank you. We will modify the comment to Figure 7 in agreement with comment20a

comment21a
Figure 10: Experiment names on the y axis differ from the main text. Write "npp" instead of "ppn". I think it would help to include the basin boundaries for orientation. An idea to better visualize the results may be to plot the difference in the subplots for the DA experiments compared to HIND instead of absolute values but that's not a necessary change.
We will correct the typos in Figure 10. Following the suggestions of the Reviewers (see comment10b) and in order to better highlight the NPP differences between the three simulations, we changed the colorbar and added the15 mg/m3 contour line on the last row. The contour refers to the difference between NPP in DAfl and  NPP in DAnn.

[Figure]

Line 340: If I understand correctly, the results thus show that nitrate suggests reduced NPP and chlorophyll enhanced NPP? Does that point to a bias in the model or representation of a specific component? (e.g. PFTs) If that's the case, it may be worth noting in the discussion.

Thank you for the comment. Following the comment of the other Reviewers (comment11b),  we decided to not infer any conclusions on the effect of chlorophyll assimilation on the primary production. We will remove the speculative conclusion on chlorophyll assimilation at L 339-341. The sentence we propose is the following:

The weaker fertilization of the surface layer in DAnn, which occurs for both macronutrients after assimilation (Figures 7 and 8), causes a reduction of the net primary production.

comment22a

Line 346: "0-300 m", Figure 11 says 0-600 m in the title

Thank you for spotting the typo. We used 0-300 and 0-600 m layers for chlorophyll and nitrate respectively. We will correct the text and the equation. We will replace subscript "300" with a more general term "maxdepth" in the equation 2.

In this work, we adopt the impact indicator $I_{ij}(t)$ described in Teruzzi et al. (2021). The impact indicator allows quantifying the integrated response of assimilating BGC Argo profiles with respect to the no assimilative run: [equation_corrected]. HIND is the reference, while Sim refers to one of the different DA setups (DAfl or DAnn). $|Sim_{ij}(t) - HIND_{ij}(t)|$ is the absolute difference between simulations (for each day and grid point), while the subscript maxdepth represents the integral over the 0-300 m and 0-600m for chlorophyll and nitrate respectively.

Line 345: When introducing the impact indicator, please add information about how that differs from other statistical metrics such as RMSE or a simple comparison between fields at the end of the simulation. What is the advantage of using this metric?

Thank you for the comment. We choose the same metrics used in similar biogeochemical assimilation experiments (Teruzzi et al., 2021). We acknowledge that other metrics can be used and that each of them can have strengths and disadvantages.

For example the use of RMSE metric considers only the location of the observation that can be unevenly distributed, thus providing misleading interpretation about the assimilation impact. The comparison between fields at the end of the simulation might limit the significance of the comparison given that many biogeochemical variables, such as chlorophyll, assume low values in December that are not representative of other biogeochemical conditions occurring during the year.

Thus, we think that the metric introduced in Teruzzi et al., 2021, by looking at the 95th of the distribution of the differences, can highlight the largest relative impact in each point of the domain and in different seasons considering the peculiarity of biogeochemical variables.

Line 360: Where does this threshold come from?

Thank you for the question. The threshold is the median value of the impact indicator (see colorbar), as we will clarify in the text:

Compared to a threshold of 0.4 (**the median of the impact indicator in the Mediterranean Sea**), the impacted areas increase from 18.2% to 29.8% in winter and from 10.8% to 14.5 in summer in the DAfl and DAnn runs

Line 368: Do you mean "initial conditions" as in using the analysis to initialise a forecast? If so, that may need clarification because it may be confused with general initial conditions for ocean simulations. For initial conditions in a general sense the QC'd oxygen profiles may not qualify.

Thank you for the comment. We meant the initial condition for simulations. Indeed, once oxygen profiles from BGC-Argo are quality checked (official ADTM QC plus the additional QC proposed in the present work), they can represent a qualified dataset for computing ICs. As shown in Fig 2, more

than 2000 profiles are available in the Mediterranean Sea for the period 2017-2018. However, some areas of the Mediterranean Sea are still undersampled by the BGC-Argo, thus it will require the integration with other in situ datasets.

Line 377: "threshold on 1mmol/m3" – can you add a value for decadal variability in the sentences before, which puts this threshold into context to illustrate it is indeed a justified choice please.
We will add a decadal variability value to provide a term of comparison for the chosen threshold.

comment23a
Line 399: "more than 30 profiles" - what was that before? How much larger is the data availability?
Following the other Reviewer comments (see also comment14b), our OSE experiment shows that the basin coverage rate of nitrate can potentially be as high as the BGC-Argo equipped with an oxygen sensor. We will better explain this concept at L399-400 as follows:
Through the integration of NN and DA, the number of nitrate profiles ingested can potentially be as high as the BGC-Argo equipped with an oxygen sensor (i.e., more than double of the nitrate profiles), which corresponds to a density of 1 profile in each 2.5deg x 2.5deg box every 10 days for the 2017-2018 period.

comment24a
Line 401: "can effectively be constrained" is that referring to previous papers such as observing system simulation experiments? If this is meant as a conclusion from your results, this statement may need more explanation.
Thank you for the comment. The statement "can effectively be constrained" refers to our results. Indeed, by increasing the density of available observations, it was possible to achieve the seasonal temporal scale and the sub-basins spatial scale for nitrate dynamics. We propose the following changes (in bold the new text) also considering comment from the other Reviewer (comment15b):
This means that seasonal sub-basins scale dynamics (e.g., bloom or stratification) can effectively be constrained, while, as stated in d'Ortenzio et al., 2021, mesoscale dynamics is still limited to be only locally studied. **Apart from an increase in the numbers of floats, a further increase of the area impacted from a float assimilation can be achieved by redefining horizontal covariance errors in the data assimilation scheme. Indeed, benefits of non-uniform correlation radius in the horizontal scale have** been previously investigated (Cossarini et al., 2019) and additional improvements could be provided by a 3D varying correlation radius (Storto et al., 2014).

Line 409: The decrease in available BGC Argo observations was not mentioned before, but feels like this should be a major motivation of this work (for the introduction)
Thank you for raising the point. We solved the issues of the BGC Argo availability by providing more accurate information in the Introduction.

Line 415 & 443: "feed-forward" - this term is suddenly introduced in the discussion and conclusion when describing the method used, it should be introduced and explained in the methods section.

We have introduced this information in the Methods following the comments: **comment12a** and **comment1c**

**comment25a**

Line 436: Since ocean colour is not assimilated in this study the statement "should be used in conjunction with…" should have a reference to literature

As discussed in the previous comment, **comment3a**, we will rewrite part of the discussion by adding the following sentences:

Previous works demonstrated the complementary of vertical profiles with respect to satellite ocean colour assimilation (Cossarini et al. 2019, and Ford et al., 2021, Skákala et al., 2021 and Teruzzi et al., 2021)

---

## Author Comment (AC2)

**#REV 2**

egusphere-2023-1588

Combining Neural Networks and Data Assimilation to enhance the spatial impact of Argo floats in the Copernicus Mediterranean biogeochemical model

General comments:

**Scientific significance:**

Reconstruction of nutrient information in CANYON-B based ANN system NN-MLP-MED relies on high accuracy of in situ O2 data while Argo on board O2 sensor is known to suffer significant sensor drift. To mitigate the Argo O2 data drift problem, authors have introduced QC O2 module for further calibration of O2 profile data. This novel approach in conducting secondary O2 calibration is a key component of this study.

Since the pioneering work of Ford et al. (2021), impact of sparsity of BGC Argo profile in ocean state estimation or data assimilation is recognised as a clear issue in BGC Argo profile data assimilation study and operational system. OSE experiment with and without NN-enlarged nitrate profile data for assimilation demonstrated that usefulness of machine-learning retrieved nutrient data can improve model representation of surface phytoplankton dynamics at certain conditions. Impact indicator study reveals clear impact of reconstructed nitrate profile assimilation in model BGC state, especially in the upper macronutrients and chlorophyll-a fields.

**Scientific quality:**

Scientific question raised in this study is clear and important one. Current density of BGC Argo floats array does not cover even basin scale ocean circulation which is an original target of CORE Argo float deployment goal. With the advancement of NN-base BGC variable retrieval methods, it is natural to test if such generated data can help us constrain ocean model for state estimation in operational settings. This study indicates positive impact of such data. However, many evaluation procedures to judge detail impact of the NN-derived nitrate data are not designed effectively to achieve its goal.

We appreciate the constructive comments and suggestions from the Reviewer. We present our point-by-point responses to the Reviewer's comments below. The Reviewer's comments are in blue, our responses follow each comment in black. In each response, we detail the changes we propose to make to the manuscript and include the proposed modified text and/or figure (in red).

For clarity, we have numbered some of the reviewers' comments so that similar ones are aggregated to provide a single response. Comments are labeled and highlighted with specific colors to distinguish reviewers (e.g. Rev. 1: **comment1a**, Rev. 2: **comment1b,** Rev. 3: → **comment1c**).

While the scientific contribution of this study is significant, there is clear problem in how it is delivered as journal paper. Overall, many statements are "speculative" or "subjective" for a data assimilation OSE study and many of statements are not supported directly by provided materials. Typical examples are presentation of RMSEs and difference between HIND and DA experiments in Figure 5 and Figures 6-9. Authors asked reader to read these number from figures rather than presenting actual numbers and the figures are generally presented not adequately for the purposes.

Thank you for the comment. We will take care of the readiness of Figures 5-9 by changing the font size and colors/textures.

**comment1b**

Authors should make clear differences between what can be concluded and what can be speculated from background knowledge. For example, authors discuss "impact of chlorophyll profile assimilation", but OSE setting has only Hindcast (HIND), DA w CHL, O2, NO3 profiles (DAfl) and DA w/ CHL, O2, NO3 profiles plus nn-derived NO3 profiles (DAnn). How can you discuss sole impact of chlorophyll assimilation with this OSE setting? Other cases can be found in the comments under "Specific comments".

We do agree, we will review the text avoiding speculative comments (see also **comment8b**). Here an example:

At Line L274, the positive "impact of chlorophyll profile assimilation" was presented in Teruzzi et al. (2021). In this work, "the impact of chlorophyll profile assimilation" can be assessed by comparing the HIND run with the other two DA setups (Figure 4 and 5). Figure 4 aims to show "*the positive impact that reconstructed nitrate profiles have on phytoplankton at the surface*". Figure 5 aims to show (i) "the positive impact obtained comparing the HIND to both the DA setups" and (ii) "*that chlorophyll assimilation is more effective than dynamical model adjustment after reconstructed nitrate assimilation*".

We will rewrite this sentence as:

L274: Since the DAfl and DAnn simulations share the same chlorophyll assimilative setup, the RMSE improvements with respect to the chlorophyll assimilation can be evaluated by comparing the HIND with DAfl or DAnn simulations (Figure 5 middle panel). For chlorophyll, a reduction of RMSE in DAfl (Figure 5 middle panel) is observed in nwm, ion and lev in winter and at depth in tyr, ion and lev in summer, which is in line to what already shown for nitrate and chlorophyll profiles assimilation in Teruzzi et al. (2021). The positive impact of DAnn on nitrate RMSE (Figure 5 top panel) is not directly transferred to the vertical chlorophyll RMSE (as observed for figure 4). This is due to the fact that direct chlorophyll assimilation is more effective than the dynamical model adjustment after nitrate and reconstructed nitrate assimilation in the areas close to the observed chlorophyll profiles.

While scientific value of this study is high, its presentation quality is rather poor as demonstrated in the long list of comments under "Technical corrections". In general, size of figures and fonts are too small in most of figures. Choice of color scheme in Figure 4 and 5 is questionable for people with color vision deficiencies. Please see guideline on the preparation of graphs: https://www.biogeosciences.net/submission.html. There are many editorial issues ranging from simple wording issue from more serious structural issues. Further detail can be found under "Technical comments". Combining issues raised in "Scientific quality" and "Presentation quality", I recommend major revision. Comments follow below.

Thank you for the comment. We will redo all figures by enlarging the size of the text and when required changing the palette/texture. Proposed changes are detailed in the line-by-line replies to the "Technical Comments". The colors in Figure 4-5 were chosen after testing their effectiveness at the website: Coblis — Color Blindness Simulator. Since the chosen palette could be problematic for the "Green-Blind/Deuteranopia test", we applied different marker styles to the lines in Figure 5. In Figure 4 the legend is in order of appearance of the bars.

**Specific comments:**

P2.l48: model tuning (Wang et al., 2020)

Please add more recent references on this topic:
Yumruktepe et al. 2023 https://gmd.copernicus.org/preprints/gmd-2023-25/
Wang and Fennel 2023 https://agupubs.onlinelibrary.wiley.com/doi/full/10.1029/2022GL101220

Thank you, we will add the references.

P6.l146: "VH is built using a Gaussian filter whose correlation radius modulates the smoothing intensity"
> What is the size of correlation radius in average? This information is important to understand how far BGC Argo profile assimilation leave impact in the analysis.

Thank you for the suggestion. The correlation radius ranges between 12-20 km. Detailed information on the tuning of correlation radius can be found in Figure 3 of Cossarini et al., 2019. We propose to clarify the point with the following sentence:

As in Cossarini et al. (2019), in this work the correlation radius is non-uniform, direction-dependent, and ranges between 12 and 20 km (16 km on average).

**comment2b**

P7.l184: "Adjusted and delayed mode data were selected for oxygen and chlorophyll".
> Can you describe which QC flag was used for selecting "good" data both for oxygen and chlorophyll?

The information will be added also considering other comment (**comment14a**) at old L184 as follows (in bold the new text):

We select AM and DM data of oxygen and chlorophyll, and all DM data for nitrate. **AM data of nitrate are included after being corrected using**

**CanyonB or WOA as explained in Johnson et al. 2021. For the three variables we use measures that are flagged as good, probably good, and interpolated.**

**comment3b**

P8.l199: **"when all four drift estimates agree in sign"**

  > Not clear what do you refer here by "four drift estimates". In P7.l196, it is mentioned that drift is evaluated at two different depths and what are the rest of two estimates? Or the number of four has nothing to do with that?

We chose two methods (RANSAC and Theil Sen) and two depths (600 and 800 m) to obtain a solid basis in evaluating if a float has an oxygen sensor drift, thus "four drift estimates" refer to the two methods applied at two depths. The calculation of the drift with two methods and at two depths avoids possible fake drift detections due to changes in the water masses. The clarification will be added in the new version of the manuscript (see **comment4b**).

**comment4b**

P8.l202-l203: "it can be assumed that O2 values at surface are already fixed by the GDACs"

  > As is stated in p2.l41, not every O2 sensor is calibrated in air and air calibration is one of the most important calibration steps to make O2 data trustable. Do you believe O2 values at surface are fixed even for the old non air calibrated sensor data? Or are old sensor data not included in this specific study period, 2017-2018?

We used data coming from sensors calibrated both in air and in water and for those floats not calibrated in air an additional check on saturation is performed. When oxygen at the surface is far from the value of the oxygen saturation, profiles are excluded as explained in section 2.5. We would like to not add saturation check details in this paragraph that is focused on the novel Oxygen QC procedure. However, we propose to modify the paragraph as follows also taking into account other comments (**comment3b** and **comment15a**), new text in bold):

L. 195 Here, the optode sensor in situ drift is evaluated through non parametric methods (RANSAC and Theil Sen) at two different depths (600 and 800 meters) **to avoid possible fake drift detection because of changes in the water masses.**

[… ]

L. 201 The presence of a drift is established when all four drift estimates **(RANSAC at 600 and 800 meters, Theil-Sen at 600 and 800 meters)** agree in sign and their average value is greater than 1 mmol m−3 y. This threshold is chosen on the basis of results in Bittig et al. (2018b).

**Then, the drift is removed from the oxygen profiles by setting the computed drift average at 600 meters and linearly interpolating toward the surface, where drift is set equal to zero. According to Thierry and Bittig, 2021, there is a lack of specific tests at depth, while several (i.e., 14) tests applied near-surface are already performed by the GDACs. The presence of near-surface tests motivates our choice to reduce the effect of our correction at the surface.**

**comment5b**

P9.l223: "profiles can be excluded when model-observation misfit is higher than given thresholds"
  > Does it mean some profiles are actually excluded during your DA run or this just describes online data selection system? I also assume "model-observation misfit" means innovation, is it right? Can you also justify the reason behind of this online data elimination procedure?

Yes, when innovation exceeds a threshold, we exclude the profile to avoid corrections that can introduce not stable dynamics. This has been discussed in Teruzzi et al., 2021. It is important to remember that this threshold is very permissive, thus very few data have been excluded. The excluded profiles for checks based on innovation ranged from 0.1% for chlorophyll to less than 1% for nitrate.

The text will include the suggestion to use "innovation" and will be changed as follows (in bold new text, as for **comment17a**):

L223 During the data assimilation, **profiles are excluded when innovation exceeds specific threshold rules.** For the chlorophyll the threshold is set to 2 mg m−3, for nitrate, the two thresholds are 2 and 3 mmol m−3 for 0-50 m and 250-600 m layers, respectively **(as in Teruzzi et al., 2021).** Oxygen thresholds are 30 and 50 mmol m−3 in the 0-150m and 150-600m layers respectively (**thresholds are roughly 3 times the standard deviation of the climatology computed on Emodnet data for the different sub-basins**). Exceeding values have to be found in at least 5 vertical levels in the specific layers. Exclusions are set to avoid corrections that can trigger unstable dynamics after the assimilation (Teruzzi et al., 2021, Storto et al., 2011, Sakov et al., 2017 and Waller et al , 2018). The excluded profiles ranged from 0.1% for chlorophyll to less than 1% for nitrate.

Additionally, we will change "misfit" with "innovation", introducing the term at old L137:

which relies on the innovation (i.e., the difference between the observations y and the model background xb)

**comment6b**

P10.l246-247: "While for the satellite comparison the model daily averages .. the model first guess is used for metrics based on BGC-Argo"
  > I believe the choice of these different RMSE metrices between satellite OC data and Argo profiles are based on the experiment settings of Argo profiles being assimilated while satellite OC data are not assimilated. By choosing the first guess state to be compared with not yet assimilated Argo profiles, you can use the Argo profiles as independent data. If it were the case, better to describe so here.

Thank you for the comment. Since also the other Reviewers suggested (**comment18a comment7b comment2c**) to clarify this point, we propose the following text to include all the different comments:

Skills performance of the simulations listed in Table 1 are evaluated by comparing model results with satellite Copernicus OC product  (**i.e., OCEANCOLOUR_MED_BGC_L3_MY_009_143 from marine.copernicus.eu, last visited in July 2023)** of chlorophyll and BGC-Argo profiles. **The satellite comparison used daily model output. The model first guess (i.e. the model state at 1pm before the assimilation) is instead used for the metrics based on BGC-Argo profiles. While the use of the first guess is a common practice in DA applications (Hollingsworth, et al., 1986), it is worth to remind that this comparison should be considered as a semi-independent validation, given that two consecutive profiles of the same BGC-Argo floats can share a certain degree of correlation.**

**comment7b**

P10.l251: "Satellite L3 products from Copernicus Marine Service catalogue .."

> Usage of this data set requires proper citation. Plus, this sentence is floating without clear connection in 3.2. Does it mean satellite OC RMSE metric is based on this weekly averaged data? If so, does weekly cycle coincide with an analysis cycle? Please make its significance clear.

Many thanks for spotting this inconsistency in the text. In the present OSE experiment, we used daily L3 maps of satellite chlorophyll from the Copernicus repository for the model validation. The Copernicus OCEANCOLOUR_MED_BGC_L3_MY_009_143 product is given as daily maps, thus the comparison uses the model daily output.  Please consider that we will change L246-247 lines also considering the comments of the other Reviewers as written in the comment above (comment6b).

**comment8b**

P11.l274-l275: "Here, improvements related to chlorophyll assimilation can be observed in nwm, ion and lev in winter and at depth in tyr, ion and lev in summer (Figure 5 middle panel)"
P11.l278: "the direct chlorophyll assimilation is more effective than .."

> Since there is no experiment with only assimilating chlorophyll in this study, it is not easy to point out degree of "improvements related to chlorophyll assimilation" and if direct chlorophyll assimilation is more effective than the dynamical model adjustment after nitrate assimilation. You need to provide extra analysis to support these statements.

Thank you for the comment. We do agree with the Reviewer that our text was in part speculative. The objective of the present work is to assess the impact of the addition of extra nitrate profiles from NN (DAnn). Thus, our present results show that there is no further improvement of RMSE of chlorophyll after DAnn with respect to the improvement shown by the DAfl run. The discussion about the relative impacts of nitrate vs chlorophyll profile assimilation have been already assessed in Cossarini et al., 2019 and Teruzzi et al., 2021. We will change the text of all the speculative conclusions. For example, we propose to change the text at L.274-280 as follows:

 L.274-280

Since the DAfl and DAnn simulations share the same chlorophyll assimilative setup, the RMSE improvements with respect to the chlorophyll assimilation can be evaluated by comparing the HIND with DAfl or DAnn simulations (Figure 5 middle panel). For chlorophyll, a reduction of RMSE in DAfl (Figure 5 middle panel) is observed in nwm, ion and lev in winter and at depth in tyr, ion and lev in summer, which is in line to what already shown for nitrate and chlorophyll profiles assimilation in Teruzzi et al. (2021). The positive impact of DAnn on nitrate RMSE (Figure 5 top panel) is not directly transferred to the vertical chlorophyll RMSE (as observed for figure 4). This is due to the fact that direct chlorophyll assimilation is more effective than the dynamical model adjustment after nitrate and reconstructed nitrate assimilation in the areas close to the observed chlorophyll profiles.

**comment9b**

P13. "3.3.1 Impacts on biogeochemical vertical dynamics".

> There is no description on how figures 6, 7, 8 and 9 are plotted. Are they sub-basin averaged value? "the basin wide averages of DAnn display .. (Figure 6)" at P13.l310 infers these figures are basin-average, but it is never be stated clearly.

Yes, figures 6, 7, 8 and 9 show averaged values over two selected sub-basins (NMW and Ion2 in map of Figure 2) and the whole Mediterranean Sea. We will add a short explanation of Figure 6,7, 8 and 9 at old L287-289.

Following comments from other reviewers (in comment20a), the new version of the section reads as follows:

[revised manuscript text omitted]

P13.l297: "Nitracline depth"
 > There is no definition of nitracline depth. Please be specific.
We will add the definition directly in the text. The definition of nitracline and phosphocholine will be added also in the captions of the Figure 7 and Figure 8.
At the Mediterranean scale, the nitrate concentration below the nitracline (i.e., the depth at which the nitrate is 2 mmol/m-3) decreases by 8% and 11% in DAfl and DAnn runs, respectively.

P13.l296-l297: "decreases by 8% and 11% in DAfl and DAnn runs, respectively"
 > Contrary to the clear difference in impact of nitrate assimilation in DAfl and Dann at nwm in Figure 7, RMSE profiles in Figure 5 (especially Summer Nitrate at Nwm) does not show such difference in the two DA experiments. Can you explain why?
Figure 7 is plotted using the model average daily outputs computed over the whole area of the sub-basin, while Figure 5 reports the RMSE statistics computed on model background values of the grid points corresponding to the locations of BGC-Argo nitrate profiles. Moreover, consider that validation uses only BGC-Argo observations and not the reconstructed profiles (see text at lines 266-267). Figure 7 shows the impact of the extra nitrate profiles assimilation over areas that are not observed by the real BGC-Argo, which motivates the inclusion of the figures.

 P13.l299: "eventually reach a stationary phase"
 > What does it mean by a stationary phase and how do you measure it?
Indeed, the term stationary was misleading. We refer to the fact that during the second year, the rate of accumulation of DA impact decreases or is

quite null. The two-year simulation shows that the assimilation corrects the initial model bias in most of the areas (e.g., the most observed sub-basins such as ion2) but that other undersampled areas require more time to be influenced by the assimilation. We propose the following modified version of the sentence (see also comment9b):

Differences between the assimilation and the reference run accumulate over time. **The rate of this accumulation is highest during the first year while during the second year it decreases and the differences remain almost constant in sub-basins with a high number of BGC-Argo and reconstructed profiles (e.g., NWM in Fig. 6).**

P13.l306: "As a consequence of both the direct assimilation of chlorophyll profiles and the dynamical model adjustment after nitrate assimilation"
 > Again, how can you argue a consequence of dynamical model adjustment only from nitrate assimilation with this OSE settings? For example, why would phytoplankton biomass change as a consequence of direct assimilation of chlorophyll not affect chlorophyll concentration in the DCM as a consequence of its dynamical model adjustment? If extra material not provided, this statement is speculative.
Thank you for the comment. The second Hovmoeller of Figure 6 shows the difference between DAfl and HIND runs. This difference is due to the direct impact of chlorophyll assimilation and indirect effects (e.g., model dynamical adjustments) after nitrate assimilation. It is correct that we can not distinguish between the two runs (DAfl and DAnn), however the original sentence was meant to highlight the difference between DAfl and HIND. To avoid any misinterpretation, we will change the sentence in L306 as follows (see also comment9b):

Considering chlorophyll (Figure 8), the main differences between DAfl and HIND are a slightly reduction of the DCM chlorophyll concentration (e.g., variation smaller than 5% with respect to HIND simulation) and a correction of the timing of the surface winter blooms (second row in Figure 8).

P13.l313: "oxygen profiles assimilation (DAfl, second row in Figure 9) provides positive or negative corrections"
 > As is described by authors in the subsequent sentences, changes in phytoplankton biomass also change oxygen through primary production and remineralization process as dynamical model adjustment. Thus, assimilation of chlorophyll and nitrate both have a potential to alter oxygen. How can you judge what can be seen in Figure 9 is sole consequence of oxygen assimilation? This sentence contradicts with statements following about impact if reconstructed nitrate profile assimilation in oxygen.
We do agree with the reviewer. By comparing DAfl and HIND we can only provide an assessment of the overall impact on oxygen of the multivariate (nitrate, chlorophyll and oxygen). We will change the sentence at LL313-314 as follows(see also comment9b):

Corrections on oxygen dynamics after the multivariate assimilation (DAfl, second row in Figure 9) are either positive or negative depending on the area and the period of the year. In particular, corrections are mostly positive in ion2, while the NWM sub-basin shows negative corrections in the subsurface layer and positive ones in the upper layer of the second year.

P13.l316-l318: "The only noticeable difference .. > This is one of the most important findings in this study as an impact of reconstructed Nitrate profile assimilation, but difference between DAfl and DAnn in figure 9 (summer period in NWM) can not be found in RMSE profiles in figure 5 (summer

Oxygen in Nwm) and we can not judge if this difference in DAnn against DAfl is improvement or not. Can you explain why?

Thank you for the comment. Figure 9 is plotted using the model's average daily outputs computed over the whole area of a given sub-basin, while Figure 5 reports the RMSE statistics computed on model background values of the grid points corresponding to the locations of BGC-Argo oxygen profiles. Given that oxygen profiles are assimilated in DAfl, the positive message is that the assimilation of extra nitrate profiles does not degrade the quality of DAfl (with oxygen assimilation). It is reasonable to expect that DAnn can perform at maximum as good as DAfl with respect to oxygen. On the other hand, Figure 9 shows an additional detail: nitrate extra profile assimilation can provide changes to the model dynamics that is wider than oxygen one. This change can impact oxygen in areas distant from the location of oxygen BGC-Argo profiles. The text is changed following all reviewers' comments (see also comment9b).

**comment10b**

P16 entire section of 3.3.2
> Since difference between DAfl and DAnn is almost impossible to see in Figure 10, readers can not confirm what is described in this subsection. Please reevaluate how to present different impact of DA settings in NPP.

Following the suggestions of the Reviewers (see comment21a) and in order to better highlight the NPP differences between the three simulations, we changed the colorbar and added the15 mg/m3 contour line on the last row. The contour refers to the difference between NPP in DAfl and NPP in DAnn.

[Figure]

comment11b

P16.l339-l341: "In fact … after chlorophyll assimilation"

How can you measure that weak negative correction of macronutrients is the main cause of reduced NPP outweighing the effect due to change in phytoplankton biomass after chlorophyll assimilation? As far as I read, there is no concrete material supporting this statement is provided. Unless extra material provided, this statement is speculative.

Thanks for the suggestion. We will remove the speculative conclusion on chlorophyll assimilation in L 339-341.

The weaker fertilization of the surface layer in DAnn, which occurs for both macronutrients after assimilation (Figures 7 and 8), causes a reduction of the net primary production.

P17. 3.3.3

 > In figure 11, figure title indicates Nitrate Iij(t) is evaluated over 0-600m depth range rather than 0-300m depth specified in equation (2). If it were the case, please specify so. If not, please fix the figure titles in figure 11.

Thank you for spotting the typo. We used 0-300 and 0-600 m layers for chlorophyll and nitrate respectively. We will correct the text and the equation title of the figures. We will replace subscript "300" with a more general term "maxdepth" in the equation 2.

In this work, we adopt the impact indicator Iij (t) described in Teruzzi et al. (2021). The impact indicator allows quantifying the integrated response of assimilating BGC Argo profiles with respect to the no assimilative run: [equation_corrected..]. HIND is the reference, while Sim refers to one of the different DA setups (DAfl or DAnn). |Simij (t) − HINDij (t)| is the absolute difference between simulations (for each day and grid point), **while the subscript maxdepth represents the integral over the 0-300 m and 0-600m for chlorophyll and nitrate, respectively.**

P17.l365: "since the same QC oxygen dataset was assimilated in DAfl and DAnn"

 > But the authors just described in P13.l316-l318 that impact of the reconstructed Nitrate is noticeable in oxygen at least at NWM where density of the reconstructed Nitrate is large. Then it does not make sense that you do not see difference in the two DA experiments. Why do you not see the difference in the Iij 95th percentiles maps for oxygen?

Thank you for your comment. We observe low differences ($10^{-3}$) mainly in the NWM, between the two DA experiments (DAfl and DAnn) maps. Since the "*oxygen assimilation updates only the oxygen itself*" *(L 154 and*  comment11a) these differences are due to the reconstructed profiles of nitrate that cause a decrease of productivity, a loss of oxygen production and a loss of remineralization. In the new version of the manuscript, we will elaborate on this topic, also linking to comment13a (the spatio-temporal availability of nitrate and reconstructed nitrate). For sake of clarity, we also provide hereafter the differences between the Impact indicator for DAfl and DAnn of oxygen:

[Figure]

Moreover, as commented in comment11a, in our DA scheme, the oxygen dynamics is not directly affected by chlorophyll and nitrate increments by the assimilation. This explains how low the differences between the DAfl and DAnn oxygen impact maps are (Winter Impact map DAfl left-DANN right):

[Figure]

As written in the text L364-365 the *oxygen impact maps (not shown) are very similar to nitrate DAnn maps:*

[Figure]

PP19.l396-l.397: "In this work, important impacts are also observed in summer for all variables, as a consequence of the increased number of assimilated profiles."
> It is not clear what does it mean by "a consequence of the increased number of assimilated profiles". Increased number of nitrate profile from DAfl to DAnn? Or about something else? As far as I understand, main reason why we see impact of DA in summer in DA experiments in this study compared to Teruzzi et al. (2021) is because satellite OC can not see DCM while Argo float profiles see the signal by multiple sensors. In that sense, you could see the impact of Argo profile assimilation no matter how small or large number of profiles is. Please reevaluate this statement.
Thanks for your comments. By comparing the maps of impact indicators for DAfl and DAnn, we show the potential additional benefit of extra nitrate profiles in a multiplatform data assimilation simulation. We will better explain this concept as follows:
In Teruzzi et al. (2021), results of the impact indicator principally showed the efficiency of ocean color assimilation in constraining chlorophyll dynamics especially during winter and the benefit of profile assimilation during summer. In this work, we show that the potential benefit of profile assimilation in summer would be bigger and wider because of the assimilation of the additional NN-reconstructed nitrate profiles.

comment14b
PP19.l399-l400: "Indeed .. box every 10 days"
> I do not understand which "results" in this study support this statement. Basin coverage rate of BGC-Argo floats equipped with oxygen sensors is simply determined by deployment plan. Or do you like to say that the new O2 QC module prove enough number of O2 profile survives to be ingested to nn module? I read 3.1, but could not get such information. Please be clearer about meaning of this statement.

Our aim here is to highlight that the OSE experiment shows that the basin coverage rate of nitrate can potentially be as high as the BGC-Argo equipped with an oxygen sensor. Considering also comments of the other Reviewer (see also comment23a), we will better explain this concept at L399-400:

Through the integration of NN and DA, the number of nitrate profiles ingested can potentially be as high as the BGC-Argo equipped with an oxygen sensor (i.e., more than double of the nitrate profiles), which corresponds to a density of 1 profile in each 2.5deg x 2.5deg box every 10 days for the 2017-2018 period.

**comment15b**

PP19.l401-l406: "while, up to … by a 3D varying correlation radius (Storto et al., 2014)">

This discussion on improvement in meso-scale dynamics look out of topic and I can not see the reason why it is needed to be discussed here. Especially confusing knowing that 2.5 degree by 2.5 degree horizontal resolution in BGC profiles potentially could be achieved by nn with oxygen profile is far below meso-scale resolving resolution of o (50km).

Thank you for the comment. By redefining horizontal covariance error we can only increase the spatial area in which each float has an impact. We will rephrase as follows considering all the reviews (in comment24a):

Apart from an increase in the numbers of floats, a further increase of the area impacted from a float assimilation can be achieved by redefining horizontal covariance errors in the data assimilation scheme. Indeed, benefits of non-uniform correlation radius in the horizontal scale have been previously investigated (Cossarini et al., 2019) and additional improvements could be provided by a 3D varying correlation radius (Storto et al., 2014).

PP19.l418: "0.50 mmol2 m−3 for nitrate"

> This information should be included in 2.2.

OK, we will add the information.

PP19.l423-l.429: "Indeed … Li et al.(2021)»

MLP base Sauzède et al. (2017) overcame of this issue by adding pressure as input variables in MLP. Why do you believe choosing other NN approach such as 1D CNN is important before using pressure or depth information in MLP-NN-MED?

As shown in Pietropolli et al., 2023 (GMDhttps://doi.org/10.5194/egusphere-2023-1876 ), MLP does not explicitly consider that close points in a profile share information (the back propagation during the training treats two close points in a profile as not-correlated values of the target variable). As a result, a profile reconstructed with MLP and T, S, O2 and pressure input from BGC-Argo can show discontinuities that need to be filtered with additional steps in the procedure (see line L176-179). This potential pitfall is overcome by 1D convolutional NN, which learns explicitly the shape of the vertical profiles during the training, thus exploiting the fact that each point of a profile shares information with its neighbors.

In Pietropolli et al., 2023 there is also a comparison between vertical profiles predicted through MLP-NN-MED and PPCon, which is the proposed 1D

CNN approach. Results demonstrate that changing the architecture leads to more smooth profils, which better approximate the original sampled vertical profiles.

**Technical corrections:**

P1.l20: "The Array for Real-time Geostrophic Oceanography"

> Please do not use this acronym for Argo. It is not official. There is a historical background why it should never be and I have it on the authority of one of the program founders who was present on the day the Argo project was first conceived: "Argo was named as a companion project to the proposed Jason altimetric satellite missions. The words indicating a putative interpretation of the letters Argo, Array for Real-time Geostrophic Oceanography, were created in a jocose moment while celebrating in a bar afterwards. It would be best to let an idea die whose origin was mediated entirely through the action of alcohol. Argo is not and was never meant to be an acronym. It should be written "Argo" and never as "ARGO".

Thanks for the clarification. We will remove it.

P1.l2: "and successfully integrated in"

> "and are successfully integrated in"

Ok, We will correct the sentence.

P7.l182: 2.4 BGC-Argo data and post-deployment oxygen quality control

> I assume subsection 2.4 is about QC-O2 module, but the module name is never referred to in this section but found in the next section, 2.5. Please make it clear that this is about QC-O2.

We will correct this inconsistency.

P7.l184: "and DM data were selected for oxygen and chlorophyll, while exclusively DM data were considered"

> Use only one expression among "delayed mode" or "DM". Not together in this sentence, but the same unification of usage of acronym would be better for sets of "Adjusted/AM" and "Real Time/RT" for the entire this manuscript after RT, AM and DM defined at p2.l32.

Thanks for the comment. We will use the full name the first time and after the acronym.

P8.Figure 2: Coordinate labels font is too small and almost unreadable. Please enlarge its size.

> It is almost impossible to distinguish the three dots in the figure. Please consider using different colors or separate maps for each type of profile.

We propose the following new figure, with new colors, new markerstyles, edges and font-size. We have tested the effectiveness of the colors at: https://www.color-blindness.com/coblis-color-blindness-simulator/.

[Figure]

**BGC-Argo availability for: 2017-2018**

P8.Figure 2: "lev=lev1+lev2+lev3+lev4; ion=ion1+ion2+ion3; tyr=tyr1+tyr2; adr=adr1+adr2; swm=swm1+swm2"
   > As far as I can read, this is the only place where aggregated sub- or macro- basins are defined. This should be properly defined in a table as suggested below. It is also recommended to use either "sub-" or "macro-" for minimizing confusion.
We will add the definition of the basins in the text.

**comment16b**
P9.l217-l220. "Finally, oxygen …"
   > I can guess, this long sentence is hard to understand. Needs reorganization with shorter sentences.
We propose the following rephrased version of the paragraph (as for **comment16a**). In bold new text:
Before integrating data in the 3D-VarBio, the same pre-assimilation assessment described in Teruzzi et al. (2021) is applied to the chlorophyll profiles. **Nitrate profiles are rejected if concentration at the surface is higher than 3 mmol m−3.**

**At surface, the oxygen profile exclusion is evaluated by calculating the difference between the uppermost oxygen measurement and the oxygen saturation (derived from temperature and salinity data from the Argo dataset as in Garcia and Gordon 1992). Profiles are excluded when this difference reaches the** threshold of 10 mmol m−3.
**At 600 meters, the difference between oxygen and a climatological reference oxygen at depth is calculated. Profiles are excluded when the difference reaches the threshold of 2 times the standard deviation of the same reference dataset. As reference dataset, we chose the EMODnet2018_int data collection** that integrates the in situ aggregated EMODnet data (Buga et al., 2018) and the datasets listed in Lazzari et al.

(2016) and Cossarini et al. (2015b). The EMODnet2018_int dataset is available for 16 sub-basins (see Figure 2) in the Mediterranean Sea.

P9.l229-l.230: "The oxygen post-deployment quality check method" and "The post deployment oxygen QC method"
   > I assume again, the QC method is referring QC O2 module described in 2.4 or not? If it were the case, please specify so explicitly. Plus, two different ways to refer the QC O2 module at the title of 3.1 and body of 3.1 is strange.
We will change the title of the Section and clarify this aspect in the introduction.

P10.l248: "is evaluated in winter (from February to April, FMA) and summer (from June to August, JJA)"
   > Since your experiment period is two years from Jan 2017 to Dec 2018, do you use both 2017 and 2018 results for this evaluation?
Yes, we use both 2017 and 2018 results. Following all the reviewers' comments, we will revise this paragraph and add this information as well.

P10.l255. "the eastern sub-basins"
   > Please define which sub-basins (lev1, lev2,…etc) are included in the definition of the eastern sub-basins.

P8.Figure 2 caption: "lev=lev1+lev2+lev3+lev4; ion=ion1+ion2+ion3; tyr=tyr1+tyr2; adr=adr1+adr2; swm=swm1+swm2"
P10.l257: "alb, swm and nwm"
P11.l263: " Alboran, South West Mediterranean, North West Mediterranean, Tyrrhenian, Ionian and Levantine Seas"
P11.l271: "is observed in nwm and tyr (winter) and in ion (summer)."
P11.l275: "in nwm, ion and lev in winter and at depth in tyr, ion and lev in summer"
   > Association of long and short names of each sub-basin such as Alboran (alb), South West Mediterranean (swm) etc. is never clearly defined in this article. Please do in section 2.3 or add extra table to do so.
We will add all the information in the text. In particular we will explaining the sub-basin, the aggregated-basin and East-West basins organization.

P11.Figue 4.
   > Figures are too small that it is hard to distinguish three bars at each domain. Please use larger size of figures.
We will enlarge the figure.

P12.Figure 5.
   > Figures are too small that it is hard to distinguish three profiles especially between DAfl and DAnn. Plus, many figures do not have x axis labels. Please use larger size of figures or reconsider different way of presentation such as scatter plots and tables of RMSEs at selected depths.
Unfortunately, the lack of the x-axisis is due to a Latex-formatting error in the figure. We will increase the space between the rows to avoid this problem. We will also consider how to improve the readability of this figure.

P13.l288: "two sub-basins"
 > Please specify names of "two sub-basins" here before referring Figure 2.
We will specify the "two sub-basins" nwm and ion2.

Figure 6, 7, 8, 9
 > Figures and font sizes are too small.  Be more specific about definitions of the second row and the third row in figure caption.
Taking into account the comments from all the reviewers, we will redo the figures and change the order of them (i.e., nitrate, phosphate, chlorophyll and oxygen). We will markedly revise section 3.3.1 to improve its readability. New version of the section is proposed at: comment9b.

P16. Figure 10. Name of experiments, HIND, DAfi and DAnn are Hind, DaIns and Dasyn in y axis label in the figures. It is confusing.
We will correct the typos in Figure 10.
P17.l348: "Here, HIND is here the reference"
   > "Here, HIND is the reference"
We will correct it.

P19.l367-l.389: Five paragraphs about oxygen QC.
> This information do not fit to "Discussion", but rather should be integrated to 2.4.
We will remove part of these paragraphs (lines 382-386) since it is already presented in section 2.4. However, we think that it is relevant to highlight that the proposed drift assessment does not interfere with multidecadal shifts in the Mediterranean. Indeed, this work on the O2 QC procedure has been presented at several conferences (e.g., ADMT meeting in Hobart 24th of Octubre 2023) and the debate on how to correct oxygen sensor drift is still open. Finally, we would like to keep in the discussion a paragraph concerning one possible further development of the O2 QC procedure (i.e., oxygen drift analysis applied to fixed isopycnals).

P20.l408: "BGC-Argo OS"
 > Please define meaning of OS. Observing system?
We will correct the acronym with the extended name *Observing system*.

---

## Author Comment (AC3)

**#REV 3  Julien Brajard, 10 Oct 2023**

The work presented in this paper is of interest to the community, as outlined by the two Reviewers. Nevertheless, as it is noted by the reviewers, the paper needs to be clarified, and the main message more clearly conveyed. I hope that all the comments and suggestions by the reviewers will help to provide an improved revised version.

We appreciate the constructive comments and suggestions from the Reviewer. We present our point-by-point responses to the Reviewer's comments below. The Reviewer's comments are in blue, our responses follow each comment in black. In each response, we detail the changes we propose to make to the manuscript and include the proposed modified text and/or figure (in red).

For clarity, we have numbered some of the reviewers' comments so that similar ones are aggregated to provide a single response. Comments are labeled and highlighted with specific colors to distinguish reviewers (e.g. Rev. 1: comment1a, Rev. 2: comment1b, Rev. 3: → comment1c).

Other comments:
comment1c

Section 2.3 I agree with Reviewer 1 that details about the neural net approach are missing. Especially the sentence "incorporating nonlinear functions, adjusting neuron count, and optimizing the training algorithm" needs to be expanded, since we could wrongly understand that the Fourier et al. approach does not incorporate nonlinear functions (while in reality, they use the nonlinear sigmoid function).

Ok, we have revised the entire section as follows (also considering suggestions of the other reviewers, e.g., comment12a):

The NN-MLP-MED (Pietropolli et al., 2023) is the evolution of previous MLP architectures developed to predict low-sampled variables (e.g., nutrients) starting from high-sampled ones (e.g., temperature) (Sauzède et al. 2017, Bittig et al. 2018c, and Fourrier et al. 2020). NN-MLP-MED is a deterministic Feed-Forward Neural Network based on a MLP structure. The NN-MLP-MED consists of the merging of 10 different MLP architectures, each one with the same input and output features, composed by the same number of  hidden layers (i.e., 2), but composed by a different number of neurons per layer. The final prediction resulting from the NN-MLP-MED is the mean of all the predictions of these components.

The data flow of this MLP-based approach follows the forward direction from the input to the output layers through the neurons which composed the layers.

In our OSE experiment, the trained NN-MLP-MED reconstructs nitrate profiles (output) from temperature and salinity (Argo), oxygen (BGC-Argo) and float date, latitude and longitude (inputs).

The NN-MLP-MED model presents some novel elements with respect to the mentioned methods (and in particular with respect to Canyon-Med in Fourrier et al. 2020). Firstly, the input dataset includes a larger sample size and wider coverage of the Mediterranean Sea region, i.e., the quality-controlled EMODnet2018_int data collection which integrates the in situ aggregated EMODnet data (Buga et al., 2018) and direct observations (i.e., campaigns) as in Lazzari et al. (2016) and Cossarini et al. (2015b).

Secondly, the quality of the input dataset benefits from a two-step quality check process, removing noisy and unreliable samples. The neural network architecture was also modified to enhance prediction performance by accurately selecting a performing nonlinear function, adjusting and optimizing the amount of neurons for each layer of the MLP model, and choosing a different optimization strategy to train the algorithm. NN-MLP-MED also includes a vertical smoothing (running mean of 5-10 m window) step and a climatological adjustment at depth 600m that is derived from EMODnet (Salon et al., 2019).

The input nitrate dataset for assimilation contains 938 BGC-Argo profiles and 2146 reconstructed nitrate profiles (Table 1). The reconstructed nitrate profiles are located 61% in the western and 39% in the eastern Mediterranean Sea, thus providing a larger and more homogeneous spatial coverage asin Figures 2.

Uncertainty of reconstructed nitrate associated to the EMODnet validation dataset is 0.5 mmol m−3, while it reaches 0.87 mmol m−3 when it predicts the BGC-Argo dataset (Pietropolli et al., 2023).

L246 "the model first guess" does it correspond to the background?
Yes, the first guess is the background. It is the state of the system before the assimilation. Given that BGC-Argo floats have a profiling (or measurement) frequency of nearly 10 days, the first guess corresponds to the 10-day predictions in the local areas around the location of a given profile. Considering also other comments, the text at old L246 will be changed as follows:
"The model first guess (i.e. the model state at 1pm before the assimilation) is instead used for the metrics based on BGC-Argo profiles. "

About the assimilation: how frequent is the assimilation update? Is it 10 days?
Generally, it is 5 days. A very small number of floats and for a limited period of time (less than 20%) can have higher frequency based on the decision of PIs of the single floats. We will improve sentence at L189-191 including temporal frequency information of floats as follows:
All the three BGC variables have a fairly homogeneous spatial coverage between the western and eastern Mediterranean Sea, except for few areas not covered (see Figure 2) and a generally 5-day temporal sampling frequency. Higher sampling frequencies (< 5 days) are registered for the 20% of profiles, while <10% have a daily frequency.

comment2c
About the validation: Can you comment a bit on the choice of using the RMSE between BGC-Argo profile and model first guess as a validation. Since a previous measurement of a BGC-Argo profile was already assimilated, can a new measurement be considered independent? It could be interesting to have a quick discussion about the lagrangian autocorrelation…
Thanks for the comment. Due to the lack of independent in situ data, our validation has used the common practice of comparing the first guess with assimilated observations (Hollingsworth, et al., 1986). Additionally, the BGC-Argo floats generally have a frequency of 10 days, which makes the use of the first guess an evaluation of a 10-day prediction in the local areas around the location of a given profile. Temporal autocorrelation between subsequent profiles for the same floats was evaluated for chlorophyll in Cossarini et al., 2019. Those results (Figure 11 in Cossarini et al., 2019)

showed that the persistence of the corrections has a half decay of about 4–5 days. Given some computation limitations this metric has not been tested for O2 and NO3. We will comment this point at L246-247 as follows (also considering suggestions at ==comment18a== and ==comment6b comment7b==):

Skills performance of the simulations listed in Table 1 are evaluated by comparing model results with satellite Copernicus OC product  (i.e., OCEANCOLOUR_MED_BGC_L3_MY_009_143 from marine.copernicus.eu, last visited in July 2023) of chlorophyll and BGC-Argo profiles. The satellite comparison used daily model output. The model first guess (i.e. the model state at 1pm before the assimilation) is instead used for the metrics based on BGC-Argo profiles. While the use of the first guess is a common practice in DA applications (Hollingsworth, et al., 1986), it is worth to remind that this comparison should be considered as a semi-independent validation, given that two consecutive profiles of the same BGC-Argo floats can share a certain degree of correlation. An analysis of the persistence of the corrections of chlorophyll showed that the half decay is of about 4–5 days (Cossarini et al., 2019), which makes the frequency float sampling (e.g. 10 days) big enough to reduce the risk of autocorrelation.

---

## Author Response (AR1)

Dear Editor,

Thank you for reviewing our manuscript. We have revised the manuscript according to the Reviewer's comments. Below, you will find our point-by-point responses detailing the main changes made in the manuscript according to the Reviewer's comments
The Reviewer's comments are in blue, our responses follow each comment in black and the new rewritten version in *black italic*. Additionally, in our response, we provide the line of insertion referring to the track-changes version of the manuscript.

We would like to sincerely thank all reviewers for dedicating the time and effort necessary to review the manuscript, and provide feedback in such a constructive and useful way.

Best regards,
On behalf of all the authors,
Carolina Amadio

General comments on the new version in response to comments of Review #1, #2 and #3.

We conducted an overall review of the manuscript, focusing on improving the English language and ensuring the avoidance of speculative conclusions (as suggested by the Reviewer 2). All figures have been redone enhancing their readiness (by refining elements such as color palette, font size, etc.).

**#Reviewer 1**

In the Introduction we have markedly revised the text and the flow to enhance its readability. We've further added information about the availability of the BGC-Argo data [Track-changes version at P2 Lines 30-37].

Line 58: May mention here or later that these examples are time series of chlorophyll, while the use of reconstructed nitrate is novel.
Thank you. We've added more information by specifying the variable(s) involved in each cited work focused on reconstructing BGC-datasets from OC [Track-changes version at P3 Lines 83-88] and from BGC-Argo [Track-changes version at P4 Lines 89-94].

Line 72: May be worth stating what the first release included for a full account of the developments
Thank you, we have corrected as follows:
"*Starting from the first release that included OC data assimilation in the open ocean (Teruzzi et al., 2014), the assimilation has progressively developed to handle coastal OC observations (Teruzzi et al., 2018), chlorophyll and nitrate profiles from BGC Argo (Cossarini et al. 2019 and Teruzzi et al. 2021 respectively).*" [Track-changes version at P4 Lines 102-104].

Line 81: Not clear at this point what "sequential modular approach" means

As described in Buizza et al. (2022) the combination of DA and NN can be addressed by 'fusing' the DA and NN modules together or keeping them as independent entities. In this work, we have chosen "*allowing the flexibility of choice between different modules depending on the needs of the overall system.*" (i.e., modular approach). To make this choice clearer in the manuscript, we have corrected the paragraph as follows:

"*In this paper, the OSE experiment, which combines data assimilation and neural network in a modular approach, aims to quantify how the Argo and BGC-Argo network can be exploited. The sequential use of the NN and DA schemes provides flexibility in using one module independently of the other, depending on the needs of the overall system (Buizza et al., 2022). The DA module used in this work is the 3DVarBio data assimilation scheme described in Teruzzi et al. 2021 and updated to assimilate oxygen BGC-Argo profiles. The NN module is the NN-MLP described in Pietropolli et al., 2023 for the Mediterranean Sea (hereafter NN-MLP-MED)*" [Track-changes version at P4 Lines 112-118].

Line 87-100: The paragraph about the MedSea oceanography feels out of place and may be covered at the beginning of the first results section.
We gently disagree. We believe that a paragraph about Mediterranean Sea oceanography in the Introduction can help the reader to understand the context (semi-enclosed regional sea but with marked internal variabilities) where our combined approach is tested.

Line 114 & Figure 1: the flow of information between 3DVarBio and OGSTM-BFM is implied to be one-way, but presumably it's two-way, with OGSTM-BFM fields also an input to 3DVarBio? Also, "3DVarBio" is used in the text, but "3D-VarBio" in the figure (and on line 116) - these should be consistent.
Thank you for noticing the typo. We have modified the arrow in Figure 1 and corrected the "*3D-VarBio*" notation.

Line 150: "preserve optimal values" - a better wording would be "preserve existing values" or "preserve background values", there's no guarantee they're optimal.
Thank you, we added the following clarification:
Specifically, for the assimilation of chlorophyll, the VB operator includes a balance scheme that maintains the ratio among the phytoplankton groups and preserves the physiological status of the phytoplankton cells (i.e., preserves the internal ratios between the chlorophyll, carbon and nutrients as described in Teruzzi et al. 2014). [Track-changes version at P7 Lines 186-189].

Line 152: "spurious assimilation" - please be more specific. "spurious correlations"?
Thank you, we have changed the text to be clearer. The localization is used to limit the impact of observations on physically distant state variables to reduce spurious error correlations.
"*including a localization function to avoid unrealistic corrections due to possible spurious error covariances in the deepest part of the water column*" [Track-changes version at P7 Lines 191-192].

Line 154: "it barely affects other variables" - is it known how model-dependent this finding is? Since the models used here and in Skákala et al. (2022) are very similar, this is a reasonable approach to take here, but it could be worth clarifying that this lack of effect on other variables is in the model, not necessarily the real world.

Thank you for the comment. In the BFM model (which is similar to the ERSEM model used in Skákala et al., 2021), few formulations depend on oxygen concentration, for example nitrification and an oxygen regulating factor for the switch between aerobic and anaerobic conditions for bacterioplankton (Vichi et al., 2004 and 2007, Di Biagio et al., 2022). In general, for simulated values in the water column, the effect of oxygen on these dynamics is very small. On the other hand, oxygen is changed by several and contrasting processes (such as Primary production and respiration) which makes multivariate covariance including oxygen not reliable.

We have proposed the following clarification:

*"VB included only a new direct relation for oxygen (i.e., oxygen assimilation updates only the oxygen itself), given that it has been shown that it barely affects other variables (Skakala et al., 2021). In the BFM model equations, few formulations depend on oxygen concentration (e.g., nitrification). Indeed, when the euphotic zone of the open ocean is well oxygenated, oxygen dynamics has a limited impact on the biogeochemical cycles."* [Track-changes version at P7 Lines 193-196].

Line 161: "we decided to not use different values of error for the two nitrate subsets in order to show the highest potential impact of the OSE." A caveat needs adding either here or in the discussion that as a result of this decision, the assimilation may be non-optimal in terms of fitting the true state (as opposed to just fitting the observations). The same could be said about the lack of accounting for representation error.

Thank you. The nitrate error used in this work is as in Mignot et al., 2019 for the floats (observation error). In the Discussion, we motivated our choice of using a uniform observation error among the two nitrate subsets.

Based on the Reviewer's comment, we have discussed the representative error and possible over-fitting towards the observations. In particular: i) the nitrate error used in this work is an evolution of the one used in Teruzzi et al. (2021) with the addition of a larger error at depth to avoid inconsistency between the deeper part of the assimilated layer (0-600 m) and the lower one; and ii) the representation error was not added in this work, since the results of the previous works demonstrated a good balance between assimilation impacts and over-fitting towards the observations (instead that true state). [Track-changes version at P28 Lines 594-605].

Line 163-4: Is there a reference for the oxygen observation error values used? If not, please state how these values were chosen.

We added the reference on the dataset used to define the oxygen observation error: *Feudale et al., (2022).*

Section 2.3: While it is fine to refer the reader to Pietropolli et al. (2023) for details, it would be helpful to have a slightly longer and clearer description of the NN-MLP-MED methodology in this section.

Line 174: "The error of reconstructed nitrate, obtained by using the EMODnet as validation dataset, was 0.5 mmol m−3". As this figure contrasts with the uncertainty value of 0.87 mmol m−3 given in the previous section, a little more context would be useful. For instance, introduce the EMODnet dataset (that hasn't been done yet), state that the NN-MLP-MED method was trained on 80% of the EMODnet data, then had an RMSE of 0.5 mmol m−3 when tested against the remaining 20% of EMODnet data, and an RMSE of 0.87 mmol m−3 when the methodology is applied to BGC-Argo data that is not in EMODnet (if I have

interpreted Pietropolli et al. (2023) correctly).

According to the suggestions of the Reviewers we propose a more detailed version of the paragraph. On the other hand, we did? not specify the percentage of data used for training, since we adopted a standard approach (80% for training and 20% for testing).

Line 180: It would be useful to put the information about added reconstructed profiles into context. As a suggestion, that could be in the form of stating for each aggregated region or the sub-regions how much reconstructed data is added. Having this information about added data per region may be useful in later sections e.g. when looking at RMSE changes between the DA runs, to enable linking the change in coverage to a change in RMSE (or highlighting where this does not link for any reason).

We have added a new figure (Figure 3 Track-changes version at P10) to fill the gaps of information on the distribution of reconstructed nitrate and nitrate data.

Line 184: "Adjusted and delayed mode data were selected for oxygen and chlorophyll, while exclusively DM data were considered for nitrate." - A sentence or two explaining the reasons for these choices would be useful.

Thank you for the comment. The information we provided is not clear enough. We propose the following changes taking into account all the reviewers' comments:

"*We collected both AM and DM data for oxygen and chlorophyll. For nitrate we selected DM data, while AM data were incorporated after undergoing correction via Canyon-b NN method or using the World Ocean Atlas (WOA18) collection (Garcia et al., 2019) as explained in Johnson et al. (2021). For the three variables we use data flagged as good, probably good, changed and interpolated values (flags 1, 2, 5 and 8)*".* [Track-changes version at P9 Lines:246-250].

Line 196: I may not understand the approach, but what happens if a float lives less than a year, which is when the largest drift occurs (Line 193)? Will the drift correction be applied to t0, or not because this is for operational purposes?

Thank you for the comment. If a float lives less than 1 year the trend analysis is not performed (the profile is never corrected).

"[..] Conversely, if the available float time series is less than one year, the profiles are not corrected because the float lifetime is considered too short to account for in situ sensor drift." [Track-changes version at P11 Lines:272-273].

Line 197: Please give more details about the splitting into "inliers and outliers".

Thank you for the comment. Both the RANSAC and Theil Sen methods automatically split the data into inliers and outliers. We can define some parameters of the methods (e.g., the "number of iterations", "min_sample") but not choose the inliers and outliers. Detailed information can be found at https://scikit-learn.org/stable/modules/linear_model.html#ransac-regression. We have proposed to following correction:

"*Used for linear and non-linear regression problems, the RANSAC and Theil-Sen methods automatically partition the oxygen dataset into inliers and outliers. In order to avoid possible biases (Dang et al. 2008 and Fischler and Bolles 1981), these methods calculate the drift based on the data subset identified as inliers.*" [Track-changes version at P11 Lines:274-276].

Line 201: If drift is expected to linearly increase with depth, why use the average drift between 600 m and 800 m, rather than just the drift at 600 m? This may be reasonable (we're not experts on oxygen sensor drift), but it's not clear from the explanation.

Thank you for the comment. We chose two methods and two depths to obtain a solid basis in evaluating if a float has an oxygen sensor drift. The calculation at two depths avoids the possible fake detection of drifts because of changes in the water masses. Thus, given that the calculations have been done at the two depths we took the arbitrary, but we think more precautionary, choice to use their average. In fact, when the drift is detected (i.e., the four values have the same sign and are higher than 1 mmol/m$^3$ in absolute term), the two values are generally quite close. The new version of the text is deeply changed [Track-changes version at P11 Lines:265-284].

Line 216-222: This paragraph needs to be clearer, especially around the oxygen saturation procedure.

Thank you for the comment. the paragraph was deeply revised:

"At surface, the oxygen profile exclusion is evaluated by calculating the difference between the uppermost oxygen measurement and the oxygen saturation (derived from temperature and salinity data from the Argo dataset as in Garcia et al. 2019). Profiles are
excluded when this difference reaches the threshold of 10 mmol m−3. At 600 meters, the difference between oxygen and a climatological reference oxygen at depth is calculated. Profiles are excluded when the difference reaches the threshold of 2 times the standard deviation of the same reference dataset. As reference dataset, we chose the EMODnet2018_int data collection that integrates the in situ aggregated EMODnet data (Buga et al., 2018)) and the datasets listed in Lazzari et al. (2016) and Cossarini et al. (2015b). The EMODnet2018_int dataset is available for 16 sub-basins in the Mediterranean Sea (Figure 2)" [Track-changes version at P11 Lines:303-311].

Line 223-227: How were these thresholds arrived at?

According to the comments from the other reviewer we have clarified the choice of the threshold and corrected the paragraph. Additionally, we changed "misfit" with "innovation". [Track-changes version at P11 Lines:311-321].

Line 243: "After removing of drift, the deep oxygen concentrations results to be closer to the EMODnet climatological data, allowing to include a higher number of profiles" - does this mean that in the absence of the drift correction the profiles would be expected to fail QC checks and be excluded, rather than the uncorrected profiles being assimilated?

Thank you for the comment. We have not statistically analyzed it, but in most cases the correction makes our data at 600m closer to the EMODnet climatology. As a consequence, we logically have inferred that it is more likely that a profile is excluded when not corrected (drift correction).

*"The removal of drift brings the oxygen concentration at 600 m closer to the EMODnet climatological data (as exemplified in Figure 4, green star). This leads us to infer that our drift correction enables the inclusion of more profiles in the assimilated oxygen datasets"* [Track-changes version at P13 Lines:341-344].

Line 246-247: "While for the satellite comparison the model daily averages are considered, the model first guess (i.e. the model state before the assimilation) is used for metrics based on BGC-Argo." - This is reasonable given that BGC-Argo is assimilated and ocean colour

not, but a clearer reasoning for the decision should be given. Furthermore, is the first guess instantaneous (at midnight? at the observation time?) or an average? Also, it states here that for the satellite comparison the model is a daily average, but two paragraphs later that the observations are a weekly average?

Thank you for the observation. The weekly was a typo, we actually used the daily L3 map of satellite chlorophyll from Copernicus. They are given as daily maps thus the comparison uses the model as daily output. The first guess for BGC-Argo comparison is instantaneous at 1pm (i.e. right before the assimilation). Here the new version:

"*Skill performances of the simulations listed in Table 1 are evaluated by comparing model results with satellite Copernicus300 OC product (i.e.,*
*OCEANCOLOUR_MED_BGC_L3_MY_009_143 from marine.copernicus.eu, last visited in July 2023) of chlorophyll and BGC-Argo profiles (Argo, 2022). The satellite comparison used daily model output. The model first guess (i.e., the model state at 1pm before the assimilation) is instead used for the metrics based on BGC-Argo profiles. While the use of the first guess is a common practice in DA applications (Hollingsworth et al., 1986), it is worth to remind that this comparison should be considered as a semi-independent validation, given that two consecutive profiles of the same BGC-Argo float can share a certain degree of correlation*" [Track-changes version at P13 Lines 346-354]

Line 248: RMSE has its place, including here, but could usefully be supplemented by other validation statistics. Furthermore, RMSE is only optimal for Gaussian variables, is this the case for the variables considered? If not, then more robust statistics may be preferable.

Following the comments of all reviewers, we clarified the RMSE results from Figure 5 (OLD VERSION, Figure 6 Track-changes version ). We have decided to use RMSE to be consistent with product uncertainties (NN-MLP-MED or previous work such as Teruzzi et al., 2021).

Line 250: "the aggregated combination" was not mentioned before. Could be done with the description of Figure 2.

Thank you for highlighting the issue.

Following all the Reviewers comment we provided information about the names of the 16 sub-basins in the Mediterranean Sea, classifying them into eastern or western sub-basins. [Track-changes version at P9 Lines 255-262]. Additionally, we've introduced a new figure and modified Figure 4 (OLD VERSION) by inserting a vertical line to divide west and east sub-basins [Track-changes version at: P10 Figure 3 and Figure 5 P15]. Furthermore, we've described the 6 aggregated basins used in the Results Section [Track-changes version at P13 Lines 357-360].

To enhance clarity, the 16 sub-basins are defined using lowercase letters, while the 6 aggregated basins are defined by uppercase letters.

Line 281: "Assimilating oxygen profiles enable reducing the model-BGC floats RMSE" - is it possible to know how much this is due to the oxygen assimilation, and how much to the chlorophyll and nitrate assimilation? The lack of impact of reconstructed nitrate is an indicator here, but some further comment would be useful.

Thank you for highlighting this point. The reduction in oxygen RMSE is to be ascribed to the oxygen assimilation. The lack of impact of reconstructed nitrate is mainly due to the fact that reconstructed nitrates come from the oxygen dataset. It means that everytime we assimilate

reconstructed nitrate we also assimilate oxygen (that directly updates and affects oxygen dynamics). The oxygen dynamics is not directly affected by chlorophyll and nitrate increments by the assimilation. [Track-changes version at P16Lines 410-412].

Section 3.3.1 may benefit from rewriting for clarity. It is difficult to pick out the key message. As a suggestion (definitely not a requirement) you may test describing the BGC differences one region at a time instead of structuring the paragraph by variable. Possibly that improves the understanding.
Thank you. We have revised Section 3.3.1. (From P18 Track-changes version) and modified the Figure 7-8-9-10.

Figure 10: Experiment names on the y axis differ from the main text. Write "npp" instead of "ppn". I think it would help to include the basin boundaries for orientation. An idea to better visualize the results may be to plot the difference in the subplots for the DA experiments compared to HIND instead of absolute values but that's not a necessary change.
We corrected the typos in Figure 10 (OLD VERSION,Figure 11 new version).

Line 340: If I understand correctly, the results thus show that nitrate suggests reduced NPP and chlorophyll enhanced NPP? Does that point to a bias in the model or representation of a specific component? (e.g. PFTs) If that's the case, it may be worth noting in the discussion.
Thank you for the comment. Following the comment of the other Reviewers, we decided to not infer any conclusions on the effect of chlorophyll assimilation on the primary production. We removed the speculative conclusion on chlorophyll assimilation at L 339-341 (OLD VERSION)

Line 345: When introducing the impact indicator, please add information about how that differs from other statistical metrics such as RMSE or a simple comparison between fields at the end of the simulation. What is the advantage of using this metric?
Thank you for the comment. We choose the same metrics used in similar biogeochemical assimilation experiments (Teruzzi et al., 2021). We acknowledge that other metrics can be used and that each of them can have strengths and disadvantages.
For example the use of RMSE metric considers only the location of the observation that can be unevenly distributed, thus providing misleading interpretation about the assimilation impact. The comparison between fields at the end of the simulation might limit the significance of the comparison given that many biogeochemical variables, such as chlorophyll, assume low values in December that are not representative of other biogeochemical conditions occurring during the year.
Thus, we think that the metric introduced in Teruzzi et al., 2021, by looking at the 95th of the distribution of the differences, can highlight the largest relative impact in each point of the domain and in different seasons considering the peculiarity of biogeochemical variables.

Line 360: Where does this threshold come from?
Thank you for the question. The threshold is the mean value of the impact indicator.

Line 368: Do you mean "initial conditions" as in using the analysis to initialise a forecast? If so, that may need clarification because it may be confused with general initial conditions for

ocean simulations. For initial conditions in a general sense the QC'd oxygen profiles may not qualify.

Thank you for the comment. We meant the initial condition for simulations. Indeed, once oxygen profiles from BGC-Argo are quality checked (official ADTM QC plus the additional QC proposed in the present work), they can represent a qualified dataset for computing ICs. As shown in Figure 2, more than 2000 profiles are available in the Mediterranean Sea for the period 2017-2018. However, some areas of the Mediterranean Sea are still undersampled by the BGC-Argo, thus it will require the integration with other in situ datasets.

Line 377: "threshold on 1mmol/m3" – can you add a value for decadal variability in the sentences before, which puts this threshold into context to illustrate it is indeed a justified choice please.

We added a value of decadal variability in the Discussion section. [Track-changes version at P21 Lines 454].

Line 399: "more than 30 profiles" - what was that before? How much larger is the data availability?

Following the other Reviewer comments, our OSE experiment shows that the basin coverage rate of nitrate can potentially be as high as the BGC-Argo equipped with an oxygen sensor. We explained better this concept as follows:

" *Through the integration of NN and DA, the count of nitrate profiles ingested can potentially be as high as the BGC Argo equipped with an oxygen sensor (i.e., more than double of the nitrate profiles), which corresponds to a density of 1 profile in each 2.5deg x 2.5deg box every 10 days for the 2017-2018 period.*" [Track-changes version at P27 Lines:573-577].

Line 401: "can effectively be constrained" is that referring to previous papers such as observing system simulation experiments? If this is meant as a conclusion from your results, this statement may need more explanation.

Thank you for the comment. The statement "can effectively be constrained" refers to our results. Indeed, by increasing the density of available observations, it was possible to achieve the seasonal temporal scale and the sub-basins spatial scale for nitrate dynamics.
We explained better this concept as follows:

*"Apart from an increase in the numbers of floats, a further increase of the area impacted from a float assimilation can be achieved by redefining horizontal covariance errors in the data assimilation scheme. Indeed, benefits of non-uniform correlation radius in the horizontal scale have been previously investigated (Cossarini et al., 2019) and additional improvements could be provided by a 3D varying correlation radius (Storto et al., 2014)".*
[Track-changes version at P27 Lines:580-583].

Line 436: Since ocean colour is not assimilated in this study the statement "should be used in conjunction with…" should have a reference to literature

As discussed in the previous comment, we revised this part of the discussion. [Track-changes version at P27 Lines:562-572].

**#Reviewer 2**

**Specific comments:**

P6.l146: "VH is built using a Gaussian filter whose correlation radius modulates the smoothing intensity"
> What is the size of correlation radius in average? This information is important to understand how far BGC Argo profile assimilation leave impact in the analysis.

Thank you for the suggestion. We've added the information on the correlation radius. "*The correlation radius ranges between 12-20 km*". [Track-changes version at P7 Lines:184-185].

P7.l184: "Adjusted and delayed mode data were selected for oxygen and chlorophyll".
> Can you describe which QC flag was used for selecting "good" data both for oxygen and chlorophyll?

We've revised the text to include essential details about our criteria for selecting BGC profiles based on their data mode and data flag.

"*We collected both AM and DM data for oxygen and chlorophyll. For nitrate we selected DM data, while AM data were incorporated after undergoing correction via Canyon-b NN method or using the World Ocean Atlas (WOA18) collection (Garcia et al., 2019) as explained in Johnson et al. (2021). For the three variables we use data flagged as good, probably good, changed and interpolated values (flags 1, 2, 5 and 8)*". [Track-changes version at P9 Lines:246-250].

P8.l199: "when all four drift estimates agree in sign"
> Not clear what do you refer here by "four drift estimates". In P7.l196, it is mentioned that drift is evaluated at two different depths and what are the rest of two estimates? Or the number of four has nothing to do with that?

We chose two methods (RANSAC and Theil Sen) and two depths (600 and 800 m).

"*In our approach, the presence of a drift is established when all four drift estimates (RANSAC at 600 and 800 m, Theil-Sen at 600 and 800 m) agree in sign and their average value (D_avg) exceeds 1 mmol m−3 y*" . [Track-changes version at P11 Lines:278-280].

P10.l246-247: "While for the satellite comparison the model daily averages .. the model first guess is used for metrics based on BGC-Argo"
> I believe the choice of these different RMSE metrices between satellite OC data and Argo profiles are based on the experiment settings of Argo profiles being assimilated while satellite OC data are not assimilated. By choosing the first guess state to be compared with not yet assimilated Argo profiles, you can use the Argo profiles as independent data. If it were the case, better to describe so here.

P10.l251: "Satellite L3 products from Copernicus Marine Service catalogue .."
> Usage of this data set requires proper citation. Plus, this sentence is floating without clear connection in 3.2. Does it mean satellite OC RMSE metric is based on this weekly averaged data? If so, does weekly cycle coincide with an analysis cycle? Please make its significance clear.

Thank you for your comments. Considering the suggestions from other Reviewers regarding this point, we've extensively revised this section, merging all comments and deeply revising this part from the old version of the manuscript. Here the new version:

"*Skill performances of the simulations listed in Table 1 are evaluated by comparing model results with satellite Copernicus300 OC product (i.e.,*

*OCEANCOLOUR_MED_BGC_L3_MY_009_143 from marine.copernicus.eu, last visited in July 2023) of chlorophyll and BGC-Argo profiles (Argo, 2022). The satellite comparison used daily model output. The model first guess (i.e., the model state at 1pm before the assimilation) is instead used for the metrics based on BGC-Argo profiles. While the use of the first guess is a common practice in DA applications (Hollingsworth et al., 1986), it is worth to remind that this comparison should be considered as a semi-independent validation, given that two consecutive profiles of the same BGC-Argo float can share a certain degree of correlation*" [Track-changes version at P13 Lines 346-354]

P11.l274-l275: "Here, improvements related to chlorophyll assimilation can be observed in nwm, ion and lev in winter and at depth in tyr, ion and lev in summer (Figure 5 middle panel)"
P11.l278: "the direct chlorophyll assimilation is more effective than .."
  > Since there is no experiment with only assimilating chlorophyll in this study, it is not easy to point out degree of "improvements related to chlorophyll assimilation" and if direct chlorophyll assimilation is more effective than the dynamical model adjustment after nitrate assimilation. You need to provide extra analysis to support these statements.
We do agree with the Reviewer that our text was in part speculative. The objective of the present work is to assess the impact of the addition of extra nitrate profiles from NN (DAnn). Thus, our present results show that there is no further improvement of RMSE of chlorophyll after DAnn with respect to the improvement shown by the DAfl run. The discussion about the relative impacts of nitrate vs chlorophyll profile assimilation have been already assessed in Cossarini et al., 2019 and Teruzzi et al., 2021. We removed the text of all the speculative conclusions.

P13. "3.3.1 Impacts on biogeochemical vertical dynamics".
  > There is no description on how figures 6, 7, 8 and 9 are plotted. Are they sub-basin averaged value? "the basin wide averages of DAnn display .. (Figure 6)" at P13.l310 infers these figures are basin-average, but it is never be stated clearly.
P13.l297: "Nitracline depth"
  > There is no definition of nitracline depth. Please be specific.
Section 3.3.1 has been completely rewritten, including definitions for phosphocline and nitracline. In the Hovmoeller figures, we plotted averaged values over two selected sub-basins (NMW and Ion2) and the entire Mediterranean Sea. Additionally, we've added a brief explanation for Figure 7-10. [Track-changes version at P13-14].

P13.l296-l297: "decreases by 8% and 11% in DAfl and DAnn runs, respectively"
  > Contrary to the clear difference in impact of nitrate assimilation in DAfl and Dann at nwm in Figure 7, RMSE profiles in Figure 5 (especially Summer Nitrate at Nwm) does not show such difference in the two DA experiments. Can you explain why?
Figure 7 (OLD VERSION) is plotted using the model average daily outputs computed over the whole area of the sub-basin, while Figure 5 (OLD VERSION) reports the RMSE statistics computed on model background values of the grid points corresponding to the locations of BGC-Argo nitrate profiles.
*"This representation offers additional details on the vertical impact of the reconstructed nitrate profile assimilation with respect to the validation of Figure 6 that considers only model points corresponding to the location of BGC-Argo profiles. Nwm and ion2 represent distinct trophic conditions in the Mediterranean Sea and are also characterized by high number of*

*assimilated reconstructed nitrate profiles (Figure 3).”* [Track-changes version at P18 Lines 419-423].

P13.l299: “eventually reach a stationary phase”
 > What does it mean by a stationary phase and how do you measure it?
The term stationary was misleading. We refer to the fact that during the second year, the rate of accumulation of DA impact decreases or is quite null. [Track-changes version at P18 Lines 437-439].

P13.l306: “As a consequence of both the direct assimilation of chlorophyll profiles and the dynamical model adjustment after nitrate assimilation”
 > Again, how can you argue a consequence of dynamical model adjustment only from nitrate assimilation with this OSE settings? For example, why would phytoplankton biomass change as a consequence of direct assimilation of chlorophyll not affect chlorophyll concentration in the DCM as a consequence of its dynamical model adjustment? If extra material not provided, this statement is speculative.
Thank you for the comment. The second Hovmoeller of Figure 6 (OLD VERSION) shows the difference between DAfl and HIND runs. This difference is due to the direct impact of chlorophyll assimilation and indirect effects (e.g., model dynamical adjustments) after nitrate assimilation. It is correct that we can not distinguish between the two runs (DAfl and DAnn), however the original sentence was meant to highlight the difference between DAfl and HIND. “Considering chlorophyll (Figure 9), the main difference between DAfl and HIND is a slight reduction of the DCM chlorophyll concentration (e.g., variation smaller than 5% with respect to HIND simulation) and a correction of the timing of the surface winter blooms (second row in Figure 9)”. [Track-changes version at P18 Lines 449-451].

P13.l313: “oxygen profiles assimilation (DAfl, second row in Figure 9) provides positive or negative corrections”
 > As is described by authors in the subsequent sentences, changes in phytoplankton biomass also change oxygen through primary production and remineralization process as dynamical model adjustment. Thus, assimilation of chlorophyll and nitrate both have a potential to alter oxygen. How can you judge what can be seen in Figure 9 is sole consequence of oxygen assimilation? This sentence contradicts with statements following about impact if reconstructed nitrate profile assimilation in oxygen.
We do agree with the reviewer. By comparing DAfl and HIND we can only provide an assessment of the overall impact on oxygen of the multivariate (nitrate, chlorophyll and oxygen). We've made the correction as follows:
*”Corrections on oxygen dynamics after the multivariate assimilation (DAfl, second row in Figure 9) are either positive or negative depending on the area and the period of the year. In particular, corrections are mostly positive in ion2, while the NWM sub-basin shows negative corrections in the subsurface layer and positive ones in the upper layer of the second year.* [Track-changes version at P19 Lines *456-460*].

P13.l316-l318: “The only noticeable difference .. > This is one of the most important findings in this study as an impact of reconstructed Nitrate profile assimilation, but difference between DAfl and DAnn in figure 9 (summer period in NWM) can not be found in RMSE profiles in figure 5 (summer Oxygen in Nwm) and we can not judge if this difference in DAnn against DAfl is improvement or not. Can you explain why?

Thank you for the comment. Figure 9 (OLD VERSION) is plotted using the model's average daily outputs computed over the whole area of a given sub-basin, while Figure 5 (OLD VERSION) reports the RMSE statistics computed on model background values of the grid points corresponding to the locations of BGC-Argo oxygen profiles. Given that oxygen profiles are assimilated in DAfl, the positive message is that the assimilation of extra nitrate profiles does not degrade the quality of DAfl (with oxygen assimilation). It is reasonable to expect that DAnn can perform at maximum as good as DAfl with respect to oxygen. On the other hand, Figure 9 (OLD VERSION) shows an additional detail: nitrate extra profile assimilation can provide changes to the model dynamics that is wider than oxygen one. This change can impact oxygen in areas distant from the location of oxygen BGC-Argo profiles. The text is changed following all reviewers' comments as for the comment above (see above comment labeled as P13.l313 comment).

P16 entire section of 3.3.2
> Since difference between DAfl and DAnn is almost impossible to see in Figure 10, readers can not confirm what is described in this subsection. Please reevaluate how to present different impact of DA settings in NPP.
Following the suggestions of all Reviewers we corrected the readability of Figure 10 (OLD VERSION). We modified and added a contour line in black in the third row to highlight the areas mainly impacted by the different assimilative setup where the difference in Net Primary Production (NPP) between Dann and DAfl exceeds 15 mgC m−2 d−1.  [Track-changes version at P23 Figure 11]

P16.l339-l341: "In fact … after chlorophyll assimilation"
How can you measure that weak negative correction of macronutrients is the main cause of reduced NPP outweighing the effect due to change in phytoplankton biomass after chlorophyll assimilation? As far as I read, there is no concrete material supporting this statement is provided. Unless extra material provided, this statement is speculative.
Thanks for the suggestion. We've removed the speculative conclusion regarding chlorophyll assimilation

P17. 3.3.3
 > In figure 11, figure title indicates Nitrate Iij(t) is evaluated over 0-600m depth range rather than 0-300m depth specified in equation (2). If it were the case, please specify so. If not, please fix the figure titles in figure 11.
Thank you for identifying the typo. We've utilized 0-300m and 0-600m layers for chlorophyll and nitrate, respectively. Additionally, we've replaced the subscript "300" with the more general term "maxdepth" in Equation 2.

P17.l365: "since the same QC oxygen dataset was assimilated in DAfl and DAnn"
 > But the authors just described in P13.l316-l318 that impact of the reconstructed Nitrate is noticeable in oxygen at least at NWM where density of the reconstructed Nitrate is large. Then it does not make sense that you do not see difference in the two DA experiments. Why do you not see the difference in the Iij 95th percentiles maps for oxygen?
Thank you for your comment. We observe low differences (10⁻³) mainly in the NWM, between the two DA experiments (DAfl and DAnn) maps. Since the "*oxygen assimilation updates only the oxygen itself" (L 154 OLD VERSION*) these differences are due to the reconstructed profiles of nitrate that cause a decrease of productivity, a loss of oxygen

production and a loss of remineralization.

Moreover, in our DA scheme, the oxygen dynamics is not directly affected by chlorophyll and nitrate increments by the assimilation. This explains the relatively minimal differences observed between the oxygen impact maps of DAfl and DAnn. In the updated manuscript, we've expanded on this topic. For sake of clarity, we also provide hereafter the differences between the Impact indicator for DAfl and DAnn of oxygen:

[Figure]

PP19.l396-l.397: "In this work, important impacts are also observed in summer for all variables, as a consequence of the increased number of assimilated profiles."

> It is not clear what does it mean by "a consequence of the increased number of assimilated profiles". Increased number of nitrate profile from DAfl to DAnn? Or about something else? As far as I understand, main reason why we see impact of DA in summer in DA experiments in this study compared to Teruzzi et al. (2021) is because satellite OC can not see DCM while Argo float profiles see the signal by multiple sensors. In that sense, you could see the impact of Argo profile assimilation no matter how small or large number of profiles is. Please reevaluate this statement.

Thank you for your comments. By comparing the impact indicator maps for DAfl and DAnn, we illustrate the potential additional benefit of incorporating extra nitrate profiles in a multiplatform data assimilation simulation. We've clarified this concept as follows:

"*Previous findings (Teruzzi et al., 2021) have primarily demonstrated the efficiency of ocean colour assimilation in constraining chlorophyll dynamics especially during winter and the advantages of assimilating BGC-Argo profiles in summer. Our work highlights the larger and more extensive benefits of profile assimilation during summer due to the incorporation of reconstructed nitrate profiles.*"  [Track-changes version at P23 Lines:568-572]

PP19.l399-l400: "Indeed .. box every 10 days"

> I do not understand which "results" in this study support this statement. Basin coverage rate of BGC-Argo floats equipped with oxygen sensors is simply determined by deployment plan. Or do you like to say that the new O2 QC module prove enough number of O2 profile survives to be ingested to nn module? I read 3.1, but could not get such information. Please be clearer about meaning of this statement.

Our aim here is to highlight that the OSE experiment shows that the basin coverage rate of nitrate can potentially be as high as the BGC-Argo equipped with an oxygen sensor. Considering also comments of the other Reviewer, we revised this concept:

" *Through the integration of NN and DA, the count of nitrate profiles ingested can potentially be as high as the BGC Argo equipped with an oxygen sensor (i.e., more than double of the*

*nitrate profiles), which corresponds to a density of 1 profile in each 2.5deg x 2.5deg box every 10 days for the 2017-2018 period.*" [Track-changes version at P27 Lines:573-577]

PP19.l401-l406: "while, up to … by a 3D varying correlation radius (Storto et al., 2014)">
This discussion on improvement in meso-scale dynamics look out of topic and I can not see the reason why it is needed to be discussed here. Especially confusing knowing that 2.5 degree by 2.5 degree horizontal resolution in BGC profiles potentially could be achieved by nn with oxygen profile is far below meso-scale resolving resolution of o (50km).
Thank you for the comment. By redefining horizontal covariance error we can only increase the spatial area in which each float has an impact. We've rephrased this concept, considering all the review comments:
*"Apart from an increase in the numbers of floats, a further increase of the area impacted from a float assimilation can be achieved by redefining horizontal covariance errors in the data assimilation scheme. Indeed, benefits of non-uniform correlation radius in the horizontal scale have been previously investigated (Cossarini et al., 2019) and additional improvements could be provided by a 3D varying correlation radius (Storto et al., 2014)".*
[Track-changes version at P27 Lines:580-583].

PP19.l423-l.429: "Indeed … Li et al.(2021)»
MLP base Sauzède et al. (2017) overcame of this issue by adding pressure as input variables in MLP. Why do you believe choosing other NN approach such as 1D CNN is important before using pressure or depth information in MLP-NN-MED?
As shown in Pietropolli et al., 2023 (GMDhttps://doi.org/10.5194/egusphere-2023-1876 ), MLP does not explicitly consider that close points in a profile share information (the back propagation during the training treats two close points in a profile as not-correlated values of the target variable). As a result, a profile reconstructed with MLP and T, S, O2 and pressure input from BGC-Argo can show discontinuities that need to be filtered with additional steps in the procedure (see line L176-179 OLD VERSION). This potential pitfall is overcome by 1D convolutional NN, which learns  explicitly the shape of the vertical profiles during the training, thus exploiting the fact that each point of a profile shares information with its neighbors.
In Pietropolli et al., 2023 there is also a comparison between vertical profiles predicted through MLP-NN-MED and PPCon, which is the proposed 1D CNN approach. Results demonstrate that changing the architecture leads to more smooth profils, which better approximate the original sampled vertical profiles.

**Technical corrections:**
P7.l182:  2.4 BGC-Argo data and post-deployment oxygen quality control
   > I assume subsection 2.4 is about QC-O2 module, but the module name is never referred to in this section but found in the next section, 2.5. Please make it clear that this is about QC-O2.
We corrected the title of the Section as follows: " *2.4 BGC-Argo data and the post-deployment QC O2 module*" [Track-changes version at P9].

 P10.l248: "is evaluated in winter (from February to April, FMA) and summer (from June to August, JJA)"
   > Since your experiment period is two years from Jan 2017 to Dec 2018, do you use both 2017 and 2018 results for this evaluation?
Yes, we used both 2017 and 2018 results. Following all the reviewers' comments, we revised

this paragraph and added this information as well. [Track-changes version at P13 Lines 354-355].

P10.l255. "the eastern sub-basins"
   > Please define which sub-basins (lev1, lev2,…etc) are included in the definition of the eastern sub-basins.
P8.Figure 2 caption: "lev=lev1+lev2+lev3+lev4; ion=ion1+ion2+ion3; tyr=tyr1+tyr2; adr=adr1+adr2; swm=swm1+swm2"
P10.l257: "alb, swm and nwm"
P11.l263: " Alboran, South West Mediterranean, North West Mediterranean, Tyrrhenian, Ionian and Levantine Seas"
P11.l271: "is observed in nwm and tyr (winter) and in ion (summer)."
P11.l275: "in nwm, ion and lev in winter and at depth in tyr, ion and lev in summer"
   > Association of long and short names of each sub-basin such as Alboran (alb), South West Mediterranean (swm) etc. is never clearly defined in this article. Please do in section 2.3 or add extra table to do so.
We provided information about the names of the 16 sub-basins in the Mediterranean Sea, classifying them into eastern or western sub-basins. [Track-changes version at P9 Lines 255-262]. Additionally, we've introduced a new figure and modified Figure 4 (OLD VERSION) by inserting a vertical line to divide west and east sub-basins [Track-changes version at: P10 Figure 3 and Figure 5 P15]. Furthermore, we've described the 6 aggregated basins used in the Results Section [Track-changes version at P13 Lines 357-360].
To enhance clarity, the 16 sub-basins are defined using lowercase letters, while the 6 aggregated basins are defined by uppercase letters.

P19.l367-l.389: Five paragraphs about oxygen QC.
> This information do not fit to "Discussion", but rather should be integrated to 2.4.
We have reduced the discussion about QC O2.

**#Reviewer 3**
Section 2.3 I agree with Reviewer 1 that details about the neural net approach are missing. Especially the sentence "incorporating nonlinear
functions, adjusting neuron count, and optimizing the training algorithm" needs to be expanded, since we could wrongly understand that the Fourier et al. approach does not incorporate nonlinear functions (while in reality, they use the nonlinear sigmoid function).
Thank you for the comment. We've revised the entire section [Track-changes version at P8 Line 210].

L246 "the model first guess" does it correspond to the background?
- About the assimilation: how frequent is the assimilation update? Is it 10 days?
Yes, the first guess is the background. It is the state of the system before the assimilation. Given that BGC-Argo floats have a profiling (or measurement) frequency of nearly 5 days, the first guess corresponds to the 5-day predictions in the local areas around the location of a given profile. Considering also other comments, the text at old L246 is changed as follows:
"*The satellite comparison used daily model output. The model first guess (i.e., the model state at 1pm before the assimilation) is instead used for the metrics based on BGC-Argo profiles*" [Track-changes version at P13 Lines 349-350].

Furthemore, the information on the BGC-Argo floats profiling frequency:

" [..] *a generally 5-day temporal sampling frequency. Higher sampling frequencies (< 5 days) are registered for the 20% of profiles.*" [Track-changes version at P9 Lines 264-265]

About the validation: Can you comment a bit on the choice of using the RMSE between BGC-Argo profile and model first guess as a validation. Since a previous measurement of a BGC-Argo profile was already assimilated, can a new measurement be considered independent? It could be interesting to have a quick discussion about the lagrangian autocorrelation…

Thanks for the comment. Due to the lack of independent in situ data, our validation has used the common practice of comparing the first guess with assimilated observations (Hollingsworth, et al., 1986). We introduced the information in the new version as follows:

"*Skill performances of the simulations listed in Table 1 are evaluated by comparing model results with satellite Copernicus300 OC product (i.e.,*

*OCEANCOLOUR_MED_BGC_L3_MY_009_143 from marine.copernicus.eu, last visited in July 2023) of chlorophyll and BGC-Argo profiles (Argo, 2022). The satellite comparison used daily model output. The model first guess (i.e., the model state at 1pm before the assimilation) is instead used for the metrics based on BGC-Argo profiles. While the use of the first guess is a common practice in DA applications (Hollingsworth et al., 1986), it is worth to remind that this comparison should be considered as a semi-independent validation, given that two consecutive profiles of the same BGC-Argo float can share a certain degree of correlation*" [Track-changes version at P13 Lines 346-354]

---

## Author Response (AR2)

**We thank the Reviewer for providing feedback. We propose to modify the manuscript according to the comments of the two reviewers as outlined in point-by-point replies. In bold our responses, in blue the actions.**

General comments:

There are some spelling and grammatical errors throughout, and inconsistent use of tenses etc. We believe the journal offers copy editing as standard, so correcting these could potentially wait for that stage – we leave that decision to the editor.

**Thank you for your comment. We have carefully revised the text, paying attention to grammar errors and inconsistent use of tenses, also integrating the Rev#2 typo corrections. If the editor suggests an additional copy editing check at this stage, we will promptly provide it**

Use of sub-basin names is inconsistent (abbreviated or not; e.g. in Section 3.3: Nwm v North Western Mediterranean; eastern sub-basin v ion2); inconsistent use of terms like "reconstructed nitrate profiles" v "recNO3" (e.g. around L327) or names of model runs (e.g. L372 "reference run" which probably refers to ).

**We appreciate the reviewer's comments.**
**-Regarding the division of the Mediterranean basin into 16 sub-basins, we have decided to identify each with lowercase acronyms. Conversely, the 6 aggregated macro basins are represented with capitalized letters. One of the 16 sub-basins coincides with one of the 6 macro basins (i.e., the "North Western Mediterranean"). Thus, it is named nwm and Nwm when referred to as one of the 16 sub-basins or 6 macro basins, respectively. To avoid confusion in section 3.3, we have modified the text and title of figures using only the "nwm" acronym.**

**-"recNO3" and "reconstructed nitrate profiles" are interchangeable terms. We have refined the phrasing for clarity (new text is underlined):**
**"A generalized slight worsening in the assimilated runs can generally be observed during the summer stratification period and especially the Eastern sub-basins. From DAfl to DAnn, the value of RMSE slightly increases in all sub-basins. These values correspond to an average worsening of about 6% in DAfl and 7.5% in DAnn compared to the HIND run.**
**Despite the introduction of a significant number of reconstructed nitrate profiles in some sub-basins (e.g., orange striped lines of nwm and ion2 in Figure 3), this inclusion does not positively impact the summer chlorophyll RMSE at the surface."**

**- We have replaced "reference run" with HIND: "Differences between the assimilation and the HIND run accumulate over time"**

Introduction: For readability and clarity, it would be useful to link the different topics better and state what the gaps and advantages are that you are addressing in the results with the modular approach. The motivation is not clear from the introduction. For example: the transition between DA and NN in the introduction (L65) could be done by stating which gap NN can fill for the DA, i.e. adding reconstructed observations which improving the DA

analysis depends on.

**Thank you for your comment. Here's the rearranged new text:**

"**In recent years, data assimilation (DA) techniques have increasingly incorporated neural network (NN)-based tools. The main strength of NN algorithms lies in their ability to approximate continuous functions (Hornik et al., 1989) in remarkably low computational times. These NN-based tools have been integrated into DA frameworks to tackle various DA challenges, such as bias correction (Kumar et al., 2015; Zhou et al., 2021), reformulation of observation operators (Storto et al., 2021), and cross-calibration (Lary et al., 2018). Furthermore, NN algorithms are frequently used as independent tools, distinct from data assimilation, for generating new products and/or reconstructing datasets (Lary et al., 2018). The use of reconstructed datasets may compensate for potential gaps in observation availability, potentially enhancing the predictive skill of numerical models".**

Introduction: Paragraph about the evolution of MedBFM (L83ff) is very nice now.
**Thank you for the very positive feedback.**

Results: From Section 3.3, you either introduce or summarise the approach taken to show the results of the analysis, which greatly helps the readability of these Sections. It would be great if you could add similar introductions to 3.1 and 3.2, to give an overview of which variables are assessed and how, etc.
**Thank you for your feedback.**

**Section 3.1: no changes will be made to the text.**
**After careful consideration, we believe that Section 3.1 effectively presents the necessary information and thus, we would prefer to keep the text as it is. Specifically, in lines 285-291, we have provided comprehensive statistics regarding the QC O2 module, also comparing our average value with values from literature. Furthermore, in lines 292-295, we have explained the implications of this correction on a single float, and in lines 296-298, we have discussed a collateral effect derived from our approach. We are confident that these sections sufficiently address the relevant aspects of our study.**

**Section 3.2 has been rephrased as follows, L314-315 has been canceled:**
**"Skill performances of the simulations listed in Table 1 are evaluated by comparing model results with (i) the satellite Marine Copernicus OC product (i.e., non-gap-filled L3 product OCEANCOLOUR_MED_BGC_L3_MY_009_143 from marine.copernicus.eu, last visited in July 2023) of chlorophyll and (ii) BGC-Argo profiles of chlorophyll, nitrate, and oxygen (Argo, 2022). The satellite OC L3 products downloaded from the Copernicus Marine Service catalogue are interpolated from 1 km to the 1/24° model resolution.**

**Specifically, we compared the daily model output with the satellite dataset and the model's first guess (i.e., the model state at 1pm before assimilation) with the BGC-Argo profiles. While the use of the first guess is a common practice in data assimilation (Hollingsworth et al., 1986), it is worth to remind that this comparison should be considered as a semi-independent validation, given that two consecutive profiles of the same BGC-Argo float can share a certain degree of correlation in their errors.**

**The Root Mean Square Error (RMSE) metric is chosen to quantify the model capability to reproduce seasonal variability of the main biogeochemical (BGC) processes at the surface (satellite dataset) or along the vertical column (BGC Argo dataset), such as phytoplankton surface bloom and dynamics during water column stratification.**

**Indeed, the RMSE is evaluated during winter (from February to April, FMA) and summer (from June to August, JJA) 2017 and 2018 within 16 sub-basins of the Mediterranean Sea (as described in Section 2.4 and in Figure 2) or in an aggregated combination of them."**

Technical comments:

L49-53 The info in the bracket (L50) interrupts the reading flow and may merit its own sentence. The information is also partially repeated in L52 (">1% per year").
**Thank you for your feedback. We have revised the text as follows:**
**"Despite efforts to correct drift during storage, which may enhance accuracy by 5-10%, it is likely that an in situ (or during deployment) drift is still observed. For instance, Maurer et al. (2021) observed significant drift rates in about 25% of the 126 floats analyzed for the Southern Ocean Carbon and Climate Observations and Modeling (SOCCOM) project. These drift rates spanned a total range of -1.1 to 1.2% per year, with a standard deviation of 0.65% per year. Similarly, Bushinsky et al. (2016) found the presence of significant drift rates in about 70% of the floats deployed in the Northern Pacific Ocean. Notably, both positive and negative drift rates were observed across various studies, including those by Johnson and Claustre (2016), Bushinsky et al. (2016) , Bittig et al. (2018b) and Maurer et al. (2021)."**

L184 "inconsistencies between the deeper (below 600 m) and the lower part of the assimilated layer." It took a few reads to grasp the distinction between "deeper" and "lower" here, suggest rewording for clarity.
**Thank you for your feedback. We have revised the text as follows:**
**"This adjustment aims to prevent inconsistencies between the lower part of the assimilated layer (450-600m) and the deeper layer of the water column (below 600m)."**

L202 "EMODnet" – It would be worth adding one or two sentences describing that data set for those unfamiliar with it, e.g. what data it is based on, if it is gridded or not, etc.
 **We have revised the text as follows:**
**"The NN-MLP-MED introduces several innovative features compared to the mentioned methods (e.g.,  CANYON-Med; Fourrier et al. 2020) leading to improved results.**

**Firstly, the input dataset encompasses a larger sample size and broader coverage of the Mediterranean Sea region. The EMODnet (European Marine Observation and Data Network) data collection, as described by Buga et al. (2018), consists of multi-platform data gathered from different research cruises and monitoring activities in Europe's marine waters and global oceans. This dataset is characterized by its multivariate nature, including various biogeochemical observations such as chlorophyll, nitrate, phosphate, dissolved oxygen, DIC, and alkalinity, collected between 1999 and 2018. Additionally, this dataset is further enriched with in situ observations spanning the**

**period from 1999 to 2016, as detailed in Lazzari et al. (2016) and Cossarini et al. (2015b).**

L214 "a balanced distribution" – in Fig. 3 it looks like there are more summer profiles added than winter profiles rather than similar numbers of profiles in both seasons.
**We have revised the text as follows (new text underlined):**
**"After incorporating the reconstructed profiles (recNO3), the nitrate dataset used for assimilation expands to 2146 profiles from the initial 938 nitrate (NO3) profiles (Table 1). Generated by the NN-MLP-MED module, the reconstructed dataset offers broad spatial coverage across the 16 regions of the Mediterranean Sea (Figure 2), as well as a quite balanced distribution of nitrate data throughout the seasons (Figure 3), with the addition of 218 reconstructed profiles of nitrate in winter and 361 in summer, respectively."**

L248 and elsewhere: "mmol m−3 y" – should this not be "mmol m−3 y-1"?
**Thank you , we have corrected it with: mmol m−3 y−1.**

L249-252 "linearly interpolating" – what is the basis for that? Is there a reference saying the drift has a linear dependence on depth/pressure? "where drift is set equal to zero." – Is this the result of QC at the DACs (mentioned in the following sentence)? These sentences are worth clarifying, in particular, which aspect is from the literature and which is an assumption made in this paper. If drift actually changes non-linearly with depth, then the correction could be introducing a source of error – this should be discussed.
**Thank you. As detailed in line 252 "The presence of near-surface tests motivates our decision to mitigate the correction's impact at the surface" rather than a "linear dependence on depth/pressure" (which we have not mentioned in the text). GDACs perform more than 14 tests before releasing data in AM and DM based on oxygen concentration at the surface. Conversely, specific tests for correction based on oxygen concentrations at depth have not yet been developed by the DACs.**
**Our methodology has been thoroughly discussed in several meetings (e.g., the 24th Argo Data Management Team Meeting in Hobart from October 23-27, 2023) and has been favorably welcomed by the BGC-Argo community. Based on the aforementioned motivations we would not introduce any changes to the text.**

L261 "and the initial conditions of oxygen which are retrieved from BGC-Argo float climatology computed after QC O2 procedure" – the initial conditions must be on the model grid. How is this achieved including QC O2 and where is the BGC-Argo climatology coming from?
**Thank you. We have added the required information as follows:**
**"[...] and the initial oxygen conditions. These conditions are derived from the BGC-Argo dataset by generating 16 climatological profiles of oxygen after performing the QC O2 procedure, and then uniformly assigning them to each grid point of the 16 sub-basins shown in Figure 2."**

L294 "After 2 years, the bias due to the drift reaches..." – As I understand it you perform a drift correction on a profile-to-profile basis. It may be worth stating somewhere if the drift is linear over time?
**As we have learnt at the meetings with researchers from different GDACs, the in situ drift typically reaches its maximum after about one year from the first deployment**

**(with a drift approximately 1%) and the rate of drift remains almost stable from the second year onward. Therefore, we have implemented the criteria that the drift is calculated only if the timeseries is longer than 1 year (L242-243)**

L294 How does the drift behave from one profile to the next and long-term over time?
**The following figure (R1 left) shows the time series of drift values at 600m calculated for the BGC-Argo float 6901765, mainly located in the Aegean Sea (aeg) and the Ionian Sea (ion2), with a few measurements in the Levantine (lev1). The figure R1 (left) exhibits some small oscillations during the initial period and a convergence to 2.5 mmol/m3/y after 2 years from the deployment (around 2017-05). The standard deviation of the 2017-2018 timeseries of the drift rate is 6% (0.15 mmol/m3/y) which is reasonably low compared to the average value of the drift.**

[Figure]

Figure R1. Time series of the drift rate (mmolO2/m3/y) for the BGC-Argo floats 6901765 (left) and 6901764 (right). Drift rate is shown for a given BGC-Argo timeseries starting from 1 Jan 2017 which is more than 1 year after the float deployment (i.e. March 2015 for both floats).

L294 Will the drift continuously exceed the threshold after one profile exceeded it?
**Generally, yes. A nice example is shown in figure R1 left (see previous comment). However, we found a few exceptions (less than 1% of the profiles in the 2017-2018 dataset). These occur when the drift rate is very close to the threshold (1mmol/m3/y). One of the few examples can be seen in Figure R1 (right), where the drift at 600 m fluctuates below and above the threshold four times in the period from November 2017 to March 2018. In these few cases, when applied, the correction is small.**
**no changes will be made to the text.**

L303-306 Thank you for adding this clarification, but it might be worth specifying "a certain degree of correlation in their errors."
**Thank you for suggesting a more accurate wording, we have corrected the sentence as you proposed.**

L315 "a composite weekly average was computed to ensure gap-free maps" – in your response to our previous review you stated "The weekly was a typo, we actually used the daily L3 map of satellite chlorophyll from Copernicus. They are given as daily maps thus the comparison uses the model as daily output." Please modify the text if this is the case, and also clarify if you used a L3 (non-gap-filled) or L4 (gap-filled) product.
**Thank you, we have modified the paragraphs and corrected the inconsistency. The**

**new version of the paragraph is proposed on page n.1 of this document (Skill performances..).**

L325: "RMSE" – do you mean RMSE reduction?
L325: "which increases in all sub-basins" – is "which" referring to the RMSE or the chlorophyll?
**Thank you for the comments. We have revised L325 as follows:**
**"From DAfl to DAnn, the value of RMSE slightly increases in all sub-basins. These values correspond to an average worsening of about 6% in DAfl and 7.5% in DAnn compared to the HIND run."**

L327: "reconstructed nitrates" – reconstructed nitrate profiles
L328 What do you mean by "shallow statistics"?
**Thank you for the feedback. We have revised L327-L328 as follows:**
**"Despite the introduction of a significant number of reconstructed nitrate profiles in some sub-basins (e.g., depicted by the orange striped lines of nwm and ion2 in Figure 3), this inclusion does not positively impact the summer chlorophyll RMSE at the surface."**

L345-348 Fig 6 middle panel does not show any large improvements in the chl statistics, regardless of assimilating chlorophyll profiles or adding more nutrient profiles.
**As explained in L333 "The statistics computed over the aggregate basin provide more robust results (e.g., they are computed over a larger number of profiles) even if possible spatial patterns of the errors can be damped. Thus, this choice might limit the analysis on whether/how different nitrate assimilation setups affect chlorophyll and oxygen dynamics (see Section 3.3)."**
**The purpose of plotting seasonal RMSE in aggregated sub-basins is to demonstrate that the assimilation of reconstructed nitrate profiles does not diminish the model's skill to reproduce bloom and stratification BGC-dynamics. Figures 7-13 aim to explain the enlarged impact experienced by the different DA setups. Based on the aforementioned motivations we would not introduce any changes to the text.**

L351-353 "As discussed in Section 2.2" – In Section 2.2 you described how the oxygen variability and oxygen assimilation does not strongly affect the wider BGC, but here you imply that this means assimilating nitrate does not affect the oxygen strongly, which is the opposite argument. Also relevant to lines 449-451.
**Thanks for the comment. Given that O2 profiles are assimilated at the same location of the NN-nitrate profile assimilated, it is not expected and observed any difference between DAnn and DAfl in terms of oxygen.**
**We have rectified the sentence at line 351-353 as follows:**
**The integration of reconstructed profiles in the DAnn simulation does not significantly affect oxygen dynamics compared to the DAfl simulation, given that oxygen has already been markedly modified by the O2 assimilation occurring at the same location as nitrate NN-reconstructed profiles. Additionally, lines 449-451 have been corrected as follows (new text underlined):  Oxygen impact maps (not shown) are very similar to the nitrate DAnn maps and do not show significant differences between the two DA simulations, since the same QC oxygen dataset was assimilated in DAfl and DAnn and the oxygen assimilation largely overcome any other potential model adjustment after nitrate assimilation.**

L368 "corrects" implies that the bias disappears completely, while the following sentence says there is more correction in Dann. It may be better to say the assimilation "reduces the bias" or something similar.
**ok, we have replaced the unclear use of "corrects" in the sentence with "reduces a general positive bias"**

L405-409 Unclear use of statistical terms: 10% change is "most significant" but 5% change is "negligible". Can you rephrase e.g. using comparative words (like smaller, larger) rather than statistical terms (significant, negligible) please. Also relevant elsewhere in the manuscript.
**Thank you for bringing this issue to our attention. We have corrected as follows:**
**"In the DAfl simulation, the most evident differences in primary production compared to the HIND simulation are located in the Eastern Mediterranean Sea with a decrease of NPP of nearly 10% in the Levantine macro-basin and in the Ionian Sea close to the Greek coast (first and second row of Figure 11). This reduction is particularly pronounced during winter. In the Western Mediterranean the impacts on primary production are less evident in both seasons with a slight reduction (5%) in winter in the Tyrrhenian Sea."**
**Use of "significant" or "negligible" has been carefully revised throughout the text.**

L414-416 The phrasing of this paragraph is ambiguous and took a few reads to be clear of the meaning. Please rephrase for clarity.
**We have modified the text as follows:**
**"As shown in Figure 3, basins lev1 and lev4 have a high number of reconstructed nitrate profiles during both winter and summer seasons. This abundance of reconstructed profiles contributes to an increase in impact in reproducing the NNP dynamics, which is spatially localized. Conversely, lev2 and lev3 the sub-basins dividing basins lev1 from lev4, contain in situ nitrate and lack of reconstructed nitrate profiles. This lack may spatially limit the impacts that assimilating reconstructed nitrate profiles could have on NPP throughout the entire Levantine region (Lev)."**

L435 Is the value of the 95th percentile (i.e. 0.1) different between DAfl and DAnn? If so, doesn't the impact parameter mix the area impacted by the DA as well as the magnitude of the changes?

**Thank you for raising the question about the threshold on I(t). This will help to clarify this aspect. Indeed, the "95th percentile" refers to the impact indicator I(t) for each period, variable and simulation, thus it is not a fixed value but it provides a map (for each period, simulation and variable; Fig. 11 and 12). Being a map that describes the areas with the "largest" (95th percentile) relative differences between DAfl or DAnn and HIND, it is consistent to compare I(t) 95th percentile for each variable and different simulations. It is also worth noting that the 0.1 threshold has been used only to give the reader a visual reference for comparing the maps, whilst the 0.1 value does not play any role in the map calculation. The value of 0.1 is calculated after merging all the DAnn and DAfl values of the I(t) 95th percentile into a unified set of data.**
**We will clarify this aspect as follows:**

**In DAfl, the extent of nitrate Iij (t) 95th above 0.1 (which represents the mean of the 95th percentile impact indicator in the Mediterranean Sea calculated after merging all the DAnn and DAfl Iij (t) 95th values) is 16.5% and 18.7% in winter and in summer respectively, with a clear spatial distribution mapping the density of BGC-Argo floats.**

L445 "impact to almost all the Mediterranean Sea" – A few sentences before you state that the impacted area increases to about 30% when including the reconstructed profiles. How do you conclude from this that the approach has the ability to encompass "almost all" of the MedSea?

**Thank you, we referred to the number of  sub-basins (over 16) involved in this spatial impact and not to the % of impact.** we have rectified as follows:

**"These results suggest that the inclusion of reconstructed nitrate assimilation has the potential to extend its impact across the majority of the 16 sub-basins of the Mediterranean Sea. However, the scarcity or absence of available data for assimilation prevents us from observing an impact in the marginal seas (Adr and Aeg), the southern part of the Ionian (ion1), and Western sub-basins (alb and swm1)."**

L483 "in each 2.5deg x 2.5deg box every 10 days for the 2017-2018 period" – Is this estimate of one float per 2.5x2.5 deg an average of all available floats over the MedSea area? Or a theoretical aim? The distribution of the measured and reconstructed profiles is highly heterogeneous. Wouldn't that affect the necessary number of floats to constrain the BGC? And is that number of one float per 2.5x2.5 deg to constrain the BGC your hypothesis or a result of a previous study (e.g. an OSSE)?

**It is an estimate coming from the resolution of our dataset rather than a theoretical aim or a result of a previous study. Our results show a high level of the impact when the density of float is higher to the proposed number.  In fact we conclude that the uneven distribution of the BGC-Argo float allows that the "mesoscale dynamics can only be locally constrained". For instance, within our BGC dataset, it may be feasible to study the mesoscale dynamics of the nwm sub-basin, while, as noted in Line 494, some sub-basins (alb, Adr, Ion1, and Aegean) are still under-sampled.** no changes will be made to the text.

L486-487 "a further increase of the area impacted from a float assimilation can be achieved by redefining horizontal covariance errors" – such an increase is only desirable if the correlations are real, otherwise the increased "impact" may actual degrade rather than improve the analysis. It would be better to talk about "optimising" this, which would better link with the next sentence.

**Thanks,** we agree and reformulate as follows:

**"Apart from an increase in the numbers of floats, a further increase of the area impacted from a float assimilation can be optimized …"**

L501 "a validation error of 0.50 mmol m−3 for nitrate and 0.87 mmol m−3 when applied to predict BGC-Argo data." For clarity, please rephrase to something like "a validation error of 0.50 mmol m−3 when used to predict nitrate from the EMODnet data set, and 0.87 mmol m−3 when used to predict nitrate from BGC-Argo data."

**Thanks,** we agree and reformulate as follows:

**"The MLP-NN-MED method exhibits a validation error of 0.50 mmol m−3 for nitrate when used to predict nitrate from the EMODnet data set, and 0.87 mmol m−3 when used to predict nitrate from BGC-Argo data (Pietropolli et al., 2023)."**

L504 "Using the same error for both datasets revealed the highest potential impact of the reconstructed nitrate." If the measure of "impact" is best matching the assimilated observations (which is what this sentence implies), then the "highest" impact would be shown by using zero observation error. Please rephrase this sentence.

**Thanks, we agree and reformulate as follows:**

"**Thus, while it is reasonable to assign a higher observation error to NN reconstructed nitrate, applying the same error to both in situ and NN reconstructed datasets has resulted in a potential overestimation of the assimilation impact that can be achieved."**

References: Bittig et al. 2018a and b are identical; spelling of "d'Ortenzio" or "D'ortenzio" in the two references; Vichi et al. 2007 a and b are identical.

**Thank you, we have corrected the reference inconsistencies.**

Report #2

**We thank the Reviewer for providing feedback. We propose to modify the manuscript according to the comments of the two reviewers as outlined in point-by-point replies. In bold our responses, in blue the actions.**

General comments:

Authors answered the reviewer's requests and questions thoroughly and major issues in the previous version of this manuscript were resolved. The improved quality of figures make this article easy to understand. Additional sentences and paragraphs, especially in introduction, and further detail on NN-MLP-MED in section 2 made the objectives of this study and article clearer. However, these additional sentences introduced additional ambiguities and editorial issues at the same time. This article could be published after making some minor corrections as suggested below.

**Thank you for your feedback, which greatly contributed to enhancing the readability of the manuscript. We are pleased to hear that the major issues identified in the previous version have been effectively addressed. We will carefully address the minor corrections you have suggested.**

Scientific/Technical questions and issues:

P4.L104: "Because of its particular characteristics" Not sure what does it mean by term "particular characteristics". If it means what are described in the paragraph.

P4.L107-P4.L119, which part of the characteristics makes the Mediterranean Sea ideal site of the OSE? For example, the presence of season- and domain-dependent DCM and nitracline depths attracts the idea of assimilating BGC profiles since they are not observable from space and commonly subject to relatively large model bias or representativeness error. Please be more specific about this point.

**Thank you for your feedback. Yes, we used the sentence "its particular characteristics," to introduce the following paragraph. However, we have rephrased and added information to emphasize that all the characteristics listed for the Mediterranean Sea were equally important to support the choice of the Mediterranean Sea as area of study.**

**"Given its characterization as a miniature ocean suitable for climate studies and considering the density of BGC-Argo profiles, the Mediterranean Sea represents an ideal site for conducting Observing System Experiment (OSE) studies to assess the feasibility of assimilating BGC-profiles and analyzing their impacts.**

**Indeed the Mediterranean Sea is an anti-estuarine semi-enclosed sea (Pinardi et al., 2015) with a complex overturning circulation. This circulation consists of horizontal mesoscale and sub-basins scale gyre structures, transitional cyclonic and anticyclonic gyres and eddies. These dynamics are influenced by bathymetric features interconnected by currents and jets (Oddo et al., 2009), along with vigorous vertical velocities. Furthermore, the shallow Sicily Strait, with a depth of approximately 500 meters, separates the Western Mediterranean from the Eastern Mediterranean. This geographical feature allows different processes to dominate in each of the two regions and limits exchanges only between surface and intermediate waters (Pinardi et al., 2015).**

**Even from a biogeochemical (BGC) perspective, the Mediterranean Sea can be roughly subdivided into the Western and Eastern Mediterranean sectors, characterized by an oligotrophic West-East gradient. This gradient results in low nutrient availability at the surface, which is generally insufficient to sustain high phytoplankton biomass (Siokou et al., 2010; Marañón et al., 2021). Additionally, there is a deeper nitracline in the east (>120m) compared to the west (<100m).**

**Chlorophyll [......... ] (Dibiagio et al., 2022).**

**While the general dynamics of biogeochemical processes can be summarized in a two-basin gradient, it's important to note that mesoscale and sub-mesoscale events can significantly impact the Mediterranean Sea at the sub-basin scale. These events can create intense local dynamics, such as, such as blooms and water column stratification, which are often associated with eddy activities and peculiar vertical circulation. Reproducing these phenomena in numerical model simulations can be more challenging, as they are prone to encountering high model bias or representativeness error."**

P5.L143-L145: "OGSTM .. it is forced by the output (..) of the NEMO3.2 model .." According to this paragraph, OGSTM solves tracer equations off-line with the output of the NEMO3.2 model. However, BGC tracer equations in BFM require external atmospheric forcing such as PAR. Can you describe the external forcing here?
**Thank you, we added the information as follows:**
**OGSTM solves for advection, diffusion, sinking terms, and considers the effects of the free surface and variable volume-layer effects on tracer transport (Salon et al., 2019). It is forced by output variables such as current, temperature (T), salinity (S), and sea surface height from the NEMO3.6 model (Clementi et al., 2017). OGSTM and NEMO3.6 share the same bathymetry and z*grid configuration, as well as open boundary and river conditions (Coppini et al., 2023). Atmospheric forcing, including solar shortwave irradiance and wind stress, is acquired as 2-D daily fields from the European Centre for Medium-Range WeatherForecasts (ECMWF), as detailed by Salon et al. (2019).**

P6.L156: ".. which relies on the misfit between the model background (xb) and the observations (y) .." This statement and the equation (1) are not correct. I am sorry to miss this mistake in the first review comments. Cost function of 3DVar is weighted sum of two terms, 1) a misfit between model control state variable (xa) and its background estimation (xb) and 2) a misfit between observations (y) and its model correspondent (H(xa)). Please fix the description and equation (1).

**Thank you for highlighting the mistake in the definition. Here the new version:**
**"This function comprises two terms: (i) the misfit between the model background (xb) and the model control state variable, or analysis (i.e., the assimilation result xa) and (ii) the mismatch between the observations (y) and the analysis (xa). Both terms are weighted by their respective error covariance matrices (B and R) as follows:**
**$J(xa) = (xa - xb)^T B^{-1}(xa - xb) + (y - H(xa))^T R^{-1}(y - H(xa))$"**

P7.L199: "from temperature and salinity (Argo), oxygen (BGC Argo) and float date .." As far as I understand, three input variables (temperature, salinity and oxygen) and coordinate information (date, lat and lon) are all from the same BGC Argo profile. If it were the case, this statement is misleading. Should you state simply " from sets of temperature, salinity, oxygen, date, latitude and longitude of the BGC Argo profiles."?.

**Yes data came from the same float. We have corrected as follows:**
**"In our OSE experiment, the trained NN-MLP-MED reconstructs nitrate profiles from sets of temperature, salinity, oxygen, date, latitude and longitude BGC-Argo profiles."**

P14.L346-L348: "This is because the direct ..." This statement is still speculative. As far as I read this article, there is no evidence supporting this statement. The OSE experiment is not designed to measure size of impact of chlorophyll-a assimilation and nitrate assimilation independently to chlorophyll-a profile analysis. Plus, comparison with HIND indicates that assimilation of chlorophyll-a profile itself is not effective to reduce profile chlorophyll RMSE in most of the area except for Lev. I suggest removing this statement.

**we agree and have decided to remove the statement**

Editorial issues:

P2. L49-L51: .. a drift in about 25% (..) and 70% of analyzed floats, respectively. Not clear what differences were found in 25% and 70% of analyzed floats, respectively.

**Following all the reviewers' comments, we have added information and rephrased the paragraph as follows:**
**"Despite efforts to correct drift during storage, which may enhance accuracy by 5-10%, it is likely that an in situ (or during deployment) drift is still observed. For instance, Maurer et al. (2021) observed significant drift rates in about 25% of the 126 floats analyzed for the Southern Ocean Carbon and Climate Observations and Modeling (SOCCOM) project. These drift rates spanned a total range of -1.1 to 1.2% per year, with a standard deviation of 0.65% per year. Similarly, Bushinsky et al. (2016) found the presence of significant drift rates in about 70% of the floats deployed in the Northern Pacific Ocean. Notably, both positive and negative drift rates were observed across various studies, including those by Johnson and Claustre (2016), Bushinsky et al. (2016) , Bittig et al. (2018b) and Maurer et al. (2021)."**

P3. L63: ", and solving problems .." > ", and solve problems .." →**OK**

P3.L78, P8.L219: "Canyon-b" > "CANYON-B" →**OK**

P7.L201: "Canyon-Med" > "CANYON-MED" →**OK**

P4.L99: " oxygen BGC-Argo profiles " > " BGC-Argo oxygen profiles" →**OK**

P4.L103: " BGC-Argo chlorophyll, nitrate, and oxygen" > " BGC-Argo chlorophyll, nitrate, and oxygen profiles" →**OK**

P4.L104: "in situ observations" > "the in situ observations" or "the BGC-Argo profiles" or "the BGC profiles" →**OK**

P4.L104: "reconstructed ones" > "NN reconstructed profiles" for clarity →**OK**

P5.L127: "reconstructed profiles " > "NN reconstructed profiles" for clarity →**OK**

P5.Figure 1. The term "OGSTSM-BFM" appears here for the first time and "OGSTM" and "BFM"are described for the first time in subsection 2.1.
**We enlarged the paragraph introducing the OGSTSM-BFM acronym.**
**"In the following sections, we introduce the components of the MedBFM system, including the transport model (OGSTM, Foujols et al., 2000; Lazzari et al., 2012; and Lazzari et al., 2016) and the biogeochemical flux model (BFM, Vichi et al., 2007a; Vichi et al., 2007b). Additionally, we describe the novel modules, namely the QC O2 procedure and the NN-MLP-MED scheme. Furthermore, we outline the dataset, which comprises BGC-Argo and NN reconstructed datasets, and discuss the revised 3DVarBio approach."**

P5.L139: ".. versions, the BFM, Biogeochemical Flux Model .." > ".. versions of the Biogeochemical Flux Model (BFM) .." →**OK**

P6.L143: "the NEMO3.2 model" > This term appears for the first time here and needs proper citation or explanation. → **OK**

P7.L185: ".. we decided to not use .." > ".. we decided not to use .." →**OK**

P7.L185-L186: "in order to show the highest potential impact of the OSE." > Do you like to say "in order to show the highest potential impact of the NN reconstructed nitrate profiles to the OSE."? → **yes**

P7.L189: "(2002)" > "(2002)." →**OK**

P7.L200: "the mentioned methods" > Not clear which methods it is referring to. Does it refer to paragraph in P3.L75-L82?

**"with respect to the previous CANYON's methods"**

P8.L226: "sub -basins:" > "sub -basins (figure 2):"→**OK**

P8.L233: "All the three BGC variables" > Which three BGC variables it is referring to? A set of (recNO3, NO3 and Chl)? If it were the case, please state "All the three BGC variables (recNO3, NO3 and Chl)" to be more specific.
**OK, we specified the variable in the text**

P9 Figure 2 legend and caption: It is helpful for reader to be indicated that oxygen profiles are assimilated at the location of blue markers here as stated in P14.L353.

**The distribution of oxygen profiles (not directly shown) can be inferred by examining the distribution of the blue dots.**

P13 Figure 5 caption: "chlorophyll RMSE" > "OC chlorophyll RMSE" for clarity.→**OK**

P13.L328: What does it mean by "summer chlorophyll shallow statistics"?
**Thank you for the feedback. We have revised L327-L328 as follows:**
**"Despite the introduction of a significant number of reconstructed nitrate profiles in some sub-basins (e.g., depicted by the orange striped lines of nwm and ion2 in Figure 3), this inclusion does not positively impact the summer chlorophyll RMSE at the surface."**

P24.L502: "higher then the one" > "higher than the one" →**OK**